# A Solvable Attention for Neural Scaling Laws

**Bochen Lyu**[1,2*]    **Di Wang**[3†]    **Zhanxing Zhu** [1◇]
[1]University of Southampton    [2]DataCanvas    [3]Independent Researcher
*bochen.lv@gmail.com  †di.wang.2718@gmail.com  ◇z.zhu@soton.ac.uk

## Abstract

Transformers and many other deep learning models are empirically shown to predictably enhance their performance as a power law in training time, model size, or the number of training data points, which is termed as the neural scaling law. This paper studies this intriguing phenomenon particularly for the transformer architecture in theoretical setups. Specifically, we propose a framework for *linear self-attention*, the underpinning block of transformer without softmax, to learn in an in-context manner, where the corresponding learning dynamics is modeled as a non-linear ordinary differential equation (ODE) system. Furthermore, we establish a procedure to derive a tractable approximate solution for this ODE system by reformulating it as a *Riccati equation*, which allows us to precisely characterize neural scaling laws for  linear self-attention with training time, model size, data size, and the optimal compute. In addition, we reveal that the linear self-attention shares similar neural scaling laws with several other architectures when the context sequence length of the in-context learning is fixed, otherwise it would exhibit a different scaling law of training time.

## 1 Introduction

Large language models (LLMs) (e.g., GPT (Brown et al., 2020) and Llama (Meta, 2024)) have made significant achievements across a variety of tasks, ranging from question answering to decision making. Adopting the transformer architecture (Vaswani et al., 2017), these LLMs are large in the sense of both parameters and training data, e.g., the largest Llama 3 model has 405B parameters and is trained on 15.6T tokens (Meta, 2024). One of the most fantastic phenomena of such LLMs is their continuing performance gaining as the model size and training steps are scaled up. More remarkably, their performance can behave predictably as a power law in the number of parameters, computation or data size (Kaplan et al., 2020; Hoffmann et al., 2022). This impressive power law behavior is termed as *neural scaling laws*.

In particular, for a model with $D$ trainable parameters, neural scaling laws state that the test loss $L(D, t)$ should obey $L(D, t) = E + AD^{-\beta} + Bt^{-\gamma}$ (Kaplan et al., 2020; Hoffmann et al., 2022) where $t$ is the number of optimization steps and $E$ captures the loss for a generative process on the data distribution. Holding across a wide range of orders of magnitude, these neural scaling laws have led to the fundamental belief that autoregressive transformer language models could successively improve their performance when scaling up. Interestingly, they also allow practitioners to determine the trade-off between model size and training time for a fixed compute budget (Hoffmann et al., 2022) or design dataset with clever pruning (Sorscher et al., 2022).

Given the significant role of neural scaling laws, the theoretical understanding of their origin and mechanism such as values of their exponents becomes increasingly important recently. Hutter (2021) designed a linear model that can exhibit power laws and showed that not all data distributions lead to power laws; Maloney et al. (2022) applied the random matrix theory to identify necessary properties of scaling laws and proposed a statistical model that captures the neural scaling laws; Bordelon et al. (2024); Nam et al. (2024) proposed different solvable models to reveal the existence of scaling laws. Although these initial attempts made simplifications on model architectures and data for the purpose of analytical tractability, they largely advanced our understandings of neural scaling laws from the theoretical perspective.

On the other hand, one important aspect commonly absent in these works is that they did not consider the transformer architecture, the universal architecture of current LLMs, which leads the theoretical understanding for neural scaling laws of modern LLMs to be still underexplored. Transformers are special not only because they employ self-attention as the primary component, but also because the way how they perform prediction—an incredible mechanism called in-context learning (Brown et al., 2020; Garg et al., 2023) that can adapt their predictions based on data given in context.

The uniqueness of transformer definitely gives rise to many intriguing questions from a theoretical perspective. *What are the origins of neural scaling laws of transformer? Does transformer induce different neural scaling laws compared to other models? Will in-context learning (e.g., context sequence length) affect neural scaling laws?* Due to the importance of transformer and its neural scaling laws, investigating these questions is of great interest and necessary.

Answering these questions from a theoretical perspective requires a thorough understanding of explicit forms of model predictions during training, which, however, is hard since it typically requires solving non-linear ODEs that usually do not admit closed-form solutions. Towards this direction, Saxe et al. (2014) modelled the learning dynamics of deep linear networks as the logistic differential equation that can be solved exactly, Pinson et al. (2023) solved the dynamics of linear convolution neural networks, and Bordelon et al. (2024) applied a DMFT approach from statistical physics to solve random feature models. For transformers, recently Zhang et al. (2023); Tarzanagh et al. (2024) established the forms of converged parameters in regression and classification settings. However, explicit forms of parameters *along the training trajectory* are still unclear, leading to a gap when investigating neural scaling laws for transformers.

In this paper, we attempt to provide initial answers for the aforementioned questions to fill the gap in part and take a step towards understanding neural scaling laws of LLMs. To conduct an amenable analysis, we focus on the self-attention, which stands at the core of the transformer architecture, in the linear case. We note that linear self-attention has been widely adopted in recent works (von Oswald et al., 2023; Li et al., 2023b; Zhang et al., 2023) to study properties of transformers. Despite that feature learning is absent, it has the advantage of providing the possibility for a clear theoretical characterization. We discuss more related works on learning dynamics, neural scaling laws, and the analysis of in-context learning for (linear) self-attention in Appendix B.

**Our Contributions.**

1. We design *a multitask sparse feature regression (MSFR) problem* for the linear self-attention block to learn in an in-context manner. More importantly, we derive *a tractable solution for linear self-attention* by modelling its in-context learning dynamics in the MSFR problem as a non-linear ODE system and reformulating the system to a set of Riccati equations. This is highly nontrivial since non-linear ODE systems are hard to solve, thus our procedure might be of broad interest.

   This solution captures dynamical behaviors of linear self-attention during training explicitly. To the best of our knowledge, this is the first closed-form solution of self-attention along the training trajectory. We highlight that it can be applied as an interesting proxy for investigating properties of self-attention and transformers due to its analytical tractability.

2. Built upon this solution, we characterize *neural scaling laws of linear self-attention* by varying time, the size of model, or the number of training data points when data obeys a power-law, which then gives us the scaling law in the optimal compute budget. In addition, we are able to characterize the role of context sequence length in neural scaling laws, revealing that if it obeys a different power-law then the time scaling law will be affected, otherwise linear self-attention would share similar neural scaling laws with other models, which well aligns with empirical observations in Kaplan et al. (2020).

## 2 SETUP OF FRAMEWORK

**Notations.** We use $\{1, \ldots, N\}$ to denote all integers between 1 and $N$. For two vectors $\boldsymbol{a}, \boldsymbol{b} \in \mathbb{R}^d$, we use $a_j$ to denote its $j$-th component, $\boldsymbol{a} \odot \boldsymbol{b}$ to denote the elementwise product, $\boldsymbol{a} \cdot \boldsymbol{b}$ to denote the inner product, and $\mathrm{diag}(\boldsymbol{a})$ to denote the $d \times d$ matrix with its diagonal elements equal to $\boldsymbol{a}$. We use $\dot{\boldsymbol{a}}$ to denote the derivative of $\boldsymbol{a}$ with respect to time. We let $\delta_{s,s'}$ be 1 if $s = s'$ and 0 otherwise.

For a matrix $\boldsymbol{A}$, we use $A_{i,j}$ to denote its $i$-th row $j$-th column component. We use $\boldsymbol{a} \sim \mathcal{P}$ to denote that $\boldsymbol{a}$ is sampled from distribution $\mathcal{P}$. We use $\boldsymbol{0}_d \in \mathbb{R}^d$ to denote the zero vector in $\mathbb{R}^d$.

In Section 2.1, we define the problem setting of MSFR problem, and present the concept of in-context learning and the generation of in-context learning data for it in Section 2.2. Finally, in Section 2.3, we describe details of linear self-attention block.

## 2.1 MULTITASK SPARSE FEATURE REGRESSION PROBLEM

There are $\mathcal{N}_s$ different tasks in total. We let $S$ be the random variable of picking a specific task among $\mathcal{N}_s$ tasks and assume that $S$ follows a power law distribution:

$$\mathcal{P}_\alpha(S = s) = Zs^{-\alpha} \tag{1}$$

where $Z = \sum_{s=1}^{\mathcal{N}_s} s^{-\alpha}$ is the normalization constant and $\alpha > 1$. Since we focus on linear self-attention, we assume the existence of a non-linear sparse feature extractor to perform the feature learning. Specifically, for an input data vector $\boldsymbol{x} \in \mathbb{R}^d$ and a task type $s \in \{1, \ldots, \mathcal{N}_s\}$, there exists a unique *feature extractor*

$$\boldsymbol{\phi}(s, \boldsymbol{x}) : \mathbb{R} \times \mathbb{R}^d \mapsto \{-1, 0, 1\}^{\mathcal{N}_s} \in \mathbb{R}^{\mathcal{N}_s}, \tag{2}$$

where only the $s$-th component of $\boldsymbol{\phi}(s, \boldsymbol{x})$ can be nonzero, i.e., $\phi_{s'}(s, \boldsymbol{x}) = \pm\delta_{s',s}$. Furthermore, given task type $s$, we let the strength for task $s$ be $\Lambda_s \in \mathbb{R}$. The target $y \in \mathbb{R}$ is now defined through

$$y(s, \boldsymbol{x}) = \Lambda_s \sum_{k=1}^{\mathcal{N}_s} \phi_k(s, \boldsymbol{x}). \tag{3}$$

We elaborate two properties of this problem before moving on. (i) The reason why this problem is termed as "multitask" is because we have $\mathcal{N}_s$ different tasks such that each has its own task strength $\Lambda_s$ and feature extractor $\boldsymbol{\phi}(s, \boldsymbol{x})$, meaning that the model should learn distinct $\Lambda_s$ for each task. (ii) If we let $\boldsymbol{\Lambda} \in \mathbb{R}^{\mathcal{N}_s}$ be the collection of all task strengths, then the target can be written as $y(s, \boldsymbol{x}) = \boldsymbol{\Lambda} \cdot \boldsymbol{\phi}(s, \boldsymbol{x})$, which is like a linear regression over the feature $\boldsymbol{\phi}(s, \boldsymbol{x})$. Since $\boldsymbol{\phi}(s, \boldsymbol{x})$ is like a one-hot vector, the problem is a "regression with sparse feature". The subtlety lies in that we must rely on all task types to learn the complete $\boldsymbol{\Lambda}$ when compared to standard linear regression. Therefore, our problem is defined as "multitask sparse feature regression".

## 2.2 IN-CONTEXT LEARNING

A remarkable ability of LLMs is that they can perform *in-context learning* to adapt to a specific task given a context in the form of instructions (Brown et al., 2020). More specifically, the goal of in-context learning is to enable a learner (e.g., a transformer) to use the context data to make a prediction for the query data. To incorporate this ability, we focus on in-context learning in this paper, and present its details formally for the MSFR problem (Section 2.1) in this section.

**Generation of in-context data.** Given task type $s$, we let the corresponding context sequence length $\psi_s = \mathcal{F}(s) \in \mathbb{R}$ and task strength $\Lambda_s = \mathcal{G}(s) \in \mathbb{R}$, i.e., each task $s$ has a constant context sequence length $\psi_s$ and a constant task strength $\Lambda_s$ determined by the maps $\mathcal{F}$ and $\mathcal{G}$, respectively. The generation is composed of four parts (see Fig. 1): (i) a task type $s \in \{1, \ldots, \mathcal{N}_s\}$ is first sampled from the distribution $\mathcal{P}_\alpha(S = s)$ (Eq. (1)), which gives us the corresponding context sequence length $\psi_s$ and task strength $\Lambda_s$; (ii) we sample $\psi_s$ different input vectors $\boldsymbol{x} \in \mathbb{R}^d$ and a query vector $\hat{\boldsymbol{x}} \in \mathbb{R}^d$ from the input data distribution $\mathcal{P}_X$, then these data vectors are organized to form a matrix $\mathbf{X} = \begin{bmatrix} \boldsymbol{x}^{(1)} & \boldsymbol{x}^{(2)} & \cdots & \boldsymbol{x}^{(\psi_s)} & \hat{\boldsymbol{x}} \end{bmatrix} \in \mathbb{R}^{d \times (\psi_s + 1)}$; (iii) we apply the feature extractor $\phi$ to each column $\boldsymbol{x}^{(i)}$ of $\mathbf{X}$ to obtain the sparse feature $\boldsymbol{\phi}(s, \boldsymbol{x}^{(i)}) \in \mathbb{R}^{\mathcal{N}_s}$ and generate the target $y^{(i)} := y(s, \boldsymbol{x}^{(i)})$ and $\hat{y} := y(s, \hat{\boldsymbol{x}})$ according to Eq. (3), then one in-context data point of the task $s$ can now be generated as

$$\Phi(s, \mathbf{X}) := \begin{bmatrix} \boldsymbol{\phi}^{(1)} & \cdots & \boldsymbol{\phi}^{(\psi_s)} & \hat{\boldsymbol{\phi}} \\ y^{(1)} & \cdots & y^{(\psi_s)} & 0 \end{bmatrix} = \begin{bmatrix} \boldsymbol{\phi}(s, \boldsymbol{x}^{(1)}) & \cdots & \boldsymbol{\phi}(s, \boldsymbol{x}^{(\psi_s)}) & \boldsymbol{\phi}(s, \hat{\boldsymbol{x}}) \\ y(s, \boldsymbol{x}^{(1)}) & \cdots & y(s, \boldsymbol{x}^{(\psi_s)}) & 0 \end{bmatrix}; \tag{4}$$

(iv) repeating the above procedure for $N$ times can give us an in-context dataset with $N$ data points, where the numbers of data points for different tasks obey the power law Eq. (1). Finally, given in-context data $\Phi(s, \mathbf{X}) \in \mathbb{R}^{(\mathcal{N}_s + 1) \times (\psi_s + 1)}$ (Eq. (4)) and loss function $L$, in-context learning aims to

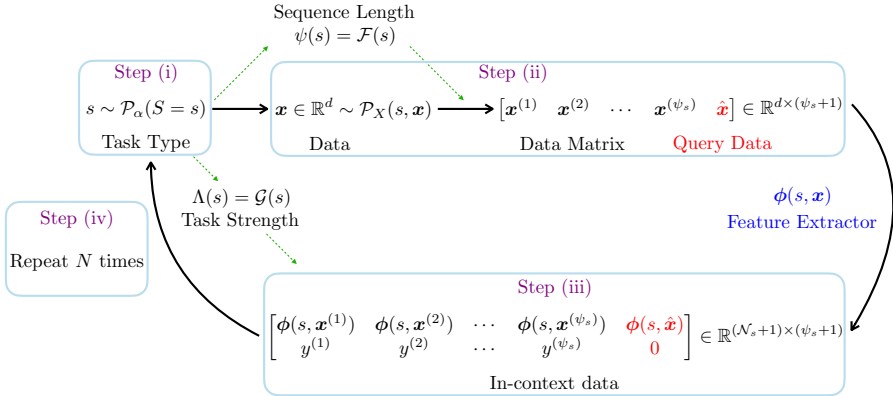

Figure 1: In-context data generation of multitask sparse feature regression (MSFR).

learn a model $f : \mathbb{R}^{(\mathcal{N}_s+1)\times(\psi_s+1)} \to \mathbb{R}$ such that $\boldsymbol{\theta}^* = \arg\min_{\boldsymbol{\theta}} L(f(\Phi;\boldsymbol{\theta}),\hat{y})$. Note that MSFR can be seen as a limiting case of in-context regression under source/capacity condition (generalization of setup in Lu et al. (2024) with that of Cui et al. (2022), see Appendix H.1).

## 2.3 LINEAR SELF-ATTENTION BLOCK

Self-attention block stands at the core of the transformer architectures (Vaswani et al., 2017). A single-head self-attention block (without residual connection) $f : \mathbb{R}^{d \times d_L} \mapsto \mathbb{R}^{d \times d_L}$ parameterized by $\boldsymbol{\theta}$ updates an input $\boldsymbol{G} \in \mathbb{R}^{d \times d_L}$ to

$$\hat{\boldsymbol{G}} := f(\boldsymbol{G};\boldsymbol{\theta}) = \boldsymbol{P}\boldsymbol{V}\boldsymbol{G}\,\mathrm{softmax}\left[(\boldsymbol{W}_K\boldsymbol{G})^T(\boldsymbol{W}_Q\boldsymbol{G})\right] \in \mathbb{R}^{d \times d_L}$$

where $\boldsymbol{\theta} = \{\boldsymbol{P},\boldsymbol{V},\boldsymbol{W}_K,\boldsymbol{W}_Q\}$, $\boldsymbol{P} \in \mathbb{R}^{d \times d_v}$ is the projection matrix, $\boldsymbol{V} \in \mathbb{R}^{d_v \times d}$ is the value matrix, and $\boldsymbol{W}_K, \boldsymbol{W}_Q \in \mathbb{R}^{d_e \times d}$ are the key matrix and query matrix, respectively. Note that $\mathrm{softmax}$ is applied column-wise.

In this paper, we study a simplified version of the self-attention by removing the $\mathrm{softmax}$ operation and merge the key matrix and query matrix as a single matrix $\boldsymbol{W}_{KQ} := \boldsymbol{W}_K^T\boldsymbol{W}_Q \in \mathbb{R}^{d \times d}$, which has been a popular choice in recent works, e.g., Zhang et al. (2023); von Oswald et al. (2023), due to its analytical tractability as well as the ability of capturing properties of the standard self-attention.

In particular, given the MSFR problem (Section 2.1) and in-context data $\Phi(s, \mathbf{X})$ (Eq. (4)), we study the in-context learning of linear self-attention block $\boldsymbol{V}\Phi\Phi^T\boldsymbol{W}_{KQ}\Phi \in \mathbb{R}^{\mathcal{N}_s \times (\psi_s+1)}$ where $\boldsymbol{W}_{KQ} \in \mathbb{R}^{(\mathcal{N}_s+1)\times(\mathcal{N}_s+1)}$. To obtain the scalar prediction of the query data $\hat{\phi}$, we adopt the output of the self-attention as $f(\Phi(s,\mathbf{X});\boldsymbol{\theta}) = [\boldsymbol{V}\Phi\Phi^T\boldsymbol{W}_{KQ}\Phi]_{s,\psi_s+1}$. Despite that this formulation makes a slight change to self-attention, it provides us the convenience to investigate its intricate in-context learning dynamics. Furthermore, decomposing $\boldsymbol{W}_{KQ}$ as $(\boldsymbol{W} \quad \boldsymbol{w}_{-1})$ where

$$\boldsymbol{W} = (\boldsymbol{w}_1 \quad \cdots \quad \boldsymbol{w}_{\mathcal{N}_s}) \in \mathbb{R}^{(\mathcal{N}_s+1)\times\mathcal{N}_s}, \quad \forall i \in \{-1,1,\ldots,\mathcal{N}_s\} : \boldsymbol{w}_i \in \mathbb{R}^{\mathcal{N}_s+1}, \quad (5)$$

we can write the output of the linear self-attention block for the task $s$ as $f(\Phi(s,\mathbf{X});\boldsymbol{\theta}) = [\boldsymbol{V}\Phi\Phi^T\boldsymbol{W}\hat{\phi}]_s$ which will be used in the rest part of this paper.

## 3 A TRACTABLE SOLUTION OF LINEAR SELF-ATTENTION

Section 2 establishes our in-context learning framework of multitask sparse feature regression problem for linear self-attention. In this section, we will closely investigate the corresponding learning dynamics by modelling it as non-linear ODE systems in Section 3.1 and give a tractable solution of it in Section 3.2.

## 3.1 IN-CONTEXT LEARNING DYNAMICS

Given the in-context dataset generated according to the procedure described in Section 2.2 with $N$ data points $\{\Phi(s^{(n)}, \mathbf{X}^{(n)})\}_{n=1}^{N}$, we use the mean-squared error (MSE) loss such that

$$\tilde{L}(\boldsymbol{\theta}) = \frac{1}{2N} \sum_{n=1}^{N} \left( f\left( \Phi(s^{(n)}, \mathbf{X}^{(n)}); \boldsymbol{\theta} \right) - \hat{y}^{(n)} \right)^2 \tag{6}$$

where $\tilde{L}$ is the empirical loss. The goal of in-context learning now becomes the traditional empirical loss minimization $\boldsymbol{\theta}^* = \arg\min_{\boldsymbol{\theta}} \tilde{L}(\boldsymbol{\theta})$, which can be solved by various optimization algorithms, and we focus on the general gradient descent in the continuous time limit, i.e., gradient flow (GF): $\dot{\boldsymbol{V}} = -\nabla_{\boldsymbol{V}} \tilde{L}(\boldsymbol{V}, \boldsymbol{W}), \quad \dot{\boldsymbol{W}} = -\nabla_{\boldsymbol{W}} \tilde{L}(\boldsymbol{V}, \boldsymbol{W})$.

To further investigate the learning dynamics, considering the formulation of the feature extractor Eq. (2) and in-context data Eq. (4) and denoting the standard basis vector in $\mathbb{R}^{\mathcal{N}_s}$ as $\boldsymbol{e}_s = (0 \;\; \cdots \;\; 0 \;\; 1 \;\; 0 \;\; \cdots)^T \in \mathbb{R}^{\mathcal{N}_s}$ for $s \in \{1, \ldots, \mathcal{N}_s\}$ such that the only nonzero component of $\boldsymbol{e}_s$ is its $s$-th component, we find that $\boldsymbol{H}_s \in \mathbb{R}^{(\mathcal{N}_s+1) \times (\mathcal{N}_s+1)}$ defined by (Appendix D.1)

$$\boldsymbol{H}_s := \Phi\left(s^{(n)}, \mathbf{X}^{(n)}\right) \Phi^T\left(s^{(n)}, \mathbf{X}^{(n)}\right) = \begin{bmatrix} \mathrm{diag}\left((\psi_s+1)\boldsymbol{e}_s\right) & \psi_s \Lambda_s \boldsymbol{e}_s \\ \psi_s \Lambda_s \boldsymbol{e}_s^T & \psi_s \Lambda_s^2 \end{bmatrix} \tag{7}$$

does not change for different $n$, where $\psi_s$ is the context sequence length and $\Lambda_s$ is the task strength for the task $s$, both of which only depend on the task type $s$. $\boldsymbol{H}_s$ is composed of the feature covariance and target (Eq. (24)). In addition, if we further decompose $V$ as $\boldsymbol{V}^T = (\boldsymbol{v}_1 \;\; \cdots \;\; \boldsymbol{v}_{\mathcal{N}_s})$, $\forall i \in \{1, \ldots, \mathcal{N}_s\} : \boldsymbol{v}_i \in \mathbb{R}^{\mathcal{N}_s+1}$, and recall the decomposition of $\boldsymbol{W}$ Eq. (5), then we can rewrite the olriginal empirical loss Eq. (6) as (Appendix D.1)

$$\textbf{Empirical loss function:} \quad \tilde{L} = \frac{1}{2} \sum_{s=1}^{\mathcal{N}_s} \frac{\#_s}{N} \left( \boldsymbol{v}_s^T \boldsymbol{H}_s \boldsymbol{w}_s - \Lambda_s \right)^2 \tag{8}$$

where $\#_s$ denotes the number of in-context data points for the task type $s$ in the dataset $\{\Phi(s^{(n)}, \mathbf{X}^{(n)})\}_{n=1}^{N}$, i.e., $\#_s = \sum_{n=1}^{N} \delta_{s,s^{(n)}}$. Eq. (8) indicates that the dynamics of $\boldsymbol{v}_s$ and $\boldsymbol{w}_s$ for different $s$ are decoupled: the $s$-th row of $\boldsymbol{V}$ and $s$-th column of $\boldsymbol{W}$ are responsible for learning and predicting the task strength of the task type $s$, rendering self-attention adapting itself to different tasks according to the in-context data. With this empirical loss function, we can now use a set of non-linear ODE systems $\forall s \in \{1, \ldots, \mathcal{N}_s\}$ :

$$\textbf{In-context learning dynamics:} \quad \dot{\boldsymbol{v}}_s = -\frac{\#_s}{N} \left( f_s - \Lambda_s \right) \boldsymbol{H}_s \boldsymbol{w}_s, \quad \dot{\boldsymbol{w}}_s = -\frac{\#_s}{N} \left( f_s - \Lambda_s \right) \boldsymbol{H}_s \boldsymbol{v}_s \tag{9}$$

to describe the in-context learning dynamics by GF where we denote $f_s = \boldsymbol{v}_s^T \boldsymbol{H}_s \boldsymbol{w}_s$ for simplicity. We note that $f_s$ is sufficient for us to investigate the dynamical behaviors of the output of self-attention for task type $s$ and the empirical loss. Thus, by abusing of definition, we refer to the solution of $f_s$ as the solution of the in-context learning dynamics, which can also be applied to give solutions of $\boldsymbol{v}_s$ and $\boldsymbol{w}_s$.

We highlight that the ODE systems above are non-linear for both $\boldsymbol{v}_s \in \mathbb{R}^{\mathcal{N}_s}$ and $\boldsymbol{w}_s \in \mathbb{R}^{\mathcal{N}_s}$, and, obviously, are different from the logistic differential equations obtained from the GF dynamics of deep linear networks (Saxe et al., 2014; Nam et al., 2024) and different from the Lotka-Volterra predator-prey model (Volterra, 1928). In this sense, the dynamics of linear self-attention (and transformer) is different from that of deep linear networks. Meanwhile, we note that non-linear ODE systems, including Eq. (9) which are non-linear ODE systems for vectors, typically do not admit closed-form solutions. Therefore we emphasize that solving Eq. (9), which might be of independent interest, to obtain the explicit dynamical behaviors of linear self-attention is novel as well as intriguing.

## 3.2 SOLUTION OF IN-CONTEXT LEARNING DYNAMICS

Although it is intractable to give the exact closed-form solution to the in-context learning dynamics Eq. (9), in this section, we will provide a solution that can be approximately exact under the following condition. We defer technical details of this section to Appendix E.

**Assumption 3.1.** $\forall s \in \{1, \ldots, \mathcal{N}_s\}$, the context sequence length $\psi_s \gg 1$.

**Procedure sketch.** Before diving into a detailed procedure for deriving the solution, we first present a rough sketch for it. The first step is to transform ODE systems Eq. (9) to a more symmetrical form Eq. (10) by changing of variables. Then we decompose Eq. (10) as two sets of ODE systems by comparing both sides of Eq. (10) to the zero-th and first orders of $\epsilon_s := 1/\psi_s$ since $\boldsymbol{H}_s$ can be decomposed as two parts, $\boldsymbol{H}_s^0$ and $\epsilon_s \boldsymbol{H}_s^1$ with $\epsilon_s \ll 1$. We then apply a change of variable again and derive a new set of ODEs Eq. (11) as Riccati equations Eq. (12) which admit closed-form solutions by noticing the existence of an important conserved quantity of the dynamics.

We now discuss the procedure in detail. Our first crucial observation is that the ODE of $\boldsymbol{v}_s$ Eq. (9) is non-linear with respect to $\boldsymbol{w}_s$, which makes it hard to solve. Therefore, we first convert Eq. (9) into a more symmetrical form by changing of variables: let $\boldsymbol{\eta}_s = \boldsymbol{v}_s + \boldsymbol{w}_s$ and $\boldsymbol{\rho}_s = \boldsymbol{v}_s - \boldsymbol{w}_s$, then the dynamics of $\boldsymbol{\eta}_s \in \mathbb{R}^{\mathcal{N}_s+1}$ and $\boldsymbol{\rho}_s \in \mathbb{R}^{\mathcal{N}_s+1}$ can be obtained according to Eq. (9)

$$\dot{\boldsymbol{\eta}}_s = -\frac{\#_s}{N}\left(\frac{g_s - h_s}{4} - \Lambda_s\right)\boldsymbol{H}_s\boldsymbol{\eta}_s, \ \dot{\boldsymbol{\rho}}_s = \frac{\#_s}{N}\left(\frac{g_s - h_s}{4} - \Lambda_s\right)\boldsymbol{H}_s\boldsymbol{\rho}_s, \quad (10)$$

where we define $g_s = \boldsymbol{\eta}_s^T \boldsymbol{H}_s \boldsymbol{\eta}_s$ and $h_s = \boldsymbol{\rho}_s^T \boldsymbol{H}_s \boldsymbol{\rho}_s$.

In this way, by solving Eq. (10), we can find the solution of the self-attention and the empirical loss function Eq. (8). However, Eq. (10) is still not directly solvable. Fortunately, recalling the definition of $\boldsymbol{H}_s$ in Eq. (7), we can rewrite $\boldsymbol{H}_s$ as a sum of two matrices $\boldsymbol{H}_s = \psi_s\left(\boldsymbol{H}_s^0 + \epsilon_s \boldsymbol{H}_s^1\right)$ where

$$\boldsymbol{H}_s^0 = \begin{bmatrix} \text{diag}(\boldsymbol{e}_s) & \Lambda_s \boldsymbol{e}_s \\ \Lambda_s \boldsymbol{e}_s^T & \Lambda_s^2 \end{bmatrix}, \quad \boldsymbol{H}_s^1 = \begin{bmatrix} \text{diag}(\boldsymbol{e}_s) & 0 \\ 0 & 0 \end{bmatrix}$$

and $\epsilon_s = 1/\psi_s \ll 1$ according to Assumption 3.1, which allows us to treat $\epsilon \boldsymbol{H}_s^1$ as an insignificant perturbation in the dynamics Eq. (10) and solve it using the perturbation analysis.

Specifically, suppose that the solutions of Eq. (10) can be written as $\boldsymbol{\eta}_s = \boldsymbol{\eta}_s^0 + \epsilon_s \boldsymbol{\eta}_s^1$ and $\boldsymbol{\rho}_s = \boldsymbol{\rho}_s^0 + \epsilon_s \boldsymbol{\rho}_s^1$ such that $\boldsymbol{\eta}_s^1$ and $\boldsymbol{\rho}_s^1$ are treated as perturbations to $\boldsymbol{\eta}_s^0$ and $\boldsymbol{\rho}_s^0$ respectively, then $g_s$ and $h_s$ can also be written in a perturbed form $g_s = g_s^0 + \epsilon_s g_s^1$ and $h_s = h_s^0 + \epsilon_s h_s^1$ accordingly (Appendix E.1). Now we can obtain ODEs for $\boldsymbol{\eta}_s^0, \boldsymbol{\eta}_s^1, \boldsymbol{\rho}_s^0$, and $\boldsymbol{\rho}_s^1$ by comparing terms to the zero-th and first orders of $\epsilon_s$ in both sides of Eq. (10), respectively (Appendix E.1). This will finally give us ODEs for $g_s^0, h_s^0, g_s^1$, and $h_s^1$, the final ODEs that we aim to solve since the output of self-attention for the task $s$ can be written as $f_s := f_s^0(t) + \epsilon_s f_s^1(t) = [(g_s^0 - h_s^0) + \epsilon_s(g_s^1 - h_s^1)]/4$.

Our strategy for finding solutions of $f_s$ is now composed of two parts using the perturbation analysis: (i) solve the ODEs for $g_s^0$ and $h_s^0$ exactly and (ii) find $\boldsymbol{\eta}_s^0$ and $\boldsymbol{\rho}_s^0$ according to the solved $g_s^0$ and $h_s^0$, then put them into ODEs of $g_s^1$ and $h_s^1$ to find their solutions. As mentioned earlier, $f_s^1(t)$ is far less significant than $f_s^0(t)$ to the dynamical behaviors of self-attention given Assumption 3.1, thus we defer the discussion of $f_s^1(t)$ to Appendix J and only focus on $f_s^0(t)$.

We now discuss the first step for $f_s^0(t)$. Our key observation is that $\boldsymbol{H}_s^0$ is like an idempotent matrix: $(\boldsymbol{H}_s^0)^2 = (\Lambda_s^2 + 1)\boldsymbol{H}_s^0$, which gives us the dynamics of $g_s$ and $h_s$ as a new set of non-linear ODEs:

$$\dot{g}_s^0 = -\left(g_s^0 - h_s^0 - 4\Lambda_s\right)a_s g_s^0/2, \ \dot{h}_s^0 = \left(g_s^0 - h_s^0 - 4\Lambda_s\right)a_s h_s^0/2, \quad (11)$$

where we let $a_s = \#_s \psi_s(\Lambda_s^2 + 1)/N$ for ease of notation. Though Eq. (11) is still a non-linear ODE system, it is much more tractable than the original in-context learning dynamics Eq. (9). Our following key observation can drastically simplify Eq. (11) even further: $\forall t \geq 0 : g_s^0 h_s^0 = 2C_s$ where $C_s$ is a constant determined by the initialization, i.e., $g_s^0 h_s^0$ is conserved for the dynamics, since $d(g_s^0 h_s^0)/dt = 0$. In this way, Eq. (11) becomes the following set of Riccati equations that can be solved (Appendix E.2) to give our main results:

$$\dot{g}_s^0 = 2a_s\Lambda_s g_s^0 - a_s(g_s^0)^2/2 + a_s C_s, \ \dot{h}_s^0 = -2a_s\Lambda_s h_s^0 - a_s(h_s^0)^2/2 + a_s C_s. \quad (12)$$

**Theorem 3.1** (Solution for in-context learning dynamics of linear self-attention: zero-th order). *For MSFR problem and the in-context learning dynamics by GF of the linear self-attention block Eq. (9), the solution $f_s(t)$ can be approximately written as an expansion $f_s^0(t) + \epsilon_s f_s^1(t)$ at large $\psi_s$ with*

$$f_s^0(t) = \Lambda_s + \frac{\lambda_s}{2}\left[\frac{1}{1 + P_s \exp(a_s\lambda_s t)} - \frac{1}{1 + Q_s \exp(a_s\lambda_s t)}\right] \quad (13)$$

*where $a_s = \#_s \psi_s(\Lambda_s^2 + 1)/N$, and*

$$\lambda_s = \sqrt{4\Lambda_s^2 + 2C_s}, \ P_s = \frac{4f_s^0(0)\Lambda_s + 2C_s + \lambda_s\sqrt{4(f_s^0(0))^2 + 2C_s}}{2(f_s^0(0) - \Lambda_s)(\lambda_s - 2\Lambda_s)}, \ Q_s = P_s\frac{2\Lambda_s - \lambda_s}{2\Lambda_s + \lambda_s}$$

*are determined by the initialization. In addition, when $\boldsymbol{v}_s^0(0) = \pm\boldsymbol{w}_s^0(0)$, the constant $C_s = 0$ and, denoting $\Delta_s = (\Lambda_s - f_s^0(0))/f_s^0(0)$, the solution can be simplified to be a standard logistic function*

$$f_s^0(t) = \frac{\Lambda_s}{1 + \Delta_s e^{-2a_s \Lambda_s t}} \quad \text{when } \boldsymbol{v}_s^0(0) = \pm\boldsymbol{w}_s^0(0). \tag{14}$$

**Remark.** The closed-form solution $f_s^0(t)$ obtained in Theorem 3.1 is to the zeroth order of $\epsilon_s$ and is a good approximation under Assumption 3.1. $a_s, P_s, Q_s$, and $\lambda_s$ jointly control the learning process, where $P_s, Q_s$, and $\lambda_s$ are determined by the initialization and task strength, e.g., when $f_s^0(0) = \Lambda_s$ both $P_s, Q_s \to \infty$ thus $f_s^0(t) = \Lambda_s$ for $t \geq 0$. $a_s$ is determined by the dataset, e.g., sequence length, and task strength. In Fig. 2, we compare the loss that is calculated using $f_s^0(t)$ with that obtained from direct empirical simulation for different context sequence lengths. It can be seen that when the context sequence length $\psi \gg 1$ (blue lines), the theoretical

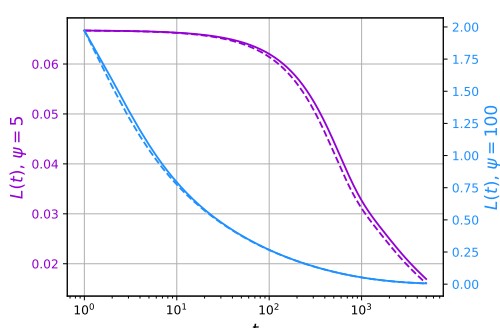

Figure 2: Loss $L(t)$ for different context sequence lengths $\psi$(5 and 100) during training. Solid lines are for theoretical predictions while dashed lines are for empirical simulations.

prediction of the test loss matches with the empirical results precisely, which validates the accuracy of $f_s^0(t)$. In addition, we emphasize that Theorem 3.1 is applicable for any $\mathcal{P}_s$ (Eq. (1)) of the task type, $\mathcal{F}(s)$ of the sequence length, and $\mathcal{G}(s)$ of the task strength for the MSFR problem in Section 2, i.e., the derivation of $f_s^0(t)$ does not require assumptions on forms of $\mathcal{P}_s$, $\mathcal{F}(s)$, and $\mathcal{G}(s)$.

$f_s^0(t)$ can explicitly characterize the dynamical behaviors of the self-attention block including the influence of various parameters, suggesting that it could contribute to the understanding of self-attention in a variety of aspects. In this paper, our focus will be neural scaling laws. For another example, $f_s^0(t)$ shows that self-attention learns different tasks in different rates that depend on various parameters such as the sequence length, task strength, number of data points, and the initialization. The difference of learning speeds for different tasks might lead to the grokking phenomenon (Power et al., 2022), since the model can quickly fit a fraction of tasks while learns the rest extremely slowly.

## 4 NEURAL SCALING LAWS FOR LINEAR SELF-ATTENTION

In this section, we examine neural scaling laws for linear self-attention with its special in-context learning dynamics. We note that this focus is different from those architectures considered in previous works (Bordelon et al., 2024; Nam et al., 2024; Hutter, 2021; Michaud et al., 2023; Maloney et al., 2022), providing us a possibility to compare self-attention with other architectures regarding neural scaling laws from a theoretical perspective. In particular, we will investigate the test loss according to the solution obtained in Theorem 3.1 under Assumption 3.1 and the relation $f_s(t) \approx f_s^0(t)$, which enables us to write the test loss as

$$\textbf{Test Loss:} \quad L(t) = \mathbb{E}_{\boldsymbol{x} \sim \mathcal{P}_X, s \sim \mathcal{P}_\alpha}\left[\ell\left(f(\Phi(s, \mathbf{X}); \boldsymbol{\theta}), y\right)\right] \approx \frac{1}{2}\sum_{s=1}^{\mathcal{N}_s}\mathcal{P}_\alpha(S = s)\left[f_s^0(t) - \Lambda_s\right]^2. \tag{15}$$

Concretely, we consider neural scaling laws of linear self-attention with respect to each one of size of the model $D$, training time $t$, and the number of training data $N$ when the other two factors are not the bottleneck of training. Finally, we consider scaling laws for the optimal compute $\mathcal{C}$. We list the discussion of these factors as follows and defer details to Appendix F.

- **Size of the model** $D$. To quantify the model size $D$, inspired by Michaud et al. (2023); Bordelon & Pehlevan (2022); Nam et al. (2024) which considered different models empirically or theoretically, we assume that there is a cutoff $D$ such that the model cannot learn any task strength $\Lambda_s$ with $s \geq D$. Specifically, we let $\boldsymbol{W}_{KQ} \in \mathbb{R}^{D \times D}, \boldsymbol{V} \in \mathbb{R}^{(D-1) \times D}$ and apply a new feature extractor such that $\boldsymbol{\phi}'(s, \boldsymbol{x}) \in \mathbb{R}^{D-1}$ only extracts the first $D - 1$

elements of the original $\phi(s, \boldsymbol{x})$. When $D$ is the bottleneck of training, we let $t, N \to \infty$ (they are sufficient for the training) to derive scaling laws with respect to $D$.

- **Training time** $t$. Training time $t$ is equivalent to the number of optimization steps. To investigate the scaling law with respect to $t$, we remove the bottleneck caused by the size of model and the number of data points by letting $N \to \infty$ and $D = \mathcal{N}_s$.

- **Number of training data** $N$. When the training is bottlenecked by $N$, we let $t \to \infty$ and the cutoff $D = \mathcal{N}_s$ following the above arguments.

- **Optimal compute** $\mathcal{C}$. This is the case when the number of data points is sufficient for the training, while training time $t$ or the size of model $D$ is the bottleneck given the compute budget $\mathcal{C} = Dt$ such that either $t$ or $D$ scales differently with $\mathcal{C}$. Specifically, if $L(t, D) = a_t t^{-\alpha_t} + a_D D^{-\alpha_D}$, then we can derive the optimal test loss as $L \propto \mathcal{C}^{-\alpha_t \alpha_D / (\alpha_t + \alpha_D)}$ given $\mathcal{C} = tD$ (Appendix F.1).

To inspect the effects of the context sequence length $\psi_s \in \mathbb{R}$ and task strength $\Lambda_s \in \mathbb{R}$ on neural scaling laws, we let $\psi_s = \mathcal{F}(s) \propto s^{-\beta}$, which is inspired by the underlying power-law correlations in language sequence (Ebeling & Pöschel, 1994), and $\Lambda_s = \mathcal{G}(s) \propto s^{-\gamma}$ with $\beta, \gamma > 0$. We will investigate two different cases. In the first case (Section 4.1), we assume that $\gamma = \beta = 0$ such that the context sequence length $\psi_s$ and task strength $\Lambda_s$ do not depend on the task type $s$. In this setting, we can compare neural scaling laws of linear self-attention with other architectures. In the second case (Section 4.2), we study positive $\beta$ and $\gamma$, which are unique to self-attention. Details of Section 4.1 and 4.2 will be deferred to F.2 and F.3, respectively.

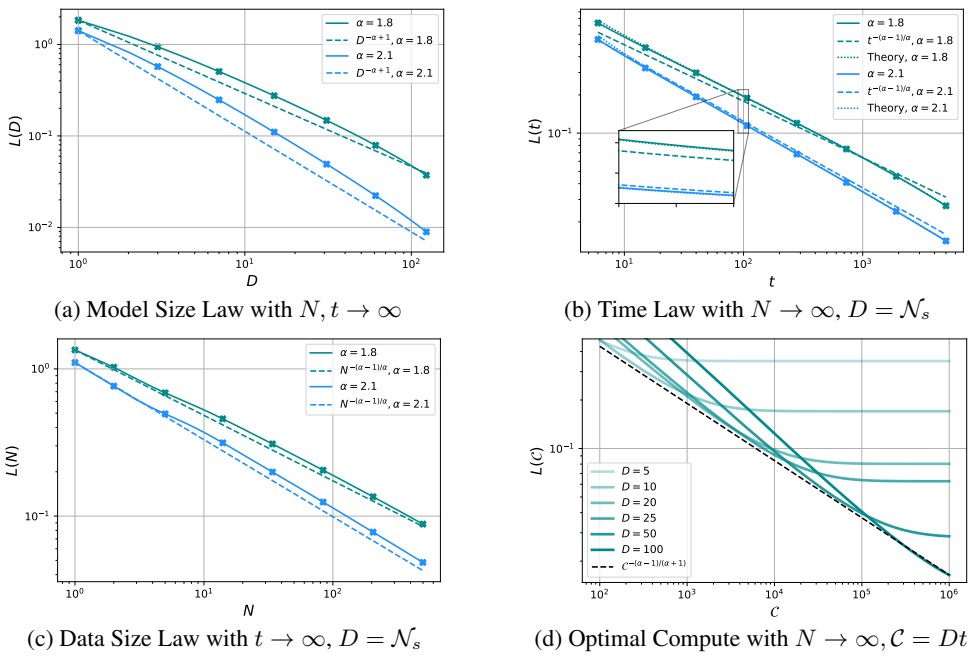

(a) Model Size Law with $N, t \to \infty$

(b) Time Law with $N \to \infty$, $D = \mathcal{N}_s$

(c) Data Size Law with $t \to \infty$, $D = \mathcal{N}_s$

(d) Optimal Compute with $N \to \infty$, $\mathcal{C} = Dt$

Figure 3: Neural scaling laws for linear self-attention with different values of $\alpha = 1.8, 2.1$. In each figure, we use solid lines to represent empirical simulation results and dashed lines for power law curves. We also plot theoretical predictions of test loss in (b) with dotted lines as a comparison. In (d), we set $\alpha = 1.8$ and use different levels of transparency to reflect different model sizes $D$ within the range $[\mathcal{N}_s/100, \mathcal{N}_s/5]$.

## 4.1 FIXED CONTEXT SEQUENCE LENGTH AND TASK STRENGTH

For simplicity we assume that the model is initialized as $C_s = C$ and $f_s^0(0) = f_0$, which implies that $\lambda_s = \lambda$, $P_s = P$, and $Q_s = Q$ for $s \in \{1, \dots, \mathcal{N}_s\}$ do not depend on the task type $s$. When the context sequence length $\psi_s$ and task strength $\Lambda_s$ are the same for different task types $s$, i.e.,

$\gamma = \beta = 0$, the test loss Eq. (15) can be written as

$$L(t) \approx \frac{Z\lambda^2}{8} \sum_{s=1}^{\mathcal{N}_s} s^{-\alpha} \left( \frac{1}{1 + Pe^{a_s\lambda t}} - \frac{1}{1 + Qe^{a_s\lambda t}} \right)^2 \tag{16}$$

according to Theorem 3.1. To derive neural scaling laws, we plan to find the asymptotic behaviors of $L(t)$ with respect to each one of $t$, $N$, and $D$ when the other two are not the bottleneck of training using Eq. (16). Then we can use the relation $\mathcal{C} = Dt$ to find the scaling law with respect to the optimal compute budget. We summarize our results in Table 1 and verify them in Fig. 3.

Table 1: Neural scaling laws for linear self-attention when $\beta = \gamma = 0$

|  | Scaling Law | Condition |
|---|---|---|
| Size of model $D$ | $D^{-\alpha+1}$ | $t \to \infty, N \to \infty$ |
| Time $t$ | $t^{-\frac{\alpha-1}{\alpha}}$ | $N \to \infty, D = \mathcal{N}_s$ |
| Number of data $N$ | $N^{-\frac{\alpha-1}{\alpha}}$ | $t \to \infty, D = \mathcal{N}_s$ |
| Compute budget $\mathcal{C}$ | $\mathcal{C}^{-\frac{\alpha-1}{\alpha+1}}$ | $N \to \infty, t \propto \mathcal{C}^{\frac{\alpha}{\alpha+1}}, D \propto \mathcal{C}^{\frac{1}{\alpha+1}}$ |

**Architecture does not matter for scaling laws when context sequence length is fixed.** The results summarized in Table 1 reveal that, when data admits a similar power-law structure, linear self-attention shares the same neural scaling laws with ReLU MLPs (Michaud et al., 2023) and diagonal linear networks (Nam et al., 2024) with respect to $t$, $N$ and $D$. Linear self-attention also exhibits a similar time scaling law as the linear models considered in Bordelon et al. (2024) and Hutter (2021). These similarities indicate that the architecture of model does not affect exponents of neural scaling laws significantly, which well aligns with the empirical conclusion reached by Kaplan et al. (2020) where they showed that transformers share similar exponents of neural scaling laws with other models when the power-law structures hold.

## 4.2 Varied Context Sequence Length and Task Strength

To further capture how the context sequence length $\psi_s$ and the task strength $\Lambda_s$ affect neural scaling laws for the in-context learning of self-attention, we let $\psi_s = \mathcal{F}(s) \propto s^{-\beta}$, $\Lambda_s = \mathcal{G}(s) \propto s^{-\gamma}$ as in Section 2.2. We can write the test loss as

$$L(t) \approx \frac{Z}{2} \sum_{s=1}^{\mathcal{N}_s} s^{-\alpha-2\gamma} \left[ \frac{\Delta \exp(-2a_s\Lambda_s t)}{1 + \Delta \exp(-2a_s\Lambda_s t)} \right]^2 . \tag{17}$$

Following a similar procedure as in Section 4.1, we derive neural scaling laws of linear self-attention for MSFR problem in Table 2, which gives us the following insights.

Table 2: Neural scaling laws for linear self-attention when both $\psi_s$ and $\Lambda_s$ depend on $s$

|  | Scaling Law | Condition |
|---|---|---|
| Size of model $D$ | $D^{-\alpha-2\gamma+1}$ | $t \to \infty, N \to \infty$ |
| Time $t$ | $t^{-\frac{\alpha+2\gamma-1}{\alpha+\beta+3\gamma}}$ | $N \to \infty, D = \mathcal{N}_s$ |
| Number of data $N$ | $N^{-\frac{\alpha+2\gamma-1}{\alpha}}$ | $t \to \infty, D = \mathcal{N}_s$ |
| Compute budget $\mathcal{C}$ | $\mathcal{C}^{-\frac{\alpha+2\gamma-1}{\alpha+3\gamma+\beta+1}}$ | $N \to \infty, t \propto \mathcal{C}^{\frac{1}{\alpha+3\gamma+\beta+1}}, D \propto \mathcal{C}^{\frac{\alpha+3\gamma+\beta}{\alpha+3\gamma+\beta+1}}$ |

**Varied context sequence length affects the scaling law of time.** Table 2 reveals that a varied context sequence length makes the learning process slower (Fig. 4a). According to Table 2, a nonzero positive $\beta$ leads to a larger exponent of time law, thus the test loss will decrease slower than the case when $\beta = 0$. This suggests that it is better to balance the context sequence length for different tasks to obtain a satisfied test loss given a limitation of optimization steps. This conclusion is special to self-attention compared to other architectures considered in previous theoretical works since they lack the place for the context sequence length. On the other hand, we also find that $\beta$ does not appear in scaling laws for the size of model and number of data points, indicating that self-attention can still admit similar scaling laws for $D$ and $N$ as other architectures when $\gamma = 0$.

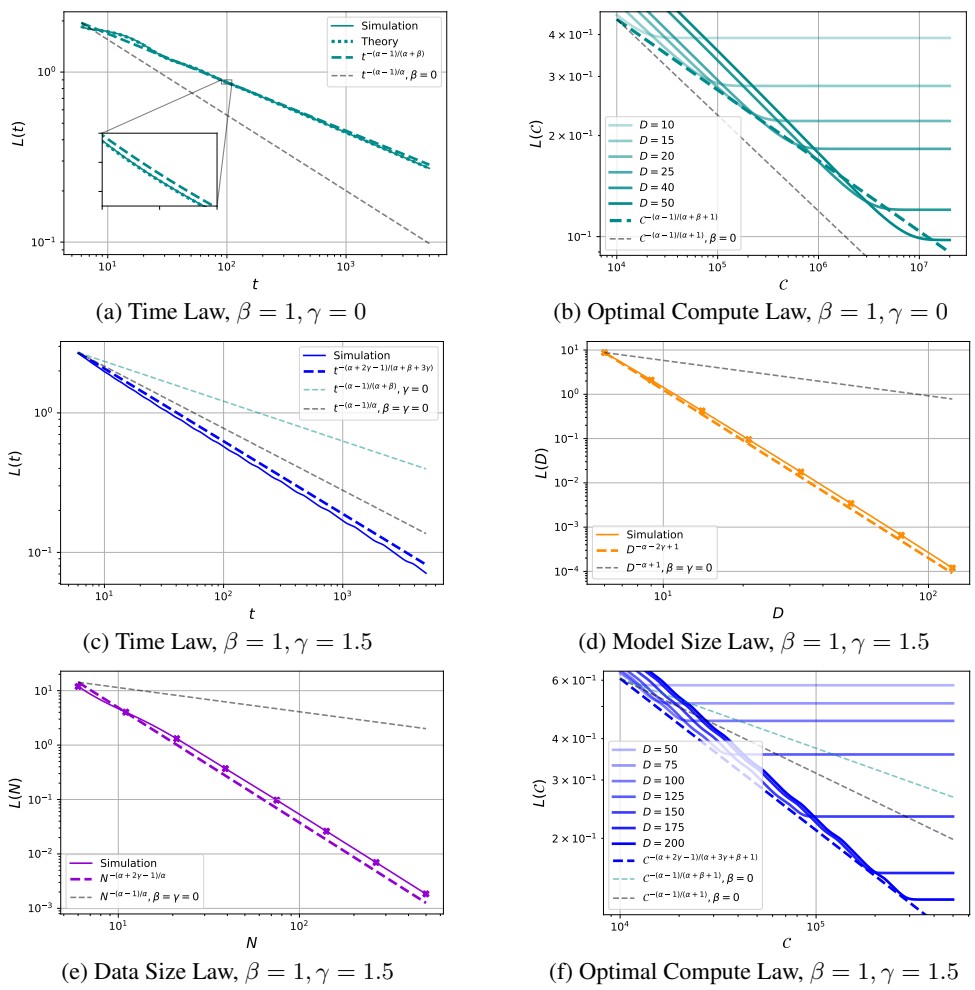

Figure 4: Neural scaling laws for self-attention with varied sequence length $\psi_s \propto s^{-\beta}$ and strength $\Lambda_s \propto s^{-\gamma}$. In each figure, we set $\alpha = 1.8$ and use solid lines to represent empirical simulation results while use dashed lines for power law curves. In addition, we plot the power law curves (black dashed lines) with $\gamma, \beta = 0$ from Table 1 as a comparison. In (a) and (b), we set $\gamma = 0$ to examine effects of $\psi_s$ on time and optimal compute laws only. In (b) and (f), we use different levels of transparency to reflect varying model sizes $D$.

**Varied task strength affects all scaling laws.** Table 2 reveals that a varied task strength reduces the requirements of the size of model (Fig. 4d) or the number of data points (Fig. 4e) in our MSFR problem. Specifically, due to the existence of a positive $\gamma$, exponents of scaling laws for both size of model and number of data points become smaller, thus the learning requires fewer number of data points or smaller size of model to achieve a similar test loss when they are the bottleneck.

## 5 CONCLUSION

In this paper, we target on understanding learning dynamics of self-attention, which stands at the core of the transformer architectures, from a theoretical perspective. For this purpose, we first design a multitask sparse feature regression problem for the self-attention to learn in an in-context manner, whose learning dynamics is then modelled as non-linear ODE systems. We then give a tractable solution to the ODE systems, which should be of broad interest since non-linear ODE systems typically do not admit closed-form solutions. We also highlight that this solution can be employed as an interesting proxy for studying a variety of properties of self-attention and transformers. Finally, we use the proposed solution to investigate neural scaling laws of self-attention with respect to each one of training time, number of data points, and size of the model when the other two are not the bottleneck of the learning process, which in turn allows us to establish the neural scaling law with respect to the optimal compute budget.

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

APPENDIX

The structure of the Appendix is as follows.

- In Appendix A, we discuss the in-context learnability of the MSFR setup.
- In Appendix B, we introduce additional related works.
- In Appendix C, we present key definitions from the main paper for convenience.
- In Appendix D, we establish the in-context learning dynamics as non-linear ODE systems.
- In Appendix E, we solve the ODE systems to the zero-th order of $\epsilon$ under Assumption 3.1.
- In Appendix F, we focus on neural scaling laws.
- In Appendix G, we list details for our numerical experiments.
- In Appendix H, we further discuss the generality of the MSFR problem and our solution by connecting the MSFR problem with other types of tasks.
- In Appendix I, we present supplement numerical experiments to extend our claims on neural scaling laws for linear self-attention to softmax self-attention.
- In Appendix J, we construct the complete solution of self-attention for the MSFR problem up to the first order of $\epsilon$.

## A IN-CONTEXT LEARNABILITY OF LINEAR SELF-ATTENTION IN THE SPARSE FEATURE REGRESSION

We provide a simple and straightforward example to explain how the model can adapt itself for different input sequences. For simplicity, we only consider one task $s$, then the prediction for a given query token $\hat{\phi}(s, \hat{x})$ and training sequence $\Phi(s, X)$ of the self-attention at $t = \infty$. We further assume a common constraint for the linear self-attention as in Zhang et al. (2023); Wu et al. (2024); Lu et al. (2024): the first $\mathcal{N}_s$ components of $\boldsymbol{v}_s$ and the last component of $\boldsymbol{w}_s$, i.e., $w_{s;\mathcal{N}_s+1}$, are zero for any $t \geq 0$. These simplifications enable us to write the output for the task $s$ as

$$f_s = v_{s;\mathcal{N}_s+1}(\infty) w_{s;s}(\infty) \psi_s \Lambda_s.$$

As $f_s = \Lambda_s$ at $t = \infty$ because the model is perfectly trained, we must have the following relation

$$v_{s;\mathcal{N}_s+1}(\infty) w_{s;s}(\infty) = 1/\psi_s.$$

What will the prediction of the self-attention for the query token $\hat{\phi}(s, \hat{x})$ be given a different input sequence $\tilde{\Phi}(s, X)$? Below we use our example to illustrate that the perfectly trained linear self-attention will adapt itself for $\tilde{\Phi}(s, X)$. In particular, let the task strength for the input tokens in $\tilde{\Phi}(s, X)$ prior to the query token be $\tilde{\Lambda}_s \neq \Lambda_s$ (the task strength is unseen in the training data). Then the prediction of the self-attention for this input sequence $\tilde{\Phi}(s, X)$ becomes

$$
\begin{aligned}
f_s &= \boldsymbol{v}_s^T(\infty) \tilde{\Phi} \tilde{\Phi}^T \boldsymbol{w}_s(\infty) \\
&= v_{s;\mathcal{N}_s+1}(\infty) w_{s;s}(\infty) \psi_s \tilde{\Lambda}_s = \tilde{\Lambda}_s,
\end{aligned}
\tag{18}
$$

which indicates that linear self-attention successfully predicts the unseen task strength $\tilde{\Lambda}_s$ by performing in-context learning to explore in-context data to adapt itself to a new task.

In addition, we can show that the above newly imposed constraint does not affect the learning dynamics, hence our method for solving the dynamics in Section 3.2 can still be applied here even without Assumption 3.1, i.e., the solution obtained here will be an exact one. To see this point more clearly, we rewrite the output of the model with the constraints as

$$f_s := \boldsymbol{v}_s \boldsymbol{H}_s \boldsymbol{w}_s = v_{s;\mathcal{N}_s+1} \psi_s \Lambda_s w_{s;s},$$

where $\boldsymbol{v}_s = v_{s;\mathcal{N}_s+1}$ and $\boldsymbol{w}_s = w_{s;s}$ are 1-dimensional vectors while $\boldsymbol{H}_s = \psi_s \Lambda_s$ is a $1 \times 1$ matrix that satisfies the idempotent-like condition. Thus the dynamics will be exactly the same as Eq. (9), and Theorem 3.1 can still be applied.

Finally, we briefly explain the reason why we impose these constraints. For the task $s$ and the input sequence $\Phi(s, X)$ with task strength $\Lambda_s$, the output of the model is

$$\boldsymbol{v}_s^T \Phi\Phi^T \boldsymbol{w}_s = \psi_s(v_{s;s}w_{s;s} + \Lambda_s(v_{s;\mathcal{N}_s+1}w_{s;s} + w_{s;\mathcal{N}_s+1}v_{s;s}) + \Lambda_s^2 v_{s;\mathcal{N}_s+1}w_{s;\mathcal{N}_s+1}).$$

We note that the first term does not depend on $\Lambda_s$ explicitly, meaning that it does not explore the relationship between $\phi(s, x)$ and $y$, thus it does not contribute to the in-context learning task. In addition, the last term depends on $\Lambda_s$ in a non-linear way, leading to an inappropriate dependence on the relationship between $x$ and $y$. Due to such terms, without any conditions, the linear self-attention is hard to perfectly adapt itself for new input sequences $\tilde{\Phi}(s, X)$ with different task strengths. This problem is caused by its structure. A possible way to achieve a perfect in-context adaption is imposing the constraints $v_{s;s} = w_{s;\mathcal{N}_s+1} = 0$ (the constraints used in our example and Zhang et al. (2023); Wu et al. (2024); Lu et al. (2024)) to eliminate the aforementioned terms. Therefore, we expect that an optimal in-context learner should set the first $\mathcal{N}_s$ components as $0$, since they contribute to many terms that are not necessary for the model to achieve perfect in-context adaption.

## B  ADDITIONAL RELATED WORKS

Due to the great success of LLMs, the understanding for the transformer architecture becomes increasingly important. As Garg et al. (2023) proposed the in-context learning for learning particular functions, besides Zhang et al. (2023), von Oswald et al. (2023); Akyürek et al. (2023) studied transformers for the linear regression problem, while Li et al. (2023a) revealed the similarity between a single-layer self-attention and gradient descent on a regression problem with softmax. Edelman et al. (2022) showed that the self-attention is able to learn sparse functions of the input sequence. For the loss landscape of single-layer transformer without $\mathrm{softmax}$, Ahn et al. (2023a) showed the existence of solution of the model parameters that can achieve perfect test loss. Furthermore, the idea of MSFR is also partly inspired by the multitask parity problem proposed by Michaud et al. (2023); Barak et al. (2022) that empirically exhibits neural scaling laws.

For other neural networks, the study of the learning dynamics is always an important topic. There is a line of research that investigated properties of the converged parameters in both classification and regression settings, e.g., Soudry et al. (2024). To characterize the whole training trajectory, Saxe et al. (2014) built an exact solution of deep linear networks that depends on time explicitly. In addition, various tools borrowed from random matrix theory and statistical physics have also been applied to investigate the learning dynamics of linear models (Spigler et al., 2020; Bordelon & Pehlevan, 2022; Simon et al., 2023; Bahri et al., 2024). There are also a fruitful results (Adlam & Pennington, 2020; d'Ascoli et al., 2020; Geiger et al., 2020) that analyzed the learning dynamics of neural networks in the limiting case with the neural tangent kernel (NTK) (Jacot et al., 2020).

### B.1  ADDITION RELATED WORKS ON LEARNING DYNAMICS OF DEEP NEURAL NETWORKS

Besides the aforementioned works on learning dynamics of neural networks or random feature models, Pinson et al. (2023) studied the learning dynamics of gradient descent for linear convolution neural networks. Particularly, they discovered an interesting interplay between the data structure and network structure that determines the phases of the network along the training trajectory. The learning dynamics is also analyzed by Gidel et al. (2019) while Braun et al. (2022); Atanasov et al. (2021) focused on different initialization regimes. Our focus in this paper is the learning dynamics of linear self-attention with the in-context learning.

### B.2  ADDITIONAL RELATED WORKS ON NEURAL SCALING LAWS

Besides Kaplan et al. (2020); Hoffmann et al. (2022), there are a number of recent works that explored scaling laws in deep neural networks empirically (Rosenfeld et al., 2021; Hestness et al., 2017; Rosenfeld et al., 2019). The study of neural scaling laws can be found in some earlier works (Caponnetto & De Vito, 2007; Steinwart et al., 2009; Ahmad & Tesauro, 1988). From the theoretical perspective, various works developed solvable models in the context of random feature models (Bahri et al., 2024; Atanasov et al., 2022; 2024; Bordelon et al., 2024; Paquette et al., 2024) to study the neural scaling laws in certain limits. In addition, Wei et al. (2022); Bordelon et al. (2021); Sharma & Kaplan (2022); Bordelon & Pehlevan (2022) also conducted theoretical analysis

on linear models to investigate the neural scaling laws. These works improve the theoretical understanding for the neural scaling law to a large extent. As a comparison, our focus in this paper is particularly on the linear self-attention with the in-context learning, which is not widely discussed in previous works.

### B.3    Additional Related Works on Analysis of in-context learning regression

The success of the transformer architecture (Vaswani et al., 2017) has encouraged a majority body of works to investigate its theoretical understanding, especially the intriguing in-context learning mechanism. A common setup along this direction is the study of linear regression using the linear self-attention (Duraisamy, 2024; Ahn et al., 2023b; Wu et al., 2024; Lu et al., 2024; Zhang et al., 2023). Our work also falls into this category as mentioned in Section 2.3. We present a more extensive comparison and connection to Zhang et al. (2023) and Lu et al. (2024) in the following.

**Comparison and connection to Zhang et al. (2023).**    Zhang et al. (2023) considered linear regression using linear self-attention in the in-context learning manner. By assuming the infinite training dataset size limit, Zhang et al. (2023) revealed that the converged linear self-attention can achieve a competitive performance compared to the best linear predictor over the test data distribution. As a comparison, our setting is also a regression for the linear self-attention to learn in an in-context manner, while our problem is a multitask version and the distribution of the task type obeys a power law. In addition, besides the converged solution of the dynamics, we derive its (approximate) form along the whole training trajectory, which in turn makes it possible to characterize the neural scaling laws with respect to time, data size, model size, and the optimal compute. And we note that the characterization for the time scaling law (and the optimal compute law) cannot be derived solely by the converged solution.

**Comparison and connection to Lu et al. (2024).**    Lu et al. (2024) proposed a solvable model of in-context learning for a linear regression task by linear self-attention. Specifically, assuming a limit where the input dimension, the context sequence length, the training task diversity, and the data size are all taken to infinity following certain ratios, they revealed a double-descent learning curve with respect to the number of examples. As a comparison, our problem is also a multitask regression for the linear self-attention to learn in an in-context way. In addition, as we will explain in Appendix H, our problem can be seen as a limiting case of the multitask in-context regression under the source-capacity condition, which is also a generalized version of the setup considered in Lu et al. (2024). Furthermore, our solution (to the first order of $\epsilon_s$ when the context length is large) captures the whole training trajectory and we study the neural scaling laws with respect to various parameters so that we do not assume that all parameters are taken to infinity together.

## C    Notations and Definitions

We present useful definitions in the main paper here for convenience.

- **Feature Extractor Eq. (2)**

$$\phi(s, \boldsymbol{x}) : \mathbb{R}^d \times \mathbb{R} \mapsto \{-1, 0, 1\}^{\mathcal{N}_s} \in \mathbb{R}^{\mathcal{N}_s},$$

- **Target Eq. (3)**

$$y(s, \boldsymbol{x}) = \Lambda_s \sum_{k=1}^{\mathcal{N}_s} \phi_k(s, \boldsymbol{x}).$$

- **An in-context data point Eq. (4)**

$$\Phi(s, \mathbf{X}) := \begin{bmatrix} \boldsymbol{\phi}_s^{(1)} & \cdots & \boldsymbol{\phi}_s^{(\psi_s)} & \hat{\boldsymbol{\phi}} \\ y_s^{(1)} & \cdots & y_s^{(\psi_s)} & 0 \end{bmatrix} = \begin{bmatrix} \boldsymbol{\phi}(s, \boldsymbol{x}^{(1)}) & \cdots & \boldsymbol{\phi}(s, \boldsymbol{x}^{(\psi_s)}) & \boldsymbol{\phi}(s, \hat{\boldsymbol{x}}) \\ y(s, \boldsymbol{x}^{(1)}) & \cdots & y(s, \boldsymbol{x}^{(\psi_s)}) & 0 \end{bmatrix}.$$

- **Empirical loss function**

$$\tilde{L} = \frac{1}{2N} \sum_{n=1}^{N} \left[ f\left(\Phi(s^{(n)}, \mathbf{X}^{(n)}); \boldsymbol{\theta}\right) - \hat{y}^{(n)} \right]^2$$

| Notation | Definition | Index |
|----------|-----------|-------|
| $\mathcal{P}_\alpha$ | distribution of task type | Eq. (1) |
| $\boldsymbol{x}$ | data vector | Page 3 |
| $\boldsymbol{\phi}(s, \boldsymbol{x})$ | feature extractor | Eq. (2) |
| $y(s, \boldsymbol{x})$ | target | Eq. (3) |
| $\psi_s$ | context sequence for task $s$ | Page 3 |
| $\Lambda_s$ | task strength for task $s$ | Page 3 |
| $\Phi(s, \mathbf{X})$ | an in-context data point | Eq. (4) |
| $\tilde{L}$ | empirical MSE loss | Eq. (6), Eq. (8) |
| $\#_s$ | number of data for task $s$ | Eq. (8) |
| $\boldsymbol{H}_s$ | $\Phi(s, \mathbf{X})\Phi(s, \mathbf{X})^T$ | Eq. (7) |
| $\boldsymbol{e}_s$ | standard basis vector in $\mathbb{R}^d$ | Page 5 |
| $\epsilon_s$ | $1/\psi_s$ | Page 6 |
| $f_s$ | output of linear self-attention for the task $s$ | Page 5 |
| $\boldsymbol{\eta}_s$ | $\boldsymbol{v}_s + \boldsymbol{w}_s$ | Page 6 |
| $\boldsymbol{\rho}_s$ | $\boldsymbol{v}_s - \boldsymbol{w}_s$ | Page 6 |
| $g_s$ | $\boldsymbol{\eta}_s^T \boldsymbol{H}_s \boldsymbol{\eta}_s$ | Eq. (10) |
| $h_s$ | $\boldsymbol{\rho}_s^T \boldsymbol{H}_s \boldsymbol{\rho}_s$ | Eq. (10) |
| $\boldsymbol{H}_s^0$ and $\boldsymbol{H}_s^1$ | decomposition of $\boldsymbol{H}_s$ | Page 6 |
| $f_s^0$ and $f_s^1$ | zero-th and first order solution of $f_s$ | Page 6 |
| $a_s$ | $\#_s \psi_s (\Lambda_s^2 + 1)/N$ | Eq. (12) |
| $g_s^0$ and $h_s^0$ | zero-th order terms of $g_s$ and $h_s$ | Page 6 |
| $g_s^1$ and $h_s^1$ | first order terms of $g_s$ and $h_s$ | Page 6 |
| $\mathcal{C}$ | $tD$, optimal compute | Page 8 |

Table 3: Notation and definition with the corresponding index

- $g_s$ **and** $h_s$

$$g_s = \boldsymbol{\eta}_s \cdot \boldsymbol{H}_s \boldsymbol{\eta}_s, \quad h_s = \boldsymbol{\rho}_s \cdot \boldsymbol{H}_s \boldsymbol{\rho}_s.$$

- **In-context learning dynamics Eq.** (9)

$$\dot{\boldsymbol{v}}_s = -\frac{\#_s}{N}\left(f_s - \Lambda_s\right)\boldsymbol{H}_s \boldsymbol{w}_s, \; \dot{\boldsymbol{w}}_s = -\frac{\#_s}{N}\left(f_s - \Lambda_s\right)\boldsymbol{H}_s \boldsymbol{v}_s \qquad (19)$$

- **Test loss Eq.** (15)

$$L(t) = \mathbb{E}_{\boldsymbol{x} \sim \mathcal{P}_X, s \sim \mathcal{P}_\alpha}\left[\ell\left(f(\Phi(s, \mathbf{X}); \boldsymbol{\theta}), y\right)\right] \approx \frac{1}{2}\sum_{s=1}^{\mathcal{N}_s} \mathcal{P}_\alpha(S = s)\left[f_s^0(t) - \Lambda_s\right]^2. \quad (20)$$

To navigate the paper, we also present a table for notation and definitions with the corresponding index in Table 3.

## D  IN-CONTEXT LEARNING DYNAMICS

In D.1, we derive a simpler form of the empirical loss function Eq. (8). In D.2 we present details of the learning dynamics Eq. (9).

### D.1  FORMULATION OF EMPIRICAL LOSS FUNCTION

By definition, the output of self-attention is

$$f\left(\Phi(s, \mathbf{X}); \boldsymbol{\theta}\right) = \left[\boldsymbol{V}\Phi\Phi^T \boldsymbol{W}_{KQ}\Phi\right]_{s, \psi_s + 1} = \left[\boldsymbol{V}\Phi\Phi^T \boldsymbol{W}\hat{\boldsymbol{\phi}}\right]_s$$

where $\boldsymbol{W}, \boldsymbol{V}$ are defined by

$$\boldsymbol{W} = \begin{pmatrix} \boldsymbol{w}_1 & \cdots & \boldsymbol{w}_{\mathcal{N}_s} \end{pmatrix} \in \mathbb{R}^{(\mathcal{N}_s + 1) \times \mathcal{N}_s}, \forall i \in \{1, \ldots, \mathcal{N}_s\} : \boldsymbol{w}_i \in \mathbb{R}^{\mathcal{N}_s + 1},$$

$$\boldsymbol{V}^T = \begin{pmatrix} \boldsymbol{v}_1 & \cdots & \boldsymbol{v}_{\mathcal{N}_s} \end{pmatrix} \in \mathbb{R}^{(\mathcal{N}_s + 1) \times \mathcal{N}_s}, \; \forall i \in \{1, \ldots, \mathcal{N}_s\} : \boldsymbol{v}_i \in \mathbb{R}^{\mathcal{N}_s + 1}.$$

**Deriving $H_s$.** To derive $H_s$ (Eq. (7)), we decompose the in-context data point $\Phi(s, \mathbf{X})$ (Eq. (4)) for the task $s$ as

$$\Phi(s, \mathbf{X}) = \begin{bmatrix} \mathbf{P}_s & \hat{\boldsymbol{\phi}}_s \\ \mathbf{y}_s^T & 0 \end{bmatrix} \tag{21}$$

where

$$\mathbf{P}_s = \begin{bmatrix} \boldsymbol{\phi}_s^{(1)} & \cdots & \boldsymbol{\phi}_s^{(\psi_s)} \end{bmatrix} \in \mathbb{R}^{\mathcal{N}_s \times \psi_s}, \quad \mathbf{y}_s = \begin{bmatrix} y_s^{(1)} \\ \vdots \\ y_s^{(\psi_s)} \end{bmatrix} \in \mathbb{R}^{\psi_s}, \tag{22}$$

and $\hat{\boldsymbol{\phi}}_s = \boldsymbol{\phi}(s, \hat{\mathbf{x}})$. Then

$$\begin{aligned} \Phi(s, \mathbf{X})(\Phi(s, \mathbf{X}))^T &= \begin{bmatrix} \mathbf{P}_s & \hat{\boldsymbol{\phi}}_s \\ \mathbf{y}_s^T & 0 \end{bmatrix} \begin{bmatrix} \mathbf{P}_s^T & \mathbf{y}_s \\ \hat{\boldsymbol{\phi}}_s^T & 0 \end{bmatrix} \\ &= \begin{bmatrix} \mathbf{P}_s \mathbf{P}_s^T + \hat{\boldsymbol{\phi}}_s \hat{\boldsymbol{\phi}}_s^T & \mathbf{P}_s \mathbf{y}_s \\ \mathbf{y}_s^T \mathbf{P}_s^T & \mathbf{y}_s^T \mathbf{y}_s \end{bmatrix}. \end{aligned} \tag{23}$$

There are four terms for us to compute to get the form of $H_s$ in Eq. (23): (i) the first one is

$$\mathbf{P}_s \mathbf{P}_s^T = \begin{bmatrix} \boldsymbol{\phi}_s^{(1)} & \cdots & \boldsymbol{\phi}_s^{(\psi_s)} \end{bmatrix} \begin{bmatrix} (\boldsymbol{\phi}_s^{(1)})^T \\ \vdots \\ (\boldsymbol{\phi}_s^{(\psi_s)})^T \end{bmatrix} = \sum_{j=1}^{\psi_s} \boldsymbol{\phi}_s^{(j)} (\boldsymbol{\phi}_s^{(j)})^T = \mathrm{diag}(\mathbf{e}_s)\psi_s \tag{24}$$

where the standard basis vector in $\mathbb{R}^{\mathcal{N}_s}$ is $\mathbf{e}_s = (0 \;\; \cdots \;\; 0 \;\; 1 \;\; 0 \;\; \cdots)^T \in \mathbb{R}^{\mathcal{N}_s}$ for $s \in \{1, \ldots, \mathcal{N}_s\}$ such that the only nonzero component of $\mathbf{e}_s$ is its $s$-th component; (ii) the second one is $\hat{\boldsymbol{\phi}}_s \hat{\boldsymbol{\phi}}_s^T = \mathrm{diag}(\mathbf{e}_s)$, which can be easily verified; (iii) the third one is

$$\mathbf{P}_s \mathbf{y}_s = \begin{bmatrix} \boldsymbol{\phi}_s^{(1)} & \cdots & \boldsymbol{\phi}_s^{(\psi_s)} \end{bmatrix} \begin{bmatrix} y_s^{(1)} \\ \vdots \\ y_s^{(\psi_s)} \end{bmatrix} = \sum_{j=1}^{\psi_s} \boldsymbol{\phi}_s^{(j)} y_s^{(j)} = \psi_s \Lambda_s \mathbf{e}_s;$$

(iv) the final one is $\mathbf{y}_s^T \mathbf{y}_s = \sum_j^{\psi_s} (y_s^{(j)})^2 = \psi_s \Lambda_s^2$. Combining these terms gives us the form of $H_s$:

$$H_s = \begin{bmatrix} \mathrm{diag}\left((\psi_s + 1)\mathbf{e}_s\right) & \psi_s \Lambda_s \mathbf{e}_s \\ \psi_s \Lambda_s \mathbf{e}_s^T & \psi_s \Lambda_s^2 \end{bmatrix}.$$

As a result, the output of self-attention for the in-context data point $\Phi(s^{(n)}, \mathbf{X}^{(n)})$ will be

$$f\left(\Phi(s^{(n)}, \mathbf{X}^{(n)})\right) = \mathbf{v}_s^T H_{s^{(n)}} \mathbf{W} \hat{\boldsymbol{\phi}}_{s^{(n)}},$$

which gives us the empirical loss as

$$\begin{aligned} \tilde{L} &\overset{\text{a}}{=} \frac{1}{2N} \sum_{n=1}^{N} \left[ f\left(\Phi(s^{(n)}, \mathbf{X}^{(n)}); \boldsymbol{\theta}\right) - \hat{y}^{(n)} \right]^2 \\ &\overset{\text{b}}{=} \frac{1}{2N} \sum_{n=1}^{N} \left[ \mathbf{v}_{s^{(n)}}^T H_{s^{(n)}} \mathbf{W} \hat{\boldsymbol{\phi}}_{s^{(n)}} - \Lambda_{s^{(n)}} \sum_{k=1}^{\mathcal{N}_s} \phi_k(s^{(n)}, \hat{\mathbf{x}}^{(n)}) \right]^2 \\ &\overset{\text{c}}{=} \frac{1}{2N} \sum_{n=1}^{N} \sum_{k=1}^{\mathcal{N}_s} \left[ \mathbf{v}_{s^{(n)}}^T H_{s^{(n)}} \mathbf{w}_{s^{(n)}} - \Lambda_{s^{(n)}} \right]^2 \left(\phi_k(s^{(n)}, \hat{\mathbf{x}}^{(n)})\right)^2 \\ &\overset{\text{d}}{=} \frac{1}{2N} \sum_{n=1}^{N} \left[ \mathbf{v}_{s^{(n)}}^T H_{s^{(n)}} \mathbf{w}_{s^{(n)}} - \Lambda_{s^{(n)}} \right]^2 \\ &\overset{\text{e}}{=} \frac{1}{2} \sum_{s=1}^{\mathcal{N}_s} \frac{\#_s}{N} \left( \mathbf{v}_s^T H_s \mathbf{w}_s - \Lambda_s \right)^2, \end{aligned} \tag{25}$$

where we use the definition of empirical loss in a, the definition of target $y$ in b, the decomposition of $\mathbf{W}$ Eq. (5) in c, $\phi_k(s, \hat{\mathbf{x}}) = \pm\delta_{s,k}$ according to definition of $\phi$ Eq. (2) in d, and recall that $\#_s$ denotes the number of in-context data points with $s^{(n)} = s$ for $n \in \{1, \ldots, N\}$ in e.

## D.2 IN-CONTEXT LEARNING DYNAMICS AS ODE SYSTEMS

We adopt the continuous time limit of gradient descent, i.e., gradient flow, to perform the empirical loss minimization

$$\dot{\boldsymbol{V}} = -\nabla_{\boldsymbol{V}} \tilde{L}(\boldsymbol{V}, \boldsymbol{W}), \quad \dot{\boldsymbol{W}} = -\nabla_{\boldsymbol{W}} \tilde{L}(\boldsymbol{V}, \boldsymbol{W}). \tag{26}$$

Specifically, for empirical loss function Eq. (8), we can directly obtain the learning dynamics as non-linear ODE systems: $\forall s \in \{1, \dots, \mathcal{N}_s\}$

$$\dot{\boldsymbol{v}}_s = -\frac{\#_s}{N} \left( f_s - \Lambda_s \right) \boldsymbol{H}_s \boldsymbol{w}_s,$$

$$\dot{\boldsymbol{w}}_s = -\frac{\#_s}{N} \left( f_s - \Lambda_s \right) \boldsymbol{H}_s \boldsymbol{v}_s$$

where recall that we denote $f_s = \boldsymbol{v}_s^T \boldsymbol{H}_s \boldsymbol{w}_s$.

## E SOLUTION OF SELF-ATTENTION

In E.1, we derive the ODEs of $g_s$ and $h_s$, while we solve the ODEs to the zero-th and first order of $\epsilon_s$ under Assumption 3.1 in E.2 and J, respectively.

### E.1 ODEs OF $g_s$ AND $h_s$

There are four steps to derive ODEs for $g_s$ and $h_s$, which will be discussed as follows.

**Step I: change of variable.** As discussed earlier, the ODE of $\boldsymbol{v}_s$ Eq. (9) is non-linear with respect to $\boldsymbol{w}_s$, thus we transform it to a more symmetrical form by the change of variable: let

$$\boldsymbol{\eta}_s = \boldsymbol{v}_s + \boldsymbol{w}_s, \quad \boldsymbol{\rho}_s = \boldsymbol{v}_s - \boldsymbol{w}_s,$$

then the dynamics of $\boldsymbol{\eta}_s \in \mathbb{R}^{\mathcal{N}_s+1}$ and $\boldsymbol{\rho}_s \in \mathbb{R}^{\mathcal{N}_s+1}$ can be obtained according to Eq. (9)

$$\dot{\boldsymbol{\eta}}_s = -\frac{\#_s}{N} \left( \frac{g_s - h_s}{4} - \Lambda_s \right) \boldsymbol{H}_s \boldsymbol{\eta}_s, \; \dot{\boldsymbol{\rho}}_s = \frac{\#_s}{N} \left( \frac{g_s - h_s}{4} - \Lambda_s \right) \boldsymbol{H}_s \boldsymbol{\rho}_s, \tag{27}$$

as a result, the ODE of $\boldsymbol{\eta}_s$ is non-linear with respect to $\boldsymbol{\eta}_s$ while that of $\boldsymbol{\rho}_s$ is non-linear to $\boldsymbol{\rho}_s$.

**Step II: deriving ODEs to different orders of $\epsilon_s$.** According to the definition of $\boldsymbol{H}_s$ in Eq. (7), we can rewrite $\boldsymbol{H}_s$ as a sum of two matrices

$$\boldsymbol{H}_s = \psi_s \left( \boldsymbol{H}_s^0 + \epsilon_s \boldsymbol{H}_s^1 \right)$$

where

$$\boldsymbol{H}_s^0 = \begin{bmatrix} \text{diag}(\boldsymbol{e}_s) & \Lambda_s \boldsymbol{e}_s \\ \Lambda_s \boldsymbol{e}_s^T & \Lambda_s^2 \end{bmatrix}, \boldsymbol{H}_s^1 = \begin{bmatrix} \text{diag}(\boldsymbol{e}_s) & 0 \\ 0 & 0 \end{bmatrix}$$

and $\epsilon_s = 1/\psi_s \ll 1$ given Assumption 3.1. Therefore we can treat $\epsilon \boldsymbol{H}_s^1$ as an insignificant perturbation in the dynamics Eq. (27). Let the solutions of Eq. (27) be

$$\boldsymbol{\eta}_s = \boldsymbol{\eta}_s^0 + \epsilon_s \boldsymbol{\eta}_s^1, \quad \boldsymbol{\rho}_s = \boldsymbol{\rho}_s^0 + \epsilon_s \boldsymbol{\rho}_s^1$$

such that $\boldsymbol{\eta}_s^1$ and $\boldsymbol{\rho}_s^1$ are treated as perturbations to $\boldsymbol{\eta}_s^0$ and $\boldsymbol{\rho}_s^0$, respectively. Then, according to the definitions of $g_s$ and $h_s$, we can also write $g_s$ and $h_s$ in the perturbed form

$$\begin{aligned} g_s := g_s^0 + \epsilon_s g_s^1 &= \boldsymbol{\eta}_s \cdot \boldsymbol{H}_s \boldsymbol{\eta}_s \\ &= \psi_s \left( \boldsymbol{\eta}_s^0 + \epsilon_s \boldsymbol{\eta}_s^1 \right) \cdot \left( \boldsymbol{H}_s^0 + \epsilon_s \boldsymbol{H}_s^1 \right) \left( \boldsymbol{\eta}_s^0 + \epsilon_s \boldsymbol{\eta}_s^1 \right) \\ &= \psi_s \boldsymbol{\eta}_s^0 \cdot \boldsymbol{H}_s^0 \boldsymbol{\eta}_s^0 + \psi_s \epsilon_s \left( \boldsymbol{\eta}_s^0 \cdot \boldsymbol{H}_s^1 \boldsymbol{\eta}_s^0 + 2\boldsymbol{\eta}_s^1 \cdot \boldsymbol{H}_s^0 \boldsymbol{\eta}_s^0 \right) + \mathcal{O}(\epsilon_s^2) \end{aligned} \tag{28}$$

$$\begin{aligned} h_s := h_s^0 + \epsilon_s h_s^1 &= \boldsymbol{\rho}_s \cdot \boldsymbol{H}_s \boldsymbol{\rho}_s \\ &= \psi_s \boldsymbol{\rho}_s^0 \cdot \boldsymbol{H}_s^0 \boldsymbol{\rho}_s^0 + \psi_s \epsilon_s \left( \boldsymbol{\rho}_s^0 \cdot \boldsymbol{H}_s^1 \boldsymbol{\rho}_s^0 + 2\boldsymbol{\rho}_s^1 \cdot \boldsymbol{H}_s^0 \boldsymbol{\rho}_s^0 \right) + \mathcal{O}(\epsilon_s^2). \end{aligned} \tag{29}$$

Putting the above perturbation forms back to Eq. (27) to the first order of $\epsilon_s$, we have

$$\dot{\boldsymbol{\eta}}_s^0 + \epsilon_s \dot{\boldsymbol{\eta}}_s^1 = -\frac{\psi_s \#_s}{4N} \left[ g_s^0 - h_s^0 + \epsilon_s \left( g_s^1 - h_s^1 \right) - 4\Lambda_s \right] \left[ \boldsymbol{H}_s^0 \boldsymbol{\eta}_s^0 + \epsilon_s \left( \boldsymbol{H}_s^1 \boldsymbol{\eta}_s^0 + \boldsymbol{H}_s^0 \boldsymbol{\eta}_s^1 \right) \right] + \mathcal{O}(\epsilon_s^2),$$

$$\dot{\boldsymbol{\rho}}_s^0 + \epsilon_s \dot{\boldsymbol{\rho}}_s^1 = \frac{\psi_s \#_s}{4N} \left[ g_s^0 - h_s^0 + \epsilon_s \left( g_s^1 - h_s^1 \right) - 4\Lambda_s \right] \left[ \boldsymbol{H}_s^0 \boldsymbol{\rho}_s^0 + \epsilon_s \left( \boldsymbol{H}_s^1 \boldsymbol{\rho}_s^0 + \boldsymbol{H}_s^0 \boldsymbol{\rho}_s^1 \right) \right] + \mathcal{O}(\epsilon_s^2).$$

$$(30)$$

Matching both sides of Eq. (30) to the zero-th and first order of $\epsilon_s$ respectively gives us

$$\dot{\boldsymbol{\eta}}_s^0 = -\frac{\psi_s \#_s}{4N} \left( g_s^0 - h_s^0 - 4\Lambda_s \right) \boldsymbol{H}_s^0 \boldsymbol{\eta}_s^0, \quad \dot{\boldsymbol{\rho}}_s^0 = \frac{\psi_s \#_s}{4N} \left( g_s^0 - h_s^0 - 4\Lambda_s \right) \boldsymbol{H}_s^0 \boldsymbol{\rho}_s^0 \qquad (31)$$

and

$$\dot{\boldsymbol{\eta}}_s^1 = -\frac{\psi_s \#_s}{4N} \left[ \left( g_s^0 - h_s^0 - 4\Lambda_s \right) \left( \boldsymbol{H}_s^1 \boldsymbol{\eta}_s^0 + \boldsymbol{H}_s^0 \boldsymbol{\eta}_s^1 \right) + \left( g_s^1 - h_s^1 \right) \boldsymbol{H}_s^0 \boldsymbol{\eta}_s^0 \right],$$

$$\dot{\boldsymbol{\rho}}_s^1 = \frac{\psi_s \#_s}{4N} \left[ \left( g_s^0 - h_s^0 - 4\Lambda_s \right) \left( \boldsymbol{H}_s^1 \boldsymbol{\rho}_s^0 + \boldsymbol{H}_s^0 \boldsymbol{\rho}_s^1 \right) + \left( g_s^1 - h_s^1 \right) \boldsymbol{H}_s^0 \boldsymbol{\rho}_s^0 \right].$$

$$(32)$$

**Step III: deriving ODEs for $g_s$ and $h_s$ to zero-th and first orders of $\epsilon_s$.** We can now obtain the ODEs for $g_s$ and $h_s$ to the zero-th order of $\epsilon_s$ by directly applying the definitions and Eq. (31):

**Zero-th Order:**

$$\begin{aligned} \dot{g}_s^0 &= \psi_s \frac{d}{dt} \boldsymbol{\eta}_s^0 \cdot \boldsymbol{H}_s^0 \boldsymbol{\eta}_s^0 = -\psi_s \left( g_s^0 - h_s^0 - 4\Lambda_s \right) \frac{\psi_s \#_s \boldsymbol{\eta}_s^0 \cdot \boldsymbol{H}_s^0 \boldsymbol{H}_s^0 \boldsymbol{\eta}_s^0}{2N} \\ &\overset{\mathrm{a}}{=} - \left( g_s^0 - h_s^0 - 4\Lambda_s \right) \frac{a_s \psi_s \boldsymbol{\eta}_s^0 \cdot \boldsymbol{H}_s^0 \boldsymbol{\eta}_s^0}{2} \\ &= - \left( g_s^0 - h_s^0 - 4\Lambda_s \right) \frac{a_s g_s^0}{2}, \\ \dot{h}_s^0 &= \left( g_s^0 - h_s^0 - 4\Lambda_s \right) \frac{a_s h_s^0}{2}, \end{aligned}$$

$$(33)$$

where we use $(\boldsymbol{H}_s^0)^2 = (\Lambda_s^2 + 1)\boldsymbol{H}_s^0$ in a and derive the equation for $h_s^0$ in a similar way. Suppose that we obtain the solution to the zero-th order by solving the above ODEs, then, according to the definition of $g_s^1$ in Eq. (28) and the definition of $h_s^1$ in Eq. (29), we only need to find the solutions of

$$m_s = \psi_s \boldsymbol{\eta}_s^1 \cdot \boldsymbol{H}_s^0 \boldsymbol{\eta}_s^0 \text{ and } n_s = \psi_s \boldsymbol{\rho}_s^1 \cdot \boldsymbol{H}_s^0 \boldsymbol{\rho}_s^0 \qquad (34)$$

to derive the solution to the first order of $\epsilon_s$, since we can obtain $\boldsymbol{\eta}_s^0 \cdot \boldsymbol{H}_s^1 \boldsymbol{\eta}_s^0$ and $\boldsymbol{\rho}_s^0 \cdot \boldsymbol{H}_s^1 \boldsymbol{\rho}_s^0$ using the solutions of Eq. (33)(see Appendix J.1). This means that based on Eq. (32), we need to solve the following ODEs:

**First Order:**

$$\begin{aligned} \dot{m}_s &= \psi_s \boldsymbol{\eta}_s^1 \cdot \boldsymbol{H}_s^0 \dot{\boldsymbol{\eta}}_s^0 + \psi_s \dot{\boldsymbol{\eta}}_s^1 \cdot \boldsymbol{H}_s^0 \boldsymbol{\eta}_s^0 \\ &= -\frac{\psi_s^2 \#_s}{4N} \left( g_s^0 - h_s^0 - 4\Lambda_s \right) \left[ \boldsymbol{\eta}_s^1 \cdot \boldsymbol{H}_s^0 \boldsymbol{H}_s^0 \boldsymbol{\eta}_s^0 + \boldsymbol{\eta}_s^0 \cdot \left( \boldsymbol{H}_s^0 \boldsymbol{H}_s^1 \boldsymbol{\eta}_s^0 + \boldsymbol{H}_s^0 \boldsymbol{H}_s^0 \boldsymbol{\eta}_s^1 \right) \right] \\ &\quad - \frac{\psi_s^2 \#_s}{4N} (g_s^1 - h_s^1) \boldsymbol{\eta}_s^0 \boldsymbol{H}_s^0 \boldsymbol{H}_s^0 \boldsymbol{\eta}_s^0 \\ &= - \left( g_s^0 - h_s^0 - 4\Lambda_s \right) \left[ \frac{a_s m_s}{2} + \frac{\psi_s^2 \#_s}{4N} \boldsymbol{\eta}_s^0 \cdot \boldsymbol{H}_s^0 \boldsymbol{H}_s^1 \boldsymbol{\eta}_s^0 \right] - (g_s^1 - h_s^1) \frac{a_s g_s^0}{4} \end{aligned}$$

$$(35)$$

and, similarly,

**First Order:**

$$\begin{aligned} \dot{n}_s &= \boldsymbol{\rho}_s^1 \cdot \boldsymbol{H}_s^0 \dot{\boldsymbol{\rho}}_s^0 + \dot{\boldsymbol{\rho}}_s^1 \cdot \boldsymbol{H}_s^0 \boldsymbol{\rho}_s^0 \\ &= \frac{\psi_s^2 \#_s}{4N} \left( g_s^0 - h_s^0 - 4\Lambda_s \right) \left[ \boldsymbol{\rho}_s^1 \cdot \boldsymbol{H}_s^0 \boldsymbol{H}_s^0 \boldsymbol{\rho}_s^0 + \boldsymbol{\rho}_s^0 \cdot \left( \boldsymbol{H}_s^0 \boldsymbol{H}_s^1 \boldsymbol{\rho}_s^0 + \boldsymbol{H}_s^0 \boldsymbol{H}_s^0 \boldsymbol{\rho}_s^1 \right) \right] \\ &\quad + \frac{\psi_s^2 \#_s}{4N} (g_s^1 - h_s^1) \boldsymbol{\rho}_s^0 \boldsymbol{H}_s^0 \boldsymbol{H}_s^0 \boldsymbol{\rho}_s^0 \\ &= \left( g_s^0 - h_s^0 - 4\Lambda_s \right) \left[ \frac{a_s n_s}{2} + \frac{\psi_s^2 \#_s}{4N} \boldsymbol{\rho}_s^0 \cdot \boldsymbol{H}_s^0 \boldsymbol{H}_s^1 \boldsymbol{\rho}_s^0 \right] + (g_s^1 - h_s^1) \frac{a_s h_s^0}{4}. \end{aligned}$$

$$(36)$$

Then we can write the solution of $g_s^1$ and $h_s^1$ as

$$g_s^1 = \psi_s \boldsymbol{\eta}_s^0 \boldsymbol{H}_s^1 \boldsymbol{\eta}_s^0 + 2m_s, \ h_s^1 = \psi_s \boldsymbol{\rho}_s^0 \boldsymbol{H}_s^1 \boldsymbol{\rho}_s^0 + 2n_s.$$

We will solve the zero-th order ODEs in the next section and discuss the solution of the first order ODEs in J.

E.2  SOLUTION OF SELF-ATTENTION: ZERO-TH ORDER

Eq. (33) is still hard to solve. Fortunately, we note that

$$\frac{d}{dt}g_s^0 h_s^0 = \left(g_s^0 - h_s^0 - 4\Lambda_s\right)\frac{a_s(g_s^0 h_s^0 - g_s^0 h_s^0)}{2} = 0, \tag{37}$$

which immediately implies that

$$\forall t \geq 0 : g_s^0 h_s^0 = 2C_s \tag{38}$$

is a constant, i.e., $g_s^0 h_s^0$ is a conserved quantity of the learning dynamics. This important quantity allows us to rewrite Eq. (33) as

$$\begin{aligned}
\dot{g}_s^0 &= 2a_s\Lambda_s g_s^0 - a_s(g_s^0)^2/2 + a_s C_s, \\
\dot{h}_s^0 &= -2a_s\Lambda_s g_s^0 - a_s(h_s^0)^2/2 + a_s C_s,
\end{aligned} \tag{39}$$

which are exactly the *Riccati equation*. To determine $C_s$, we note that

$$C_s = (\boldsymbol{\eta}_s^0 \cdot \boldsymbol{H}_s^0 \boldsymbol{\eta}_s^0)(\boldsymbol{\rho}_s^0 \cdot \boldsymbol{H}_s^0 \boldsymbol{\rho}_s^0)_{t=0} = \left(\|\boldsymbol{v}_s^0(0)\|_{\boldsymbol{H}_s}^2 + \|\boldsymbol{w}_s^0(0)\|_{\boldsymbol{H}_s}^2\right)^2 - 4\boldsymbol{v}_s^0(0) \cdot \boldsymbol{H}_s \boldsymbol{w}_s^0(0), \tag{40}$$

where we use $\|a\|_H^2 = a^T H a$ for a positive definite matrix $H$ and vector $a$ and we note that $C_s = 0$ if $\boldsymbol{v}_s^0(0) = \pm\boldsymbol{w}_s^0(0)$. In the following, we adopt a series of change of variable to solve Eq. (39). For simplicity, we omit the $s$ subscript and recover it in the final solution.

**Step I.**  Let

$$p = -\frac{ag}{2}, \; q = -\frac{ah}{2}, \tag{41}$$

then we can transform Eq. (39) to

$$\begin{aligned}
\dot{p} &= -\frac{a\dot{g}}{2} = 2a\Lambda p + p^2 - \frac{a^2 C}{2} \\
\dot{q} &= -\frac{a\dot{h}}{2} = -2a\Lambda q + q^2 - \frac{a^2 C}{2}.
\end{aligned} \tag{42}$$

**Step II.**  Let $p = -\dot{\gamma}/\gamma, q = -\dot{\theta}/\theta$, then

$$\dot{p} = -\frac{\ddot{\gamma}\gamma - \dot{\gamma}^2}{\gamma^2} = -\frac{\ddot{\gamma}}{\gamma} + p^2. \tag{43}$$

Putting it back to Eq. (42), we can obtain the ODE of $\gamma$ and $\theta$ as follows

$$\ddot{\gamma} + 2a\Lambda\left(-\frac{\dot{\gamma}}{\gamma}\right)\gamma - \frac{a^2 C\gamma}{2} = 0 \implies \ddot{\gamma} - 2a\Lambda\dot{\gamma} - \frac{a^2 C\gamma}{2} = 0,$$

$$\ddot{\theta} + 2a\Lambda\dot{\theta} - \frac{a^2 C\theta}{2} = 0. \tag{44}$$

**Step III.**  Eq. (44) are just the second-order linear ODEs, which can be solved following a standard approach. Specifically, let

$$\gamma = e^{ct}, \theta = e^{bt},$$

then putting them back into Eq. (44) gives us

$$\begin{aligned}
c^2 - 2\Lambda ac - \frac{a^2 C}{2} = 0 &\implies c = \Lambda a \pm \frac{\sqrt{4\Lambda^2 a^2 + 2a^2 C}}{2} \\
b^2 + 2\Lambda ab - \frac{a^2 C}{2} = 0 &\implies b = -\Lambda a \pm \frac{\sqrt{4\Lambda^2 a^2 + 2a^2 C}}{2}.
\end{aligned} \tag{45}$$

For ease of notation, we denote

$$S = \Lambda a, \; \xi = \frac{\sqrt{4S^2 + 2a^2 C}}{2}, \; \sigma_+ = S + \xi, \; \sigma_- = S - \xi \tag{46}$$

then solutions of $\gamma$ and $\theta$ are

$$\gamma = Ae^{\sigma_+ t} + Be^{\sigma_- t}, \quad \theta = Ee^{-\sigma_- t} + Fe^{-\sigma_+ t} \tag{47}$$

where $A, B, E,$ and $F$ are constants that need to be determined by the initial condition. Now we are ready to recover the solution to Eq. (33):

$$
\begin{aligned}
g &= -\frac{2p}{a} = 2\frac{\dot{\gamma}}{a\gamma} = \frac{2}{a}\frac{\sigma_+ Ae^{\sigma_+ t} + B\sigma_- e^{\sigma_- t}}{Ae^{\sigma_+ t} + Be^{\sigma_- t}} \\
h &= -\frac{qa}{2} = 2\frac{\dot{\theta}}{a\theta} = -\frac{2}{a}\frac{\sigma_- Ee^{-\sigma_- t} + \sigma_+ Fe^{-\sigma_+ t}}{Ee^{-\sigma_- t} + Fe^{-\sigma_+ t}}.
\end{aligned} \tag{48}
$$

Recall that $f = (g - h)/4$ in Eq. (27), we are now able to derive the solution of the zero-th order solution of self-attention:

$$
\begin{aligned}
f &= \frac{g - h}{4} \\
&= \frac{1}{2a}\left[\frac{\sigma_+ Ae^{\rho t} + B\sigma_- e^{\sigma_- t}}{Ae^{\sigma_+ t} + Be^{\sigma_- t}} + \frac{\sigma_- Ee^{-\sigma_- t} + \sigma_+ Fe^{-\sigma_+ t}}{Ee^{-\sigma_- t} + Fe^{-\sigma_+ t}}\right] \\
&= \frac{1}{2a}\left[\sigma_+ + \sigma_- + \frac{B(\sigma_- - \sigma_+)e^{\sigma_- t}}{Ae^{\sigma_+ t} + Be^{\sigma_- t}} + \frac{F(\sigma_+ - \sigma_-)e^{\sigma_- t}}{Ee^{\sigma_+ t} + Fe^{\sigma_- t}}\right] \\
&= \Lambda + \frac{\lambda}{2}\left[\frac{F}{Ee^{a\lambda t} + F} - \frac{B}{Ae^{a\lambda t} + B}\right]
\end{aligned} \tag{49}
$$

where we define

$$\lambda = \sqrt{4\Lambda^2 + 2C},$$

in the last equality.

**Step IV.** It is now left for us to determine $A, B, E$ and $F$ according to the initial condition, which can be obtained by noting the conserved quantity $gh$ in Eq. (38) and can be satisfied when

$$\frac{\sigma_+ Ae^{\sigma_+ t} + B\sigma_- e^{\sigma_- t}}{Ae^{\sigma_+ t} + Be^{\sigma_- t}}\frac{\sigma_- Ee^{-\sigma_- t} + \sigma_+ Fe^{-\sigma_+ t}}{Ee^{-\sigma_- t} + Fe^{-\sigma_+ t}} = 2C \tag{50}$$

$$\implies \sigma_+^2 AF + \sigma_-^2 BE + \frac{a^2 C}{2}(AF + BE) = 0. \tag{51}$$

Noticing that

$$\sigma_+^2 = S^2 + 2S\xi + \xi^2, \quad \sigma_-^2 = S^2 - 2S\xi + \xi^2,$$

we can further simplify Eq. (51) to

$$AF\sigma_+ = BE\sigma_-. \tag{52}$$

On the other hand, considering the initial condition by letting $t = 0$ in Eq. (49)

$$f(0) = \Lambda + \frac{\lambda}{2}\left[\frac{1}{E/F + 1} - \frac{1}{A/B + 1}\right], \tag{53}$$

if we denote

$$P = \frac{E}{F}, \quad \bar{Q} = \frac{\sigma_-}{\sigma_+} = \frac{2\Lambda - \lambda}{2\Lambda + \lambda}, \quad D = \frac{2(f(0) - \Lambda)}{\lambda}, \tag{54}$$

then one can easily see that $A/B = \bar{Q}E/F$ and Eq. (53) becomes

$$D(1 + P)(1 + P\bar{Q}) + P - P\bar{Q} = 0$$

$$\implies P = \frac{-[D(\bar{Q} + 1) + 1 - \bar{Q}] \pm \sqrt{[D(\bar{Q} + 1) + 1 - \bar{Q}]^2 - 4D^2\bar{Q}}}{2D\bar{Q}}. \tag{55}$$

Recall that $g, h > 0$ according to their definitions ($\boldsymbol{H}$ is positive-definite), we only take the minus sign in the above solution, which can be simplified by conducting some tedious algebra as

$$P = \frac{4f(0)\Lambda + 2C + \sqrt{4\Lambda^2 + 2C}\sqrt{4f(0)^2 + 2C}}{2(f(0) - \Lambda)(\sqrt{4\Lambda^2 + 2C}) - 2\Lambda} \tag{56}$$

We can now summarize the solution of self-attention to the zero-th order of $\epsilon_s$ to prove Theorem 3.1 by recovering the subscript $s$ in Eq. (49) with solution of $P$ in Eq. (56):

$$f_s^0(t) = \Lambda_s + \frac{\lambda_s}{2} \left[ \frac{1}{1 + P_s \exp(a_s \lambda_s t)} - \frac{1}{1 + Q_s \exp(a_s \lambda_s t)} \right] \tag{57}$$

where $a_s = \#_s \psi_s (\Lambda_s^2 + 1)/N$, and

$$\lambda_s = \sqrt{4\Lambda_s^2 + 2C_s}, \; P_s = \frac{4 f_s^0(0) \Lambda_s + 2 C_s + \lambda_s \sqrt{4(f_s^0(0))^2 + 2 C_s}}{2(f_s^0(0) - \Lambda_s)(\lambda_s - 2\Lambda_s)}, \; Q_s = P_s \frac{2\Lambda_s - \lambda_s}{2\Lambda_s + \lambda_s}.$$

### E.3 Zero-th Order Solution for special balanced initialization

When $\boldsymbol{v}_s^0(0) = \boldsymbol{w}_s^0(0)$, we have $C_s = 0$, which implies that $F = \sigma_- = 0$ according to Eq. (46) and Eq. (51), and

$$\lambda_s = 2\Lambda_s.$$

As a result, we can rewrite the solution as

$$f_s^0(t) = \Lambda_s - \frac{\Lambda_s}{1 + A/B e^{2 a_s \Lambda_s t}} = \frac{\Lambda_s}{1 + \frac{\Lambda_s - f_s^0(0)}{f_s^0(0)} \exp(-2 a_s \lambda_s t)} \tag{58}$$

where we use the initial condition Eq. (53) in the second equality.

## F Neural Scaling Laws of Self-Attention

We first present the overall procedure for deriving neural scaling laws in F.1, then discuss them in detail for fixed context sequence length in F.2 and for varied context length in F.3. We use $a \sim b$ to mean that $a$ is approximately equal to $b$ (by neglecting irrelevant coefficients and constants) if $a, b \in \mathbb{R}$.

### F.1 Procedure

For convenience, we first present the test loss described in the main paper

$$L(t) \approx \frac{1}{2} \sum_{s=1}^{\mathcal{N}_s} \mathcal{P}_\alpha(S = s) \left[ f_s^0(t) - \Lambda_s \right]^2. \tag{59}$$

Note that as $a_s$ is determined by the dataset and it satisfies the following relation

$$a_s = \frac{\psi_s \#_s (\Lambda_s^2 + 1)}{N} \rightarrow \mathcal{P}_\alpha(S = s) \psi_s (\Lambda_s^2 + 1) = Z s^{-\alpha} \psi_s (\Lambda_s^2 + 1) \tag{60}$$

as the number of training data points $N \to \infty$. Furthermore, as long as $\#_s \neq 0$ such that $a_s \neq 0$, a crucial property of $f_s^0(t)$ is

$$\lim_{t \to \infty} f_s^0(t) = \Lambda_s, \tag{61}$$

otherwise it would be

$$\lim_{t \to \infty} f_s^0(t) = f_s^0(0) = \Lambda_s + \frac{\lambda_s}{2} \left( \frac{1}{1 + P_s} - \frac{1}{1 + Q_s} \right), \tag{62}$$

which means that self-attention cannot learn the task $s$ if there is no data point for it in the training set and is similar to the one shot learner property of diagonal linear networks in Nam et al. (2024). These properties will be repeatedly applied in the following sections. And we now discuss procedures for deriving different neural scaling laws.

**Size of the model $D$.** To quantify the model size $D$, we assume that there is a cutoff $D$ such that the model cannot learn any task strength $\Lambda_s$ with $s \geq D$. As a result, according to the solution for the task $s$ in Eq. (57), we have

$$\forall t \geq 0, s \geq D : \; f_s^0(t) - \Lambda_s = f_s^0(0) - \Lambda_s. \tag{63}$$

When $D$ is the bottleneck of training, we let $t, N \to \infty$ (they are sufficient for the training) to derive scaling laws respect to $D$. Thus the over all test loss will be

$$L(D) \sim \frac{1}{2} \sum_{s=D}^{\mathcal{N}_s} Z s^{-\alpha} \left[ f_s^0(0) - \Lambda_s \right]^2 \tag{64}$$

To derive the neural scaling law of model size $D$, we only need to find the asymptotic behavior of Eq. (64) by letting $\mathcal{N}_s \to \infty$ and replacing the summation with an integral.

**Training time $t$.** Training time $t$ is equivalent to the number of optimization steps. To investigate the scaling law respect to $t$, we remove the bottleneck caused by the size of model and the number of data points by letting $N \to \infty$ (thus we have Eq. (60)) and $D = \mathcal{N}_s$ (thus Eq. (63) does not satisfy for any $D \in \{1, \ldots, \mathcal{N}_s\}$). As a result, the overall test loss will be

$$L(t) \sim \frac{1}{2} \int_{s=1}^{\infty} Z s^{-\alpha} \left[ f_s^0(t) - \Lambda_s \right]^2 ds \tag{65}$$

where we let $\mathcal{N}_s \to \infty$ and replace the summation with integral. We sill study the asymptotic behavior of Eq. (65) with the Laplace method (Bender & Orszag, 1978) to investigate the time scaling law.

**Number of training data points $N$.** When the training is bottlenecked by $N$, we let $t \to \infty$ (thus Eq. (61) will be satisfied for all task types $s$ if there exist training data points for them otherwise Eq. (62) would be satisfied) and the cutoff $D = \mathcal{N}_s$ (thus Eq. (63) does not satisfy for any $D \in \{1, \ldots, \mathcal{N}_s\}$). As a result, we conclude that the probability of $\lim_{t \to \infty} f_s^0(t) = f_s^0(0)$ is exactly the same as the probability that the training data set $\{\Phi(s^{(n)}, \mathbf{X}^{(n)})\}_{n=1}^{N}$ does not have any training data point for the task $s$, i.e, $\forall n \in \{1, \ldots, N\} : s^{(n)} \neq s$. Therefore, we can rewrite the test loss as

$$\begin{aligned}
L(N) &\sim \frac{1}{2} \int_{s=1}^{\infty} Z s^{-\alpha} \left[ f_s^0(0) - \Lambda_s \right]^2 (1 - \mathcal{P}(S = s))^N ds \\
&= \frac{1}{2} \int_{s=1}^{\infty} Z s^{-\alpha} \left[ f_s^0(0) - \Lambda_s \right]^2 (1 - Z s^{-\alpha})^N ds
\end{aligned} \tag{66}$$

where, again, we let $\mathcal{N}_s \to \infty$ and replace the summation with the integral, and we sill study the asymptotic behavior of Eq. (65) with the Laplace method to investigate the data scaling law.

**Optimal compute $\mathcal{C}$.** This is the case when the number of data points is sufficient for the training ($N \to \infty$), while training time $t$ or the size of model $D$ is the bottleneck given the compute budget $\mathcal{C} = Dt$ such that either $t$ or $D$ scales differently with $\mathcal{C}$. Specifically, if

$$L(t, D) = a_t t^{-\alpha_t} + a_D D^{-\alpha_D},$$

then we can rewrite the test loss as

$$L(D) = a_t \mathcal{C}^{-\alpha_t} D^{\alpha_t} + a_D D^{-\alpha_D}. \tag{67}$$

To obtain the optimal loss given $D$, we let

$$\partial_D L(D) = 0 \implies D = \left( \frac{a_D \alpha_D}{a_t \alpha_t} \right)^{\frac{1}{\alpha_t + \alpha_D}} \mathcal{C}^{\frac{\alpha_t}{\alpha_t + \alpha_D}}, \;\; t = \left( \frac{a_D \alpha_D}{a_t \alpha_t} \right)^{-\frac{1}{\alpha_t + \alpha_D}} \mathcal{C}^{\frac{\alpha_D}{\alpha_t + \alpha_D}}.$$

As a result, we can derive the optimal compute budget test loss as

$$L(\mathcal{C}) \propto \mathcal{C}^{-\alpha_t \alpha_D / (\alpha_t + \alpha_D)} \tag{68}$$

given $\mathcal{C} = tD$, where $\alpha_t, \alpha_D$ can be obtained from the neural scaling laws for time and model size, respectively.

## F.2 Neural Scaling Laws with Fixed Sequence Length and Strength

In this case, according to Eq. (16), the test loss can be written as

$$
L(t) \approx \frac{Z\lambda^2}{8} \sum_{s=1}^{\mathcal{N}_s} s^{-\alpha} \left( \frac{1}{1 + P\exp\left[s^{-\alpha} Z\lambda\psi(\Lambda^2+1)t\right]} - \frac{1}{1 + Q\exp\left[s^{-\alpha} Z\lambda\psi(\Lambda^2+1)t\right]} \right)^2 .
$$

(69)

In the following, we will investigate neural scaling laws using the above test loss and the procedures described in F.1.

**Model Scaling Law.** According to Eq. (64), the model scaling law can be obtained from studying the behavior of

$$
L(D) \sim \int_{s=D}^{\infty} Z s^{-\alpha} \frac{\lambda^2}{8} \left[ \frac{1}{1+P} - \frac{1}{1+Q} \right]^2 ds \propto D^{-\alpha+1}
$$

(70)

where $D$ is the cuttoff of the task such that our model will only learn the first $D-1$ tasks and we let $t \to \infty$.

**Time scaling law.** Let $\mathcal{N}_s \to \infty$ and replace the summation with integral in the test loss, we have (omitting irrelevant coefficients and we denote $\bar{\psi} = Z\lambda\psi(\Lambda^2+1)$ for ease of notation)

$$
L(t) \sim \int_1^{\infty} \left( \frac{1}{e^{-s^{-\alpha}\bar{\psi}t} + P} - \frac{1}{e^{-s^{-\alpha}\bar{\psi}t} + Q} \right)^2 e^{-2s^{-\alpha}\bar{\psi}t - \alpha \ln s} ds
$$

(71)

Now let

$$
F(s) := s^{-\alpha}\bar{\psi} + \frac{\alpha}{t} \ln s
$$

then, applying the Laplace method (Bender & Orszag, 1978), for large $t$

$$
\begin{aligned}
L(t) &\propto \int_1^{\infty} e^{-F(s)t} ds \\
&\overset{\mathrm{a}}{\sim} \int_{c-\varepsilon}^{c+\varepsilon} e^{-(F(c)+F''(c)(s-c)^2)t} ds \\
&\sim e^{-F(c)t} \int_{c-\varepsilon}^{c+\varepsilon} e^{-F''(c)t(s-c)^2} ds \\
&\overset{\mathrm{b}}{\sim} e^{-F(c)t} \frac{\sqrt{2\pi}}{\sqrt{F''(c)t}}
\end{aligned}
$$

(72)

where we expand $F(s)$ around its minimal $F(c)$ in $\mathrm{a}$ and $\mathrm{b}$ is simply the Gauss integral. We can solve $F'(c) = 0$ to determine the value of $c$:

$$
-\alpha c^{-\alpha-1}\bar{\psi} + \frac{\alpha}{ct} = 0 \implies c = (t\bar{\psi})^{\frac{1}{\alpha}}.
$$

(73)

This further gives us

$$
F(c) = t^{-1} + \frac{\ln(\bar{\psi}t)}{t} \implies e^{-F(c)t} = \frac{1}{e\bar{\psi}t}
$$

(74)

and

$$
\begin{aligned}
F''(c) &= \alpha(\alpha+1)\bar{\psi}c^{-\alpha-2} - \frac{\alpha}{c^2 t} \\
&= \alpha(\alpha+1)\bar{\psi}(t\bar{\psi})^{-\frac{(\alpha+2)}{\alpha}} - \frac{\alpha}{\bar{\psi}^{\frac{2}{\alpha}} t^{1+\frac{2}{\alpha}}} \\
&= \alpha^2 \bar{\psi}^{-\frac{2}{\alpha}} t^{-1-\frac{2}{\alpha}}.
\end{aligned}
$$

(75)

Putting $F(c)$ and $F''(c)$ obtained above back to Eq. (72) immediately gives us the time scaling law:

$$
L(t) \sim \frac{\sqrt{2\pi}}{e\alpha\bar{\psi}^{1-\frac{2}{\alpha}}} t^{-1+\alpha^{-1}} \propto t^{-\frac{\alpha-1}{\alpha}}.
$$

(76)

**Data Scaling Law.** In this case the bottleneck of training is the number of data while $t \to \infty$ and $D = \mathcal{N}_s$. According to Eq. (66), the test loss has the form of

$$L(N) \sim \int_1^\infty Zs^{-\alpha} \frac{\lambda^2}{8} \left[ \frac{1}{1+P} - \frac{1}{1+Q} \right]^2 (1 - Zs^{-\alpha})^N ds$$

$$\sim \int_1^\infty e^{-N(\alpha \ln s/N - \ln(1-Zs^{-\alpha}))} ds. \tag{77}$$

We apply the Laplace method again to study the asymptotic behavior of $L(N)$ to derive the data scaling law. Let

$$F(s) = \alpha \frac{\ln s}{N} - \ln(1 - Zs^{-\alpha}), \tag{78}$$

then we can expand $F(s)$ around its minimal $F(c)$ where the value of $c$ is determined by $F'(c) = 0$:

$$F'(s) = \frac{\alpha}{Ns} - \frac{\alpha Zs^{-\alpha-1}}{1 - Zs^{-\alpha}} \tag{79}$$

$$\implies c = ((N+1)Z)^{\frac{1}{\alpha}} \tag{80}$$

The loss function can be written as

$$L(N) \sim e^{-F(c)N} \frac{\sqrt{2\pi}}{\sqrt{F''(c)N}}, \tag{81}$$

where

$$F(c) = \frac{\ln((N+1)Z)}{N} - \ln\left(1 - \frac{1}{N+1}\right)$$

$$\sim N^{-1}\ln(NZ) + N^{-1} \tag{82}$$

where we assume that $N \gg 1$ in the second line. It is now left for us to find the value of $F''(c)$, which is done as follows.

$$F''(s)|_{s=c} = -c^{-2}\frac{\alpha(N+1)}{N} + \frac{\alpha c^{-2}}{1 - Zc^{-\alpha}} + \frac{\alpha}{c}\frac{Z\alpha c^{-\alpha-1}}{(1 - Zc^{-\alpha})^2}$$

$$= \alpha c^{-2} \left( \frac{Zc^{-\alpha}}{1 - Zc^{-\alpha}} + \frac{Z\alpha c^{-\alpha}}{(1 - Zc^{-\alpha})^2} - \frac{1}{N} \right)$$

$$\sim \alpha(\alpha+1)Zc^{-\alpha}c^{-2} \sim \alpha(\alpha+1)Z^{-\frac{2}{\alpha}}N^{-(\alpha+2)/\alpha}$$

where we use $N \gg 1$ again in the last line. Putting $F(c)$ and $F''(c)$ back to Eq. (81) gives us the scaling law with respect to $N$:

$$L(N) \sim \frac{1}{NZe} \frac{\sqrt{2\pi}}{Z^{-\frac{1}{\alpha}}\sqrt{\alpha(\alpha+1)NN^{-1-2/\alpha}}} \propto N^{-\frac{\alpha-1}{\alpha}}. \tag{83}$$

## F.3 Neural Scaling Laws with Varied Sequence Length and Strength

In general, we can derive neural scaling laws with a similar spirit as in the previous section. Additionally, we assume for simplicity that $\Lambda_s^2 \gg 1$ such that $a_s \approx \#_s \psi_s \Lambda_s^2 / N$ and the model is initialized as $\boldsymbol{v}_s^0(0) = \pm \boldsymbol{w}_s^0(0)$ and $f_s^0(0) = O(1)$ for all $s$. As a result, the test loss Eq. (17):

$$L(t) \propto \frac{Z}{2} \sum_{s=1}^{\mathcal{N}_s} s^{-\alpha-2\gamma} \left[ \frac{\Delta \exp(-2a_s\Lambda_s t)}{1 + \Delta \exp(-2a_s\Lambda_s t)} \right]^2, \tag{84}$$

which can be easily derived using Theorem 3.1 and $\psi_s \propto s^{-\beta}$, $\Lambda_s \propto s^{-\gamma}$ and we also initialize the model such that the initial prediction of the model is equally away from the true strength for different tasks to exclude influence from other aspects, i.e., $\Lambda_s/f_s^0(0)$ are similar for all $s$. As we assume that $\Lambda_s^2 \gg 1$ in Section 4.2, we can rewrite $a_s$ when $N \to \infty$ as

$$a_s \sim Z\bar{Z}s^{-\alpha-\beta-2\gamma} \tag{85}$$

where we use $\bar{Z}$ to denote irrelevant normalization constants.

**Model scaling law.** According to Eq. (64), the model scaling law can be obtained from studying the behavior of

$$L(D) \sim \int_{s=D}^{\infty} Z s^{-\alpha-2\gamma} \left[ \frac{\Delta}{1+\Delta} \right]^2 ds \propto D^{-\alpha-2\gamma+1} \tag{86}$$

where $D$ is the cuttoff of the task such that our model will only learn the first $D-1$ tasks and we let $t \to \infty$.

**Time scaling law.** With a similar procedure as in previous section, we will apply the Laplace method to derive the time scaling law. Specifically,

$$L(t) \propto \int_1^{\infty} \frac{\exp\left[ -\left( 4\tilde{Z}s^{-\alpha-\beta-3\gamma} + \frac{\alpha+2\gamma}{t} \ln s \right) t \right]}{(1 + \Delta e^{-2a_s \Lambda_s t})^2} ds \tag{87}$$

where we use $\tilde{Z}$ to absorb all irrelevant constants. Now we let

$$F(s) = 4\tilde{Z}s^{-\alpha-\beta-3\gamma} + \frac{\alpha+2\gamma}{t} \ln s, \tag{88}$$

then the asymptotic behaviors of $L(t)$ can be written as

$$L(t) \sim e^{-F(c)t} \frac{\sqrt{2\pi}}{\sqrt{F''(c)t}} \tag{89}$$

where $F'(c) = 0$ as before. Note that the first derivative of $F(s)$ is

$$F'(s) = -4\tilde{Z}(\alpha+\beta+3\gamma)s^{-(\alpha+\beta+3\gamma+1)} + \frac{\alpha+2\gamma}{st} \tag{90}$$

$$\implies c := (\tilde{M}t)^{\frac{1}{\alpha+\beta+3\gamma}} = \left( 4\tilde{Z}\frac{\alpha+\beta+3\gamma}{\alpha+2\gamma}t \right)^{\frac{1}{\alpha+\beta+3\gamma}}. \tag{91}$$

Therefore, we obtain that at $s = c$

$$F(c) = \frac{\alpha+2\gamma}{\alpha+\beta+3\gamma} \left( 1 + \ln(\tilde{M}t) \right) t^{-1} \tag{92}$$

$$\implies e^{-F(c)t} = \frac{1}{(e\tilde{M})^{\frac{\alpha+2\gamma}{\alpha+\beta+3\gamma}}} t^{-\frac{\alpha+2\gamma}{\alpha+\beta+3\gamma}}. \tag{93}$$

Furthermore, the second derivative of $F(s)$ with respect to $s$ is

$$F''(s) = \tilde{M}(\alpha+2\gamma)(\alpha+\beta+3\gamma+1)s^{-(\alpha+\beta+3\gamma+2)} - \frac{\alpha+2\gamma}{s^2 t}, \tag{94}$$

which gives us

$$F''(c) = \tilde{M}^{-\frac{2}{\alpha+\beta+3\gamma}} t^{-1-\frac{2}{\alpha+\beta+3\gamma}} (\alpha+2\gamma)(\alpha+\beta+3\gamma). \tag{95}$$

Putting $F(c)$ and $F''(c)$ back to Eq. (87) gives us the time scaling law

$$L(t) \propto t^{-\frac{\alpha+2\gamma-1}{\alpha+\beta+3\gamma}}. \tag{96}$$

**Data Scaling Law.** Similar to previous section, in this case the bottleneck of training is $N$ and we let $t \to \infty$ and $D = \mathcal{N}_s$. According to Eq. (66), the test loss will become

$$L(N) \propto \int_1^{\infty} s^{-\alpha-2\gamma}(1 - Zs^{-\alpha})^N ds$$

$$= \int_1^{\infty} e^{-N((\alpha+2\gamma) \ln s/N - \ln(1-Zs^{-\alpha}))} ds. \tag{97}$$

Following a similar procedure, we let

$$F(s) = (\alpha+2\gamma)\frac{\ln s}{N} - \ln(1 - Zs^{-\alpha}) \tag{98}$$

and we let $F'(c) = 0$. Then the loss function can be written as

$$L(N) \sim e^{-F(c)t} \frac{\sqrt{2\pi}}{\sqrt{F''(c)N}}. \tag{99}$$

To obtain $c$, we need to compute

$$F'(s) = \frac{\alpha + 2\gamma}{sN} - \frac{\alpha Z s^{-\alpha-1}}{1 - Zs^{-\alpha}} \implies c = \left( \frac{Z(N + 1 + 2\gamma/\alpha)}{1 + 2\gamma/\alpha} \right)^{\frac{1}{\alpha}}. \tag{100}$$

Note that we need to make sure

$$Z(N + 1 + 2\gamma/\alpha) > 1 + 2\gamma/\alpha, \tag{101}$$

which can be easily satisfied, to apply the Laplace method. By conducting some algebra, we obtain the following results:

$$F(c) = \frac{\alpha + 2\gamma}{\alpha N} \ln \left( \frac{Z(N + 1 + \frac{2\gamma}{\alpha})}{1 + \frac{2\gamma}{\alpha}} \right) - \ln \left( 1 - \frac{1 + \frac{2\gamma}{\alpha}}{N + 1 + \frac{2\gamma}{\alpha}} \right)$$

$$\implies e^{-F(c)N} \propto \left( \frac{1 + \frac{\gamma}{\alpha}}{NM} \right)^{\frac{\alpha + 2\gamma}{\alpha}}$$

where we use $N \gg 1$ in the second line to study the asymptotic behavior. Furthermore, the second derivative is

$$F''(c) = -c^{-2} \frac{\alpha(N+1)}{N} + \frac{\alpha c^{-2}}{1 - Zc^{-\alpha}} + \frac{\alpha}{c} \frac{Z\alpha c^{-\alpha-1}}{(1 - Zc^{-\alpha})^2} - c^{-2} \frac{2\gamma}{N}$$

$$\sim \alpha c^{-2} \left( \frac{Zc^{-\alpha}}{1 - Zc^{-\alpha}} + \frac{Z\alpha c^{-\alpha}}{(1 - Zc^{-\alpha})^2} - \frac{2\gamma}{\alpha N} \right) \tag{102}$$

which, noting that

$$Zc^{-\alpha} \sim \frac{1 + \frac{2\gamma}{\alpha}}{N} \tag{103}$$

according to the solution of $c$ Eq. (100) gives us

$$F''(c) \propto N^{-\frac{2}{\alpha}-1}. \tag{104}$$

Combining these results gives us the complete data scaling law

$$L(N) \propto N^{-\frac{\alpha+\gamma-1}{\alpha}}. \tag{105}$$

## G NUMERICAL EXPERIMENT DETAILS

For all numerical experiments, we generate the dataset exactly as the process described in Section 2.2. The model structure is a linear self-attention as specified in Section 2.3. If not specified, we set the initialization as

$$\boldsymbol{v}_s(0) = A \times \mathbf{1}_{\mathcal{N}_s+1}, \quad \boldsymbol{w}_s(0) = \boldsymbol{v}_s(0) + 0.1 \times A \times \mathbf{1}_{\mathcal{N}_s+1}. \tag{106}$$

where $A = 0.1$ is a constant and we use $\mathbf{1}_d \in \mathbb{R}^d$ to represent a vector with all elements equal to 1. For the discrete GD training, we set the learning rate as $10^{-3}$ and the number of total optimization steps as 5000. The theoretical prediction using the solution $f_s^0(t)$ is simulated with the forward Euler method such that $t = k\eta$ where $k$ is the optimization step and $\eta$ is the learning rate.

**Fig. 2.** $\mathcal{N}_s = 100$ for both $\psi = 5$ and 100.

**Fig. 3.** $\mathcal{N}_s = 500$. We set the context sequence length as $\psi = 100$, and the task strength $\Lambda = 0.5$.

**Fig. 4.** In this case, we set $\mathcal{N}_s = 500$, and set the initialization as

$$\boldsymbol{v}_s(0) = \boldsymbol{w}_s(0) = A \times \mathbf{1}_{\mathcal{N}_s+1} \tag{107}$$

where $A = 10^{-5}$.

## H GENERALITY OF THE MSFR PROBLEM

In this section, we demonstrate the generality of the MSFR problem. Specifically, we show that our method can also be applied to (or is a limiting case of) other generalized types of the MSFR setup considered in Section 2.1, namely multitask in-context regression under the source-capacity condition (Appendix H.1), MSFR with approximately sparse feature (Appendix H.2), and tasks with idempotent-like $\boldsymbol{H}_s$ (Appendix H.3). We also highlight that our solution can be applied to study other properties of attention besides the neural scaling laws considered in Section 4.1 and 4.2.

### H.1 MULTITASK IN-CONTEXT REGRESSION UNDER THE SOURCE-CAPACITY CONDITION

Interestingly, the MSFR problem can be seen as a limiting case of the multitask version of the in-context regression under the source-capacity condition (Cui et al., 2022), which is defined as follows and can be seen as a generalization of the setup in Lu et al. (2024).

**Multitask in-context regression under the source-capacity condition.** Following the settings of the MSFR problem in Section 2.1, there are $\mathcal{N}_s$ different tasks in total. We let $S$ be the random variable of picking a specific task among $\mathcal{N}_s$ tasks and assume that $S$ follows the power law distribution Eq. (1). In the following, we will show that each task is constructed as an in-context regression, thus we term this setting as multitask in-context regression. We do not use the sparse feature extractor $\phi(s, \boldsymbol{x})$ defined in Eq. 2. Instead, following Cui et al. (2022), we use the feature extractor $\tilde{\phi}(s, \boldsymbol{x}) \in \mathbb{R}^{\mathcal{N}_s}$ such that for data $\boldsymbol{x} \sim \mathcal{P}_X$

$$\boldsymbol{\Sigma}_s = \mathbb{E}_{\boldsymbol{x} \sim \mathcal{P}_X} \left[ \tilde{\phi}(s, \boldsymbol{x}) \tilde{\phi}(s, \boldsymbol{x})^T \right] = \mathrm{diag}\left( \tilde{\boldsymbol{\omega}}^s \right) = \mathrm{diag}\left( \mathscr{P}_s\left( \boldsymbol{\omega} \right) \right) \tag{108}$$

where $\tilde{\boldsymbol{\omega}}^s = \begin{bmatrix} \tilde{\omega}_1^s & \tilde{\omega}_2^s & \cdots & \tilde{\omega}_{\mathcal{N}_s}^s \end{bmatrix}^T \in \mathbb{R}^{\mathcal{N}_s}$ and $\boldsymbol{\omega} = \begin{bmatrix} \omega_1 & \omega_2 & \cdots & \omega_{\mathcal{N}_s} \end{bmatrix}^T \in \mathbb{R}^{\mathcal{N}_s}$ such that $\boldsymbol{\omega}$ satisfies the source/capacity condition

$$\omega_k \propto k^{-\tau}, \tag{109}$$

and $\mathscr{P}_s$ is a simple rearrangement of elements of $\boldsymbol{\omega}$ such that

$$\begin{aligned} \tilde{\omega}_s^s &= \omega_1 \\ \tilde{\omega}_{s+1}^s &= \omega_2 \\ &\vdots \\ \tilde{\omega}_{\mathcal{N}_s}^s &= \omega_{\mathcal{N}_s-s+1} \\ \tilde{\omega}_1^s &= \omega_{\mathcal{N}_s-s+2} \\ &\vdots \\ \tilde{\omega}_{s-1}^s &= \omega_{\mathcal{N}_s} \end{aligned} \tag{110}$$

i.e., the $s$-th eigenvalue of $\boldsymbol{\Sigma}_s$ is the largest given task type $s$. Finally, given task type $s$, we let the strength for task $s$ be $\boldsymbol{\Lambda}_s \in \mathbb{R}^{\mathcal{N}_s}$ and the target $y \in \mathbb{R}$ is

$$y(s, \boldsymbol{x}) = \boldsymbol{\Lambda}_s \cdot \tilde{\phi}(s, \boldsymbol{x}). \tag{111}$$

The in-context regression data $\tilde{\Phi}(s, \mathbf{X})$ is now generated according to the process in Section 2.2:

$$\tilde{\Phi}(s, \mathbf{X}) = \begin{bmatrix} \tilde{\phi}(s, \boldsymbol{x}^{(1)}) & \cdots & \tilde{\phi}(s, \boldsymbol{x}^{(\psi_s)}) & \tilde{\phi}(s, \hat{\boldsymbol{x}}) \\ y(s, \boldsymbol{x}^{(1)}) & \cdots & y(s, \boldsymbol{x}^{(\psi_s)}) & 0 \end{bmatrix} \tag{112}$$

while we now assume that the sequence length $\psi_s$ is fixed for each task $s$.

**MSFR is a limit of multitask in-context regression under source-capacity condition.** We note that, a large $\tau$ in Eq. (109) indicates that the spectrum of the covariance matrix $\boldsymbol{\Sigma}_s$ shows a very fast decay. If $\tau$ is large enough, we can conclude that only the largest eigenvalue of $\boldsymbol{\Sigma}_s$ (i.e., $\tilde{\omega}_s^s$) is significant, thus

$$\boldsymbol{\Sigma}_s = \mathbb{E}_{\boldsymbol{x} \sim \mathcal{P}_X} \left[ \tilde{\boldsymbol{\phi}}(s, \boldsymbol{x}) \tilde{\boldsymbol{\phi}}(s, \boldsymbol{x})^T \right] \xrightarrow{\text{large } \tau} \boldsymbol{\phi}(s, \boldsymbol{x}) \boldsymbol{\phi}(s, \boldsymbol{x})^T, \tag{113}$$

where $\boldsymbol{\phi}(s, \boldsymbol{x})$ is defined in Eq. (2). Since $\mathbb{E}_{\boldsymbol{x} \sim \mathcal{P}_X} \left[ \tilde{\boldsymbol{\phi}}(s, \boldsymbol{x}) \tilde{\boldsymbol{\phi}}(s, \boldsymbol{x})^T \right]$ and $\boldsymbol{\phi}(s, \boldsymbol{x}) \boldsymbol{\phi}(s, \boldsymbol{x})^T$ determine the in-context learning dynamics (because they are the main component of $\boldsymbol{H}_s$ according to Eq. (23) and $\boldsymbol{H}_s$ captures the in-context learning dynamics Eq. (9)), we conclude that the MSFR problem can be seen as a limiting case for the multitask in-context regression under source-capacity condition with large $\tau$. Therefore, we expect that the neural scaling laws derived in the MSFR problem can be generalized to the multitask in-context regression under source-capacity condition.

**Numerical Experiments.** To validate the above claims, we conduct numerical experiments to investigate neural scaling laws of softmax attention and we use the parameterization $\boldsymbol{W}_K^T \boldsymbol{W}_Q$ rather than merging them as a single matrix $\boldsymbol{W}_{KQ}$. For the feature $\tilde{\boldsymbol{\phi}}(s, \boldsymbol{x})$, we use

$$\tilde{\boldsymbol{\phi}}(s, \boldsymbol{x}) \sim \mathcal{N}(0, \boldsymbol{\Sigma}_s) \tag{114}$$

where $\mathcal{N}(0, \boldsymbol{\Sigma}_s)$ is a Gaussian distribution with zero mean and covariance $\boldsymbol{\Sigma}_s = \text{diag}(\tilde{\boldsymbol{\omega}}^s)$ as in Eq. (108). We let $\tau = 3$ in Eq. (109). To make $\sum_{j=1}^{\psi_s} \boldsymbol{\phi}(s, \boldsymbol{x}^{(j)}) \boldsymbol{\phi}(s, \boldsymbol{x}^{(j)})^T / \psi_s$ close enough to $\boldsymbol{\Sigma}_s$, we let $\psi_s = 1000$ for all $s$. The task strength $\boldsymbol{\Lambda}_s \sim \mathcal{N}(0, I)$. The softmax attention is trained by GD with learning rate $2 \times 10^{-1}$. We report the neural scaling laws with respect to time $t$ and with respect to the optimal compute in Fig. 5, where one can clearly see that the softmax self-attention for the multitask in-context regression under the source/capacity condition also displays neural scaling laws similar to those derived in the MSFR problem (Table 1). These numerical experiments support our claims and the generality of the MSFR problem.

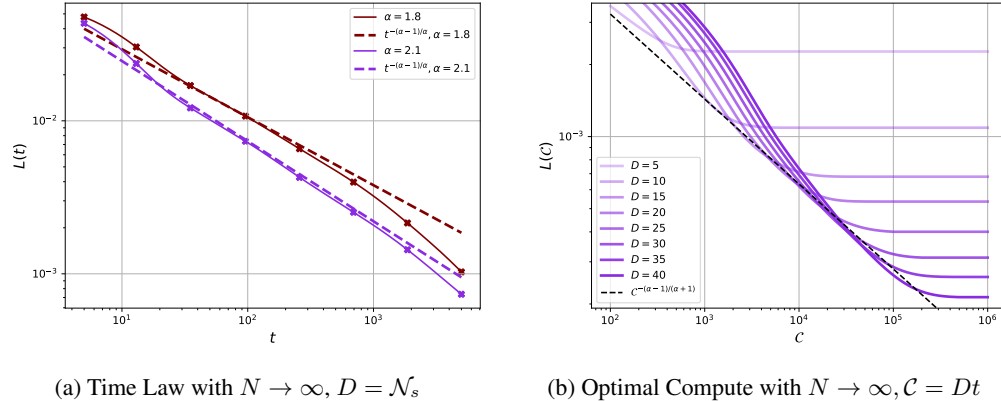

(a) Time Law with $N \to \infty, D = \mathcal{N}_s$       (b) Optimal Compute with $N \to \infty, \mathcal{C} = Dt$

Figure 5: Neural scaling laws for softmax self-attention in the multitask in-context regression under the source-capacity condition. In each figure, we use solid lines to represent empirical simulation results and dashed lines for power law curves. In (b), we set $\alpha = 1.8$.

## H.2 MSFR with Approximately Sparse Feature

In the MSFR problem in Section 2.1, we let the feature extractor be

$$\boldsymbol{\phi}(s, \boldsymbol{x}) : \mathbb{R} \times \mathbb{R}^d \to \{-1, 0, 1\}^{\mathcal{N}_s} \in \mathbb{R}^{\mathcal{N}_s} \tag{115}$$

such that the feature $\boldsymbol{\phi}(s, \boldsymbol{x})$ is sparse. In fact, our solution to the zero-th order of $\epsilon_s$, i.e., $f_s^0(t)$, can still be exact under Assumption 3.1 when the above sparsity condition is relaxed for large sequence length $\psi_s$. We give an example as follows.

**MSFR with approximately sparse feature.** For the MSFR problem in Section 2.1, we now consider a new feature extractor $\tilde{\phi}(s, \boldsymbol{x})$ such that

$$\tilde{\phi}(s, \boldsymbol{x}) = \phi(s, \boldsymbol{x}) + \boldsymbol{\zeta}(s, \boldsymbol{x}), \tag{116}$$

where $\boldsymbol{\zeta}(s, \boldsymbol{x}) \in \mathbb{R}^{\mathcal{N}_s}$ can be a random noise to the first order of $\epsilon_s$ ($\boldsymbol{\zeta}$ does not need to be sparse). We call this task *MSFR with approximately sparse feature*. This task will give us the same set of non-linear ODEs to the zero-th order of $\epsilon_s$ under Assumption 3.1 as that for the original MSFR problem in Section 2.1. Therefore, Theorem 3.1 can still be applied in this case.[1]

**Numerical Experiments.** In Fig. 6, we let $\boldsymbol{\zeta} \sim \mathcal{N}(0, \epsilon_s^2 \boldsymbol{I})$ be a Gaussian noise vector for each task $s$. We compare the loss calculated according to $f_s^0(t)$ in Theorem 3.1 with that obtained from empirical simulation. It can be seen that our theoretical prediction is still highly exact with the existence of the noise vector $\boldsymbol{\zeta}$ when the context sequence length $\psi_s$ is large.

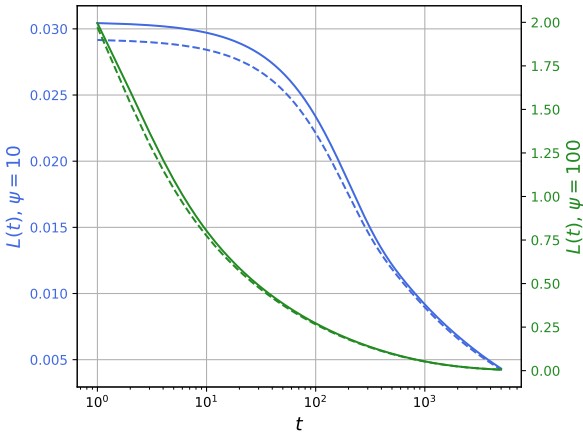

Figure 6: Loss $L(t)$ of MSFR with approximately sparse feature for different context sequence lengths $\psi$(10 and 100) during training. Solid lines are for theoretical predictions while dashed lines are for empirical simulations.

### H.3 GENERAL TASKS WITH IDEMPOTENT-LIKE $H_s$

From a mathematical perspective, besides the MSFR problem considered in this paper, our strategy for solving the ODEs Eq. (9) can be applied to any cases when the matrix $\boldsymbol{H}_s$ in Eq. (9) is idempotent-like without Assumption 3.1:

$$\boldsymbol{H}_s^2 = \mu_s \boldsymbol{H}_s \tag{117}$$

where $\mu_s \in \mathbb{R}$ is a constant. In such cases, the solution of the model prediction is still $f_s^0(t)$ in Theorem 3.1 except for that we now define $a_s = \#_s \psi_s \mu_s / N$ and $f_s^0(t)$ is exact as we do not need Assumption 3.1. We think it will be an interesting future direction to explore other tasks (for self-attention or other machine learning models) where $\boldsymbol{H}_s$ has the idempotent-like structure.

**Numerical Experiments.** To verify the above claim, we consider a simple example where (we omit the subscript $s$ and consider the case where we only have one type of task)

$$\boldsymbol{H} = \sum_{i=1}^{3} E_i \boldsymbol{u}_i \boldsymbol{u}_i^T, \quad \boldsymbol{u}_i \cdot \boldsymbol{u}_j = \delta_{i,j} \tag{118}$$

and we let $E_i = 2$ for $i = 1, 2, 3$, which will give us $\mu = 2$ in Eq. (117). The learning dynamics is Eq. (9) with $\#_s = N$. In Fig. 7, we compare the loss calculated according to the solution in Theorem 3.1 with that obtained from empirical simulation. It can be seen that our theoretical prediction matches with the empirical simulation well because it is an exact solution in this case.

---

[1]We note that our characterization of $f_s^1(t)$ in Appendix J is no longer applicable in this case, and we believe the characterization of $f_s^1(t)$ and the generalization of our methods to more complicate feature extractor $\tilde{\phi}(s, \boldsymbol{x})$ can be an interesting future direction.

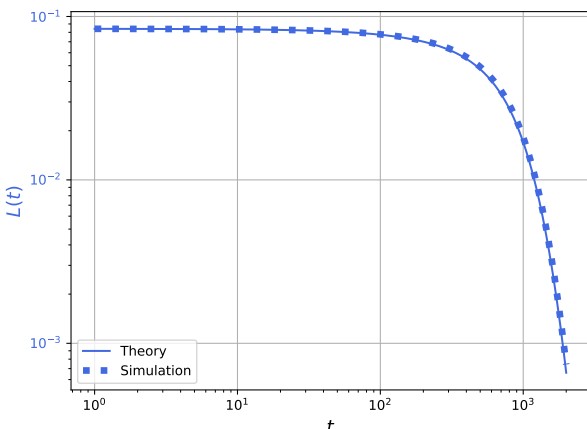

Figure 7: Loss $L(t)$ for learning dynamics Eq. (9) with idempotent-like $\boldsymbol{H}$.

# I  ADDITIONAL NUMERICAL EXPERIMENTS

In this section, we conduct additional experiments to explore the generality of our conclusion for the neural scaling laws. In particular, in Appendix I.1, we explore the neural scaling laws of softmax self-attention for the MSFR problem where we train the model with GD, while in Appendix I.2 we train the model with AdamW.

## I.1  NEURAL SCALING LAWS OF SOFTMAX SELF-ATTENTION FOR MSFR PROBLEM

We replace the linear self-attention with the softmax self-attention in the numerical experiments of Fig. 3 and Fig. 4 to investigate the neural scaling laws for the MSFR problem. For completeness, we adopt the $\boldsymbol{W}_K^T \boldsymbol{W}_Q$ decomposition rather than a single merged $\boldsymbol{W}_{KQ}$, i.e., $f(\boldsymbol{G}; \boldsymbol{\theta}) = \boldsymbol{V} \boldsymbol{G} \operatorname{softmax}\left[\boldsymbol{G}^T \boldsymbol{W}_K^T \boldsymbol{W}_Q \boldsymbol{G}\right]$. All the other settings are the same as those of Section 4.1 and 4.2.

**Fixed Context Sequence Length.**  In Fig. 8, we report the neural scaling laws when the context sequence length is fixed as in Section 4.1. It can be seen that the scaling laws with respect to time $t$, model size $D$, data size $N$, and the optimal compute $\mathcal{C}$ are similar to those reported in Table 1.

**Varied Context Sequence Length.**  For the varied context sequence length, we let $\psi_s = \mathcal{F}(s) \propto s^{-\beta}$ as in Section 4.2 while we keep $\Lambda_s$ fixed. We note that the neural scaling laws with respect to the model size $D$ and data size $N$ are not affected by a varied context sequence length as reflected in Table 2, which is due to the fact that GD can still learn the task strength $\Lambda_s$ for the task $s$ as $t \to \infty$ when the context sequence length is varied. We report the scaling laws with respect to time $t$ in Fig. 9, where we can see that the softmax self-attention still admits a similar time scaling law compared to the linear self-attention for varied context sequence length. As a result, the optimal compute scaling law of softmax self-attention will also be similar to that of linear self-attention, as it is a consequence of the time scaling law and model size scaling law and these laws do not change.

These numerical experiments reveal that our claims regarding neural scaling laws for the linear self-attention can be generalized to the softmax self-attention.

## I.2  NEURAL SCALING LAWS OF SOFTMAX SELF-ATTENTION TRAINED BY ADAMW

To examine the effects of optimization algorithms on neural scaling laws in the MSFR problem, we train softmax self-attention with AdamW and we also use the $\boldsymbol{W}_K^T \boldsymbol{W}_Q$ parameterization. We focus on the case where the context sequence length and the task strength are fixed. We present our parameters in the following table.

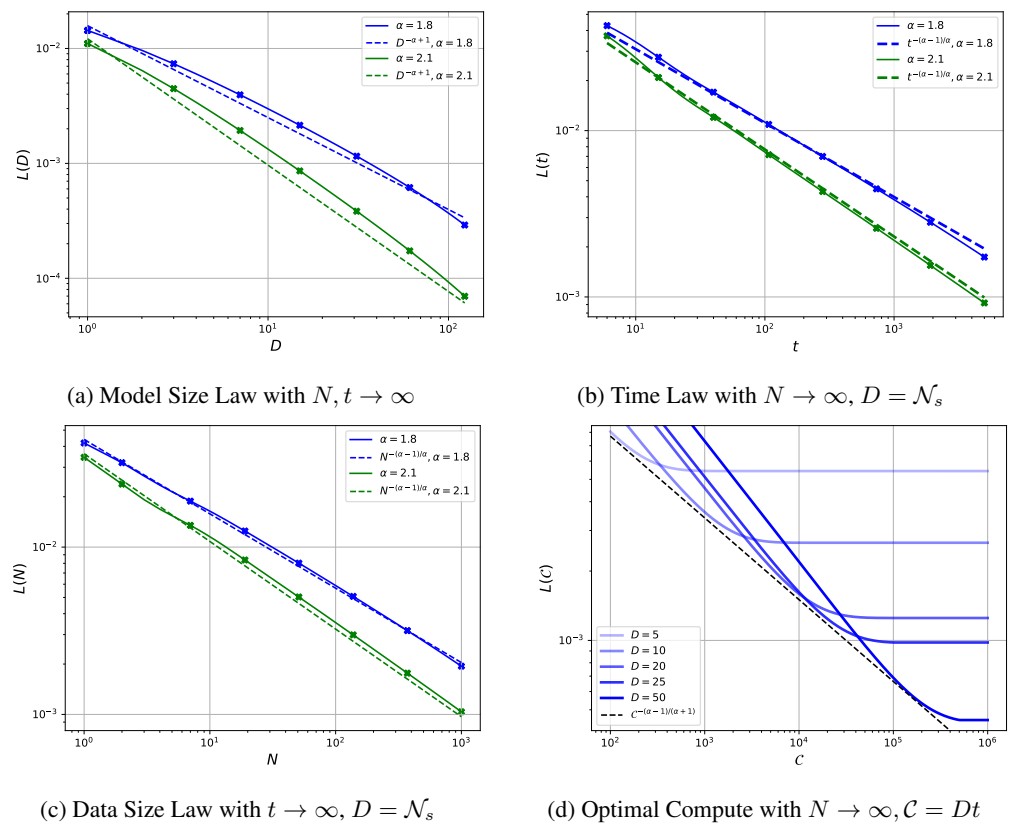

(a) Model Size Law with $N, t \to \infty$

(b) Time Law with $N \to \infty, D = \mathcal{N}_s$

(c) Data Size Law with $t \to \infty, D = \mathcal{N}_s$

(d) Optimal Compute with $N \to \infty, \mathcal{C} = Dt$

Figure 8: Neural scaling laws for softmax self-attention trained by GD with different values of $\alpha = 1.8, 2.1$ when the context sequence length is fixed. In each figure, we use solid lines to represent empirical simulation results and dashed lines for power law curves. In (d), we set $\alpha = 1.8$.

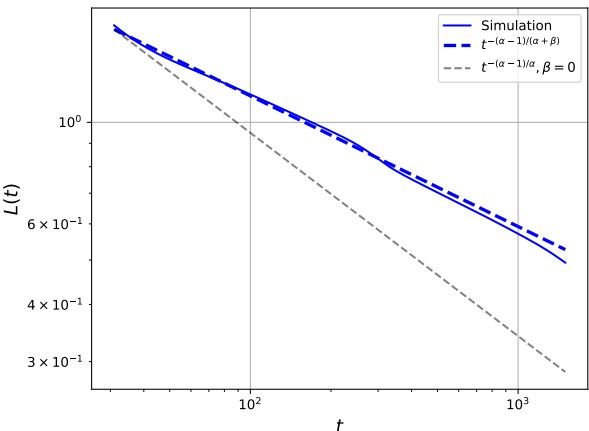

Figure 9: Neural scaling laws with respect to time $t$ for softmax self-attention trained by GD with $\alpha = 1.8$ when the context sequence length is fixed. We let $N \to \infty, D = \mathcal{N}_s$. Solid lines represent empirical simulation results while dashed lines represent power law curves obtained from Table 2.

**Neural scaling laws with respect to model size $D$ and data size $N$.** We expect that AdamW will show similar neural scaling laws with respect to the model size $D$ and data size $N$ when compared to GD. This is because AdamW can still learn the task strength $\Lambda_s$ for the task $s$ given sufficient training time $t$ (Fig. 10c), which is similar to GD. We report the corresponding neural scaling laws

| learning rate $\eta$ | $5 \times 10^{-3}$ |
|---|---|
| $\beta_1$ | 0.9 |
| $\beta_2$ | 0.999 |
| weight decay | $10^{-5}$ |
| eps | $10^{-8}$ |

Table 4: Parameters for AdamW

in Fig. 10a and 10b, where it can be seen that the softmax self-attention trained by AdamW still admits similar neural scaling laws with respect to $D$ and $N$.

**Neural scaling law with respect to time $t$.** However, AdamW typically exhibits a very different dynamics during training compared to GD, as AdamW has a very different learning dynamics (e.g., it converges faster than GD). Thus we expect that AdamW will lead to a very different time scaling law (Fig. 10c), which will further lead to a different neural scaling law for the optimal compute (Fig. 10d). We additionally note that these observations are similar to the observations in Hoffmann et al. (2022), where the authors revealed that, when compared to Adam, AdamW shows a different test loss behavior against the optimization steps (training time), indicating that the type of optimization algorithm can affect the time scaling laws.

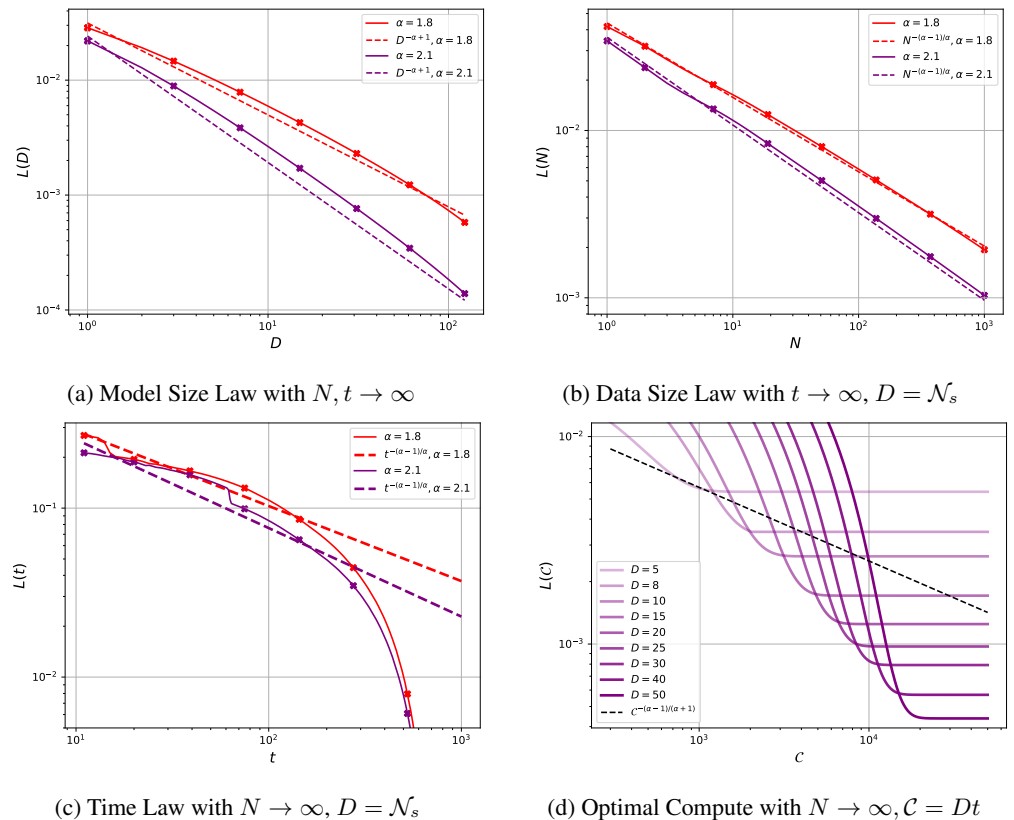

(a) Model Size Law with $N, t \to \infty$

(b) Data Size Law with $t \to \infty, D = \mathcal{N}_s$

(c) Time Law with $N \to \infty, D = \mathcal{N}_s$

(d) Optimal Compute with $N \to \infty, \mathcal{C} = Dt$

Figure 10: Neural scaling laws for softmax self-attention trained by AdamW with different values of $\alpha = 1.8, 2.1$. In each figure, we use solid lines to represent empirical simulation results and dashed lines for power law curves that are obtained Table 1 (when the self-attention is trained by GD). In (d), we set $\alpha = 2.1$.

## J  COMPLETE SOLUTION OF SELF-ATTENTION UP TO THE FIRST ORDER

We first discuss how to derive the solution of model parameters $\boldsymbol{v}_s^0(t)$ and $\boldsymbol{w}_s^0(t)$ to the zero-th order of $\epsilon_s$, then present the solution to the first order of $\epsilon_s$, which can give us the complete solution of self-attention up to the first order of $\epsilon_s$. In the following sections, we omit the subscript $s$ for convenience and recover it in the final solution.

### J.1  ZERO-TH ORDER SOLUTION OF MODEL PARAMETERS

E.2 gives us the solution of the model output $f_s^0(t)$. To obtain the forms of $\boldsymbol{v}^0(t)$ and $\boldsymbol{w}^0(t)$, we only need to solve $\boldsymbol{\eta}^0$ and $\boldsymbol{\rho}^0$ since

$$\boldsymbol{v}^0(t) = \frac{\boldsymbol{\eta}^0 + \boldsymbol{\rho}^0}{2}, \quad \boldsymbol{w}^0(t) = \frac{\boldsymbol{\eta}^0 - \boldsymbol{\rho}^0}{2}$$

according to their definitions. The ODEs of $\boldsymbol{\eta}^0$ and $\boldsymbol{\rho}^0$ Eq. (31) can be rewritten using the components (note that $\eta_s^0$ is the $s$-th component of $\boldsymbol{\eta}^0$) as:

$$
\begin{aligned}
\dot{\eta}_s^0 &= -\frac{\psi\#}{4N}\left(g^0 - h^0 - 4\Lambda\right)\left(\eta_s^0 + \Lambda\eta_{\mathcal{N}_s+1}^0\right) \\
\dot{\eta}_{\mathcal{N}_s+1}^0 &= -\Lambda\frac{\psi\#}{4N}\left(g^0 - h^0 - 4\Lambda\right)\left(\eta_s^0 + \Lambda\eta_{\mathcal{N}_s+1}^0\right) \\
\dot{\rho}_s^0 &= \frac{\psi\#}{4N}\left(g^0 - h^0 - 4\Lambda\right)\left(\rho_s^0 + \Lambda\rho_{\mathcal{N}_s+1}^0\right) \\
\dot{\rho}_{\mathcal{N}_s+1}^0 &= \Lambda\frac{\psi\#}{4N}\left(g^0 - h^0 - 4\Lambda\right)\left(\rho_s^0 + \Lambda\rho_{\mathcal{N}_s+1}^0\right).
\end{aligned}
\tag{119}
$$

An interesting property of these ODEs is that

$$\frac{d}{dt}\left(\Lambda\eta_s^0 - \eta_{\mathcal{N}_s+1}^0\right) = 0 \implies \Lambda\eta_s^0 - \eta_{\mathcal{N}_s+1}^0 = \bar{C}, \quad \Lambda\rho_s^0 - \rho_{\mathcal{N}_s+1}^0 = \tilde{C}, \tag{120}$$

which can gives us a relation between $\eta_s^0$ and $\eta_{\mathcal{N}_s+1}^0$ and a similar one between $\rho_s^0$ and $\rho_{\mathcal{N}_s+1}^0$. On the other hand, since we already know the solution of $g^0$ from E.2 and $g^0$ can be written as

$$g^0 = (\eta_s^0 + \Lambda\eta_{\mathcal{N}_s+1}^0)^2, \tag{121}$$

we can solve $\boldsymbol{\eta}^0$ and $\boldsymbol{\rho}^0$ based on these relations, which will also give us $\boldsymbol{v}^0$ and $\boldsymbol{w}^0$.

### J.2  SOLUTION UP TO THE FIRST ORDER

We now discuss the solution to the first-order of $\epsilon_s$. According to the definition of $g_s^1$ and $h_s^1$ in Eq. (28) and (29), they can be rewritten as

$$
\begin{aligned}
g^1 &= \psi\boldsymbol{\eta}^0 \cdot \boldsymbol{H}^1\boldsymbol{\eta}^0 + 2m \\
h^1 &= \psi\boldsymbol{\rho}^0 \cdot \boldsymbol{H}^1\boldsymbol{\rho}^0 + 2n
\end{aligned}
\tag{122}
$$

where $m$ and $n$ follow the dynamics Eq. (35) and Eq. (36). Therefore, to obtain the formulations of $g^1$ and $h^1$, the ODEs of $m$ and $n$ will be the only equations that need to be solved since we can obtain $\boldsymbol{\eta}^0 \cdot \boldsymbol{H}^1\boldsymbol{\eta}^0$ directly from J.1. In the following, we focus on how to solve $m$ and $n$. For convenience, we first present ODEs for $m$ and $n$ ( Eq. (35) and Eq. (36) without the subscript $s$)

$$
\begin{aligned}
\dot{m} &= -\left(g^0 - h^0 - 4\Lambda\right)\left[\frac{am}{2} + \frac{\psi^2\#}{4N}\boldsymbol{\eta}^0 \cdot \boldsymbol{H}^0\boldsymbol{H}^1\boldsymbol{\eta}^0\right] - (g^1 - h^1)\frac{ag^0}{4}, \\
\dot{n} &= \left(g^0 - h^0 - 4\Lambda\right)\left[\frac{an}{2} + \frac{\psi^2\#}{4N}\boldsymbol{\rho}^0 \cdot \boldsymbol{H}^0\boldsymbol{H}^1\boldsymbol{\rho}^0\right] + (g^1 - h^1)\frac{ah^0}{4}.
\end{aligned}
\tag{123}
$$

The above equations are too complex, thus we attempt to reformulate them to simpler forms: we can obtain a new set of ODEs from the above equations

$$\frac{d}{dt}mh^0 = \dot{h}^0 m + \dot{m}h^0$$

$$= -(g^0 - h^0 - 4\Lambda)\frac{\#\psi^2}{4N}\boldsymbol{\eta}^0 \cdot \boldsymbol{H}^0 \boldsymbol{H}^1 \boldsymbol{\eta}^0 h^0 - (g^1 - h^1)\frac{ag^0 h^0}{4} \quad (124)$$

$$\frac{d}{dt}ng^0 = \dot{g}^0 n + \dot{n}g^0$$

$$= (g^0 - h^0 - 4\Lambda)\frac{\#\psi^2}{4N}\boldsymbol{\rho}^0 \cdot \boldsymbol{H}^0 \boldsymbol{H}^1 \boldsymbol{\rho}^0 g^0 + (g^1 - h^1)\frac{ag^0 h^0}{4}, \quad (125)$$

which implies that

$$\frac{d}{dt}(mh^0 + ng^0) = -(g^0 - h^0 - 4\Lambda)\frac{\#\psi^2}{4N}\left(\boldsymbol{\eta}^0 \cdot \boldsymbol{H}^0 \boldsymbol{H}^1 \boldsymbol{\eta}^0 h^0 - \boldsymbol{\rho}^0 \cdot \boldsymbol{H}^0 \boldsymbol{H}^1 \boldsymbol{\rho}^0 g^0\right). \quad (126)$$

The above equation gives us a relation between $mh^0$ and $ng^0$. Fortunately, according to the definitions of $g^0$, $h^0$, $\boldsymbol{H}^0$, and $\boldsymbol{H}^1$ (Eq. (7)), we can expand the terms inside the second bracket of the above equation:

$$\boldsymbol{\eta}^0 \cdot \boldsymbol{H}^0 \boldsymbol{H}^1 \boldsymbol{\eta}^0 h^0 - \boldsymbol{\rho}^0 \cdot \boldsymbol{H}^0 \boldsymbol{H}^1 \boldsymbol{\rho}^0 g^0 = \sqrt{g^0 h^0}\left[\eta_s^0(\rho_s^0 + \Lambda\rho_{\mathcal{N}_s+1}^0) - (\eta_s^0 + \Lambda\eta_{\mathcal{N}_s+1}^0)\rho_s^0\right]$$

$$= \sqrt{2C}(\bar{C}\rho_{\mathcal{N}_s+1}^0 - \tilde{C}\eta_{\mathcal{N}_s+1}^0) \quad (127)$$

where we use Eq. (120) in the second equality. If the model is initialized as $\bar{C} = \tilde{C} = 0$, then under this condition, we can immediately conclude that

$$\frac{d}{dt}(mh^0 + ng^0) = 0 \implies \forall t \geq 0 : mh^0 = -ng^0 + \hat{C}, \quad (128)$$

where $\hat{C}$ is determined by the initial condition and we let $\hat{C} = 0$ in the following. Noting that

$$g^1 - h^1 := r + 2(m - n) = \psi(\boldsymbol{\eta}^0 \cdot \boldsymbol{H}^1 \boldsymbol{\eta}^0 - \boldsymbol{\rho}^0 \cdot \boldsymbol{H}^1 \boldsymbol{\rho}^0) + 2(m - n), \quad (129)$$

Eq. (128) allows us to simplify the ODEs for $m$ and $h$ further by interchangeably using $mh^0$ and $-ng^0$:

$$\dot{m} = -\frac{am}{2}\left(g^0 - h^0 - 4\Lambda + g^0 - \frac{ng^0}{m}\right) - \frac{ag^0 r}{4} - \frac{\psi^2 \#}{4N}\left(g^0 - h^0 - 4\Lambda\right)\boldsymbol{\eta}^0 \cdot \boldsymbol{H}^0 \boldsymbol{H}^1 \boldsymbol{\eta}^0$$

$$= -am\left(g^0 - 2\Lambda\right) - \frac{ag^0 r}{4} - \frac{\psi^2 \#}{4N}\left(g^0 - h^0 - 4\Lambda\right)\boldsymbol{\eta}^0 \cdot \boldsymbol{H}^0 \boldsymbol{H}^1 \boldsymbol{\eta}^0 \quad (130)$$

$$\dot{n} = -an\left(h^0 + 2\Lambda\right) + \frac{ah^0 r}{4} + \frac{\psi^2 \#}{4N}(g^0 - h^0 - 4\Lambda)\boldsymbol{\rho}^0 \cdot \boldsymbol{H}^0 \boldsymbol{H}^1 \boldsymbol{\rho}^0. \quad (131)$$

Eq. (130) and (131) are exactly solvable since they are simply first order linear ODEs. Specifically, let

$$\varrho(t) = -a\left(g^0 - 2\Lambda\right),$$

$$\vartheta(t) = -\frac{ag^0 r}{4} - \frac{\psi^2 \#}{4N}\left(g^0 - h^0 - 4\Lambda\right)\boldsymbol{\eta}^0 \cdot \boldsymbol{H}^0 \boldsymbol{H}^1 \boldsymbol{\eta}^0 \quad (132)$$

then Eq. (130) can be rewritten as

$$\dot{m} = \varrho(t)m + \vartheta(t). \quad (133)$$

The standard procedure for solving this is letting $\dot{u} = -u\varrho$ and multiplying $u$ to both sides of Eq. (133), then we obtain

$$\frac{d}{dt}um = u\vartheta \implies um = \int u(t)\vartheta(t)dt + \text{const}. \implies m = \frac{\int u(t)\vartheta(t)dt + \text{const}.}{u}. \quad (134)$$

Similarly, to solve $n$, we let

$$\varepsilon(t) = a\left(h^0 + 2\Lambda\right)$$

$$\varphi(t) = \frac{ah^0 r}{4} + \frac{\psi^2 \#}{4N}(g^0 - h^0 - 4\Lambda)\boldsymbol{\rho}^0 \cdot \boldsymbol{H}^0 \boldsymbol{H}^1 \boldsymbol{\rho}^0 \quad (135)$$

$$\dot{z} = z\varepsilon(t)$$

then

$$n = \frac{\int z(t)\varphi(t)dt + \text{const.}}{z}. \tag{136}$$

As a result, the solution of self-attention up to the first order of $\epsilon_s$ for the task type $s$ under Assumption 3.1 now becomes

**Solution:** $f = \dfrac{g^0 - h^0}{4}$

$$+ \epsilon \left[ r + 2 \left( \frac{\int u(\tau)\vartheta(\tau)d\tau + \text{const.}}{u} - \frac{\int z(\tau)\varphi(\tau)d\tau + \text{const.}}{z} \right) \right]. \tag{137}$$

### J.3  FORMULATION OF THE FIRST ORDER SOLUTION

We examine each term in Eq. (137) in the following. Since we let $\bar{C} = \tilde{C} = 0$ in Eq. (120), $g^0$ and $h^0$ can be written explicitly as

$$g^0 = \psi(\Lambda^2 + 1)^2(\eta_s^0)^2, \quad h^0 = \psi(\Lambda^2 + 1)^2(\rho_s^0)^2 \tag{138}$$

**Form of $r$.**  Note that $r$ is defined as

$$r = \psi(\boldsymbol{\eta}^0 \cdot \boldsymbol{H}^1 \boldsymbol{\eta}^0 - \boldsymbol{\rho}^0 \cdot \boldsymbol{H}^1 \boldsymbol{\rho}^0) = \psi \left[ (\eta_s^0)^2 - (\rho_s^0)^2 \right]$$
$$= \frac{g^0 - h^0}{(\Lambda^2 + 1)^2} \tag{139}$$

**Form of $u$ and $\int u(\tau)\vartheta(\tau)d\tau$.**  We first derive $u$:

$$\dot{u} = -u\varrho \implies u = \exp\left( -\int \varrho(t)dt \right) \tag{140}$$

where

$$-\int \varrho(t)dt = \int a(g^0 - 2\Lambda)dt = -2a\Lambda t + \int 2\frac{\dot{\gamma}}{\gamma}dt$$
$$= -2a\Lambda t + 2\ln\gamma \tag{141}$$

where we use Eq. (48) in the second line. Putting the above integral back to the expression of $u$, we obtain

$$u = (e^{-a\Lambda t}\gamma)^2 = \left( Ae^{\xi t} + Be^{-\xi t} \right)^2, \tag{142}$$

where $\xi = \sqrt{4\Lambda^2 a^2 + 2a^2 C}/2$ is defined in Eq. (46). We now derive $\int u\vartheta dt$. To start, we examine each term of $\vartheta$ first:

$$-\frac{ag^0 r}{4} = -\frac{a\left( (g^0)^2 - 2C \right)}{4(\Lambda^2 + 1)^2}$$

$$-\frac{\psi^2 \#}{4N} \left( g^0 - h^0 - 4\Lambda \right) \boldsymbol{\eta}^0 \cdot \boldsymbol{H}^0 \boldsymbol{H}^1 \boldsymbol{\eta}^0 = -\frac{\psi \#}{4(\Lambda^2 + 1)N} \left( g^0 - h^0 - 4\Lambda \right) g^0$$
$$= -\frac{a\left( (g^0)^2 - 2C - 4\Lambda g^0 \right)}{4(\Lambda^2 + 1)^2} \tag{143}$$
$$\implies \vartheta = -\frac{a\left( (g^0)^2 - 2C - 2\Lambda g^0 \right)}{2(\Lambda^2 + 1)^2}.$$

Using this in the integral and considering the form of $u$ in Eq. (142), we have

$$\int u\vartheta dt = -\frac{a}{2(\Lambda^2 + 1)^2} \int (Ae^{\xi t} + Be^{-\xi t})^2 \left[ (g^0)^2 - 2C - 2\Lambda g^0 \right] dt$$
$$= -\frac{2}{a(\Lambda^2 + 1)^2} \int \left( A\sigma_+ e^{\xi t} + B\sigma_- e^{-\xi t} \right)^2 dt + \frac{aC}{(\Lambda^2 + 1)^2} \int (Ae^{\xi t} + Be^{-\xi t})^2 dt$$
$$+ \frac{2\Lambda}{(\Lambda^2 + 1)^2} \int (Ae^{\xi t} + Be^{-\xi t}) \left( A\sigma_+ e^{\xi t} + B\sigma_- e^{-\xi t} \right) dt \tag{144}$$

where we frequently use the solution of $g^0$ in Eq. (33).

**Form of $z$ and $\int z(\tau)\varphi(\tau)d\tau$.** By the similar procedure of deriving $u$ and $\int u\vartheta dt$, we can also derive $z$ and $\int z\varphi dt$.

$$z = \exp\left(\int \varepsilon dt\right) = (e^{a\Lambda t}\theta)^2 = \left(Ee^{\xi t} + Fe^{-\xi t}\right)^2 \tag{145}$$

where we use Eq. (48) in the second equality. Similar to the derivation of $\vartheta$, we can derive $\varphi$ as follows:

$$\frac{ah^0 r}{4} = -\frac{a\left((h^0)^2 - 2C\right)}{4(\Lambda^2 + 1)^2}$$

$$\frac{\psi^2 \#}{4N}\left(g^0 - h^0 - 4\Lambda\right)\boldsymbol{\rho}^0 \cdot \boldsymbol{H}^0\boldsymbol{H}^1\boldsymbol{\rho}^0 = \frac{\psi\#}{4(\Lambda^2 + 1)N}\left(g^0 - h^0 - 4\Lambda\right)h^0$$

$$= -\frac{a\left((h^0)^2 - 2C + 4\Lambda h^0\right)}{4(\Lambda^2 + 1)^2} \tag{146}$$

$$\implies \varphi = -\frac{a\left((h^0)^2 - 2C + 2\Lambda h^0\right)}{2(\Lambda^2 + 1)^2}$$

thus

$$\int z\varphi dt = -\frac{a}{2(\Lambda^2 + 1)^2}\int (Ee^{\xi t} + Fe^{-\xi t})^2 \left[(h^0)^2 - 2C + 2\Lambda h^0\right] dt$$

$$= -\frac{2}{a(\Lambda^2 + 1)^2}\int \left(E\sigma_- e^{\xi t} + F\sigma_+ e^{-\xi t}\right)^2 dt + \frac{aC}{(\Lambda^2 + 1)^2}\int (Ee^{\xi t} + Fe^{-\xi t})^2 dt$$

$$+ \frac{2\Lambda}{(\Lambda^2 + 1)^2}\int (Ee^{\xi t} + Fe^{-\xi t})\left(E\sigma_- e^{\xi t} + F\sigma_+ e^{-\xi t}\right) dt, \tag{147}$$

where we frequently use the solution of $h^0$ in Eq. (33).

**Results of integrals.** It is now left for us to solve all the integrals to obtain the complete solution. We list the results below.

1.
$$\int (Ae^{\xi t} + Be^{-\xi t})(\sigma_+ Ae^{\xi t} + B\sigma_- e^{-\xi t})dt = \frac{A^2\sigma_+}{2\xi}e^{2\xi t} - \frac{B^2\sigma_-}{2\xi}e^{-2\xi t} + 2AB\Lambda at$$

$$\int (Ee^{\xi t} + Fe^{-\xi t})(E\sigma_- e^{\xi t} + F\sigma_+ e^{-\xi t})dt = \frac{E^2\sigma_-}{2\xi}e^{2\xi t} - \frac{F^2\sigma_+}{2\xi}e^{-2\xi t} + 2EF\Lambda at$$

2.
$$\int (\sigma_+ Ae^{\xi t} + B\sigma_- e^{-\xi t})^2 dt = \frac{A^2\sigma_+^2}{2\xi}e^{2\xi t} - \frac{B^2\sigma_-^2}{2\xi}e^{-2\xi t} + 2\sigma_+\sigma_- ABt$$

$$\int (\sigma_- Ee^{\xi t} + \sigma_+ Fe^{-\xi t})^2 dt = \frac{E^2\sigma_-^2}{2\xi}e^{2\xi t} - \frac{F^2\sigma_+^2}{2\xi}e^{-2\xi t} + 2\sigma_+\sigma_- EFt$$

3.
$$\int (Ae^{\xi t} + Be^{-\xi t})^2 dt = \frac{A^2}{2\xi}e^{2\xi t} - \frac{B^2}{2\xi}e^{-2\xi t} + 2ABt$$

$$\int (Ee^{\xi t} + Fe^{-\xi t})^2 dt = \frac{E^2}{2\xi}e^{2\xi t} - \frac{F^2}{2\xi}e^{-2\xi t} + 2EFt.$$

**Complete solution.** With these integrals, we are now ready to find the explicit forms of the solution Eq. (137). In particular, we have

$$\int u\vartheta dt = \frac{A^2 e^{2\xi t}}{\xi(\Lambda^2 + 1)^2 a}\left(-\sigma_+^2 + \frac{a^2 C}{2} + a\Lambda\sigma_+\right) + \frac{B^2 e^{-2\xi t}}{\xi(\Lambda^2 + 1)^2 a}\left(\sigma_-^2 - \frac{a^2 C}{2} - a\Lambda\sigma_-\right)$$

$$+ \frac{ABt}{(\Lambda^2 + 1)^2 a}\left(4\Lambda^2 a^2 + 2a^2 C - 4\sigma_+\sigma_-\right)$$

$$= -\frac{A^2 e^{2\xi t}\Lambda}{(\Lambda^2 + 1)^2}\left(1 + \frac{\Lambda a}{\xi}\right) + \frac{B^2 e^{-2\xi t}\Lambda}{(\Lambda^2 + 1)^2}\left(-1 + \frac{\Lambda a}{\xi}\right) + \frac{4aABt}{(\Lambda^2 + 1)^2}\left(\Lambda^2 + C\right) \tag{148}$$

and

$$\int z\varphi dt = \frac{E^2 e^{2\xi t}}{\xi(\Lambda^2+1)^2 a}\left(-\sigma_-^2 + \frac{a^2 C}{2} + a\Lambda\sigma_-\right) + \frac{F^2 e^{-2\xi t}}{\xi(\Lambda^2+1)^2 a}\left(\sigma_+^2 - \frac{a^2 C}{2} - a\Lambda\sigma_+\right)$$
$$+ \frac{EFt}{(\Lambda^2+1)^2 a}\left(4\Lambda^2 a^2 + 2a^2 C - 4\sigma_+\sigma_-\right)$$
$$= \frac{E^2 e^{2\xi t}\Lambda}{(\Lambda^2+1)^2}\left(1 - \frac{\Lambda a}{\xi}\right) - \frac{F^2 e^{-2\xi t}\Lambda}{(\Lambda^2+1)^2}\left(1 + \frac{\Lambda a}{\xi}\right) + \frac{4aEFt}{(\Lambda^2+1)^2}\left(\Lambda^2 + C\right). \quad (149)$$

These equations are sufficient for us to find $m$ and $n$. For $m$ we have

$$m = \frac{\int u\vartheta dt}{u} = \frac{-Q^2 e^{2\xi t}\Lambda\left(1 + \frac{\Lambda a}{\xi}\right) + e^{-2\xi t}\Lambda\left(-1 + \frac{\Lambda a}{\xi}\right) + 4aQt\left(\Lambda^2 + C\right)}{(\Lambda^2+1)^2(Qe^{\xi t} + e^{-\xi t})^2} \quad (150)$$

where $Q$ has already be determined in Theorem 3.1. For $n$ we have

$$n = \frac{\int z\varphi dt}{z} = \frac{P^2 e^{2\xi t}\Lambda\left(1 - \frac{\Lambda a}{\xi}\right) - e^{-2\xi t}\Lambda\left(1 + \frac{\Lambda a}{\xi}\right) + 4aPt\left(\Lambda^2 + C\right)}{(\Lambda^2+1)^2(Pe^{\xi t} + e^{-\xi t})^2} \quad (151)$$

where $P$ has already be determined in Theorem 3.1. Finally, by using the solved $m$ and $n$ above and $r$ (Eq. (139)) in Eq. (137) and recovering the subscript $s$ in all relevant terms, we obtain the complete solution of self-attention up to the first order of $\epsilon$ under Assumption 3.1. Note that as $t \to \infty$, we can easily verify that in Eq. (129) $m - n = -2\Lambda/(\Lambda^2+1)^2$ and $r = 4\Lambda/(\Lambda^2+1)^2$, thus $g^1 - h^1 = 0$. As a result, $f_s^0(t) + \epsilon_s f_s^1(t) = \Lambda_s$ as desired.

