# OpenReview forum: "A Solvable Attention for Neural Scaling Laws"
_ICLR.cc/2025/Conference — ICLR 2025 Poster_

### Official Review · Reviewer_9RQV · 2024-10-28

**Soundness:** 3
**Presentation:** 2
**Contribution:** 3
**Rating:** 8
**Confidence:** 4

**Summary:**

This manuscript studies the dynamics of learning a particular structured in-context regression task with linear attention. Its main results are approximate exponents for the decay of error with time, data, and model size.

**Edit following author response: I have raised my score from 5 to 8**

**Strengths:**

I found this to be a generally interesting paper, and its topic should be of broad interest. I do have some concerns that preclude my recommending publication in its current form, but I think these should be addressable.

**Weaknesses:**

Here I list some concerns; their order should not be ascribed any meaning.

- The title is inaccurate and vague, since the authors are not the first to study linear attention, and their solutions are approximate. I would suggest switching to something more descriptive, and would strongly suggest mentioning "linear attention" in the title. This critique extends to the abstract, which is similarly vague.

- In that vein, please refer to "linear attention" or "softmax attention" rather than calling everything "self-attention"; this will make the paper less confusing to read.

- The review of models for neural scaling laws in Lines 44-51 is inadequate. The authors should cite the original work on source-capacity conditions in kernel regression by Caponnetto and de Vito (from 2007!), and more of the substantial literature on scaling laws in random feature models that precedes the paper of Maloney et al. Suitable references are reviewed, for instance, in the cited work of Bordelon et al. or in the expository work of Atanasov et al. (https://arxiv.org/abs/2405.00592). The authors should also cite Paquette et al (https://arxiv.org/abs/2405.15074), which is contemporaneous with Bordelon et al 2024.

- The authors don't discuss the growing literature on in-context learning of linear regression, which is closely related to their analysis. First, a more extensive comparison to Zhang et al (which I mention is now published as https://jmlr.org/papers/v25/23-1042.html) is required. Next, it could be interesting to compare the results presented here to those of Lu et al https://arxiv.org/abs/2405.11751, who studied the asymptotic behavior of linear attention trained using ridge regression, i.e., to convergence.

- The paper would be much improved if the authors could relate their data model to other tasks and to natural data. At the end of the day, you're doing gradient flow on the MSE, so at least in certain limits you should be able to replace the specific data model with a Gaussian covariate model of matched moments. Would you get the same scaling laws if you studied in-context regression under source-capacity conditions, i.e., the generalization of the task studied in Lu et al to structured covariates and targets?

- The main analytical results of the paper are approximations, and those approximations aren't clearly organized. For instance, the statement of Theorem 3.1 references Assumption 3.1 rather than directly stating that its result is an expansion at large $\psi_s$. It would be clearer to state this directly. Also, though the authors provide reasonably compelling numerical evidence that their approximations produce reasonable results, it would be better to make this mathematically precise, i.e., to give explicit error bounds. Otherwise, please state your results as Results rather than Theorems.

**Questions:**

- The discussion in lines 64-71 is not very accurate, as the cited paper of Bordelon et al studies scaling laws for *random feature models*, not neural networks that can learn features.

- Why do you bother to write (6) in terms of a generic loss $\ell$ rather than just writing the MSE, which is used throughout? This creates unnecessary overhead for the reader.

- Unless I'm mistaken, all experiments use linear attention. The claims regarding architecture (in)-dependence of certain scalings would be much stronger if you could add experiments with softmax attention.

---

> ### Author Response · Authors · 2024-11-22
> **Response, Part I/II**
>
> We thank the reviewer a lot for the valuable comments and insightful suggestions. Below we address your concerns point by point. We make corresponding changes in the revision (highlighted by blue fonts).
>
> ---
>
> ### **Weaknesses Part**
>
> 1. *"The title is inaccurate ... which is similarly vague."*
>
>      **Response:** We thank the reviewer for the suggestion. Throughout the whole revision, we now accurately state "linear self-attention" when we discuss it, rather than simply stating "attention". We also state in the abstract that our solution is an approximate one. For the title, currently we are not allowed to change it (there is no option for us to change the title during rebuttal), and we will change the title to more accurately reflect our scope once we are allowed to do so.
>
> 2. *"In that vein, please refer to ... less confusing to read."*
>
>     **Response:** Please see our response to point 1.
>
> 3. *"The review of models for neural scaling laws in ... contemporaneous with Bordelon et al 2024."*
>
>     **Response:** We thank the reviewer for the advice and the suggestion of these related works. We add a more adequate discussion on neural scaling laws in **Appendix A.2** (page 15).
>
> 4. *"The authors don't discuss the growing ... ridge regression, i.e., to convergence."*
>
>     **Response:** We add additional discussion about literature on in-context learning of linear regression in **Appendix A.3** (page 16). We also conduct a more detailed comparison and connection between our work and Zhang et al and Lu et al in Appendix A.3.
>
> 5. *"The paper would be much improved ... to structured covariates and targets?"*
>
>     **Response:** We thank the reviewer for the nice suggestion.
>
>     In fact, the MSFR problem considered in this paper can be seen as a limiting case of the (multitask) in-context regression under the source/capacity condition (i.e., with structured covariance and can be seen as a generalization of the task studied in Lu et al .(2024)) when the eigenvalue spectrum of the feature covariance matrix $\Sigma = \text{diag}(\omega_1, \dots, \omega_k)$ decays fast, i.e., $\omega_k\propto k^{-\tau}$ for large $\tau$. When $\tau$ is large, only the first eigenvalue $\omega_1$ is significant. This can be the case when the feature obeys the Gaussian distribution with zero mean and the aforementioned covariance matrix $\Sigma$. In such case, the feature in Eq.(2) of the MSFR problem can exhibit a very similar feature covariance matrix.
>
>     In addition, since the covariance matrix is crucial for the in-context learning dynamics (because the covariance matrix determines the matrix $\boldsymbol H$ in Eq.(9) and $\boldsymbol H$ determines the learning dynamics), we expect that MSFR problem will show similar neural scaling laws as the regression problem under source-capacity condition. Please see **Appendix H.1** (page 32 of the revision) where we discuss this more closely and conduct numerical experiments to verify it (Fig.5, page 34).
>
>     The MSFR problem can also be generalized to the case when the feature is only approximately sparse (**Appendix H.2**, page 34). Finally, we note that the solution presented in Theorem 3.1 is an exact solution when $\boldsymbol H$ admits an idempotent-like structure (**Appendix H.3**, page 35).
>
> 6. *"The main analytical results of the paper are ... rather than Theorems"*
>
>     **Response:** We state clearly that the solution is an expansion at large $\psi_s$ in Theorem 3.1. The approximation can be bounded by $f_s^1(t)$ which is derived in Appendix G, i.e., $|f_s(t) - f_s^0(t)| = \epsilon_s |f_s^1(t)| + \mathcal{O}(\epsilon_s^2)$. In addition, we would like to mention that, when $H_s$ obeys the idempotent-like structure as $H_s^0$, the solution in Theorem 3.1 is an exact solution. Thus we prefer to write it as Theorem currently, which also provides more convenience for the reader.
>
> ---
>
> ### **Questions Part**
>
> We thank the reviewer for these questions.
>
> 1. *"The discussion in lines ... networks that can learn features."*
>
>     **Response:** Our original statement was "linear model" and we did not claim that "Bordelon studied neural networks". We now change the statement to the more accurate "random feature model" in the revision.
>
> 2. *"Why do you bother to ... overhead for the reader."*
>
>     **Response:** We change this equation by explicitly writting out the form of the MSE loss (Eq.(6) in page 5).

---

> > ### Author Response · Authors · 2024-11-22
> > **Response, Part II/II**
> >
> > ### **Questions Part (continuing)**
> >
> > 3. *"Unless I'm mistaken, all experiments ... with softmax attention."*
> >
> >     **Response:** We supplement numerical experiments to examine neural scaling laws for the softmax attention, where, for completeness, we use the parameterization $W_{K}^TW_{Q}$ rather than a single merged $W_{KQ}$ in all the experiments. These additional experiments are presented in **Appendix I.1** (page 36 to 37). We show that softmax self-attention still displays the same neural scaling laws as linear self-attention when it is trained by gradient descent (Fig.8, page 36).
> >
> > ---
> >
> > **Reference**
> >
> > Lu et al. (2024). Asymptotic theory of in-context learning by linear attention.

---

> > > ### Comment · Reviewer_9RQV · 2024-11-22
> > >
> > > Thank you for your detailed reply. Having read your responses to the other reviewers and the substantially expanded revised manuscript, I think this paper should be accepted, and will therefore raise my score.

---

> > > > ### Author Response · Authors · 2024-11-23
> > > >
> > > > We thank the reviewer so much for the quick reply and we really appreciate the reviewer for raising score.

---

### Official Review · Reviewer_7YzX · 2024-10-30

**Soundness:** 3
**Presentation:** 4
**Contribution:** 3
**Rating:** 8
**Confidence:** 3

**Summary:**

The paper establishes a theoretical framework to study the gradient flow learning dynamics of a single linear attention layer on a synthetic multi-task regression task.
It provides closed-form solutions for the dominant ODE dynamics based on perturbation theory and uses them to derive neural scaling laws w.r.t. training time, data set size, model size, and compute budget.

**Strengths:**

- **S1: Presentation & Contribution** Although the paper's content is relatively dense and introduces a lot of symbols, the presentation is very clear and, in my opinion, strikes a very good balance between mathematical detail and prose explanations.
  Specifically, I appreciated the author's clear outlines before executing the mathematical steps, which helped me understand many of the details, and convinced me that the contribution is non-trivial.

- **S2: Generality** I believe the synthetic multi-task sparse regression task, as well as the derived solutions of the dominant ODE dynamics can be of interest to study other phenomena in transformers, not only neural scaling laws.
  Therefore, the presented framework seems useful for future works.

**Weaknesses:**

The following are mostly minor concerns which, if addressed, would strengthen my confidence.

- **W1 (minor): Gap to 'practical' attention** The paper studies a relatively simplistic attention layer without softmax, skip connections, MLPs, depth, etc.
  I think this is okay, because the contribution is to derive closed-form expressions for the dynamics, which is likely intractable when including these components.
  It would be interesting to further investigate how useful the theoretical predictions are in a setting that more closely resembles a practitioner's setup.
  Specifically:
  1. Theoretically: How does the parameterization $W_K^\top W_Q$ (used in practise), rather than $W_{KQ}$, affect the dynamics, i.e. how would the presented results change?
  2. Empirically: How well does the theoretical prediction (linear attention) match the simulation results in Figs. 3/4 when using softmax attention?
  3. Empirically: Many attention-based models are trained with AdamW. How well does the theoretical prediction (gradient flow) match the simulation results in Figs. 3/4 when training with AdamW+small learning rate?

- **W2 (minor): presentation/editing suggestions**
  - It would be great if the authors could provide some practical recommendations based on the theoretical insights into neural scaling laws in Section 4.
  - It would be great if the authors could highlight more connections and differences with related works throughout the text, e.g. other works studying the learning dynamics of MLPs and CNNs (the author may want to cite https://proceedings.mlr.press/v202/pinson23a/pinson23a.pdf), or contrasting the neural scaling laws in Tables 1,2 with other empirical or theoretical results.
  - In the paragraph above Theorem 3.1, could you highlight which set of equations have the same form as the Riccati equation?
    I also believe it would be good to extract some of the currently inlined math into environments to facilitate looking up symbols, specifically on page 6.
  - Typos:
    - Line 108: Why not just use $a \approx b$ to indicate approximately equal?
      Is this notation used in the main text?
    - Equation 2: I think it should be $\mathbb{R} \times \mathbb{R}^d$
    - Equation 7: Can you say what $H_s$ is?
      It looks like it contains the second moments of the data and labels.
    - L269: 'can be rather exact' sounds weird; do you mean 'approximate, tractable solution'
    - L285: Maybe use '^T' instead of '\dot' for consistency with L253.
    - L297: $\epsilon$ should be $\epsilon_s$.
    - L331: To me it sounds weird to say the solution is 'considerably exact'.
      Maybe replace with 'is a good approximation'.
    - L379/381: 'with respect' is missing 'to'
    - Figs. 3d/4b/4f: Add a legend for each curve.
    - L536: 'a variety properties' misses an 'of'

**Questions:**

Please see the questions from W1. Additional minor questions:

4. L387: Why does the compute not scale with $N$, the data set size?
   I believe that computing a gradient should scale linearly in $N$.

---

> ### Author Response · Authors · 2024-11-22
> **Response, Part I/II**
>
> We thank the reviewer so much for the insightful comments and appreciation of our work. We address your concerns and questions in the following point by point. We also make the corresponding changes in the revision following your suggestions.
>
>
> ### **Weaknesses Part**
>
> ---
>
> #### **W1 Part**
>
> 1. *"Theoretically: How does the parameterization ... presented results change?"*
>
>    **Response:** We find that this is indeed an insightful and highly nontrivial question. To see the reason why this question is challenging, we consider a simple theoretical setting: the model prediction is now (we omit all the subscript $s$ for simplicity) $$f = \mathbf{v}^T \mathbf{H} \mathbf{w} q$$ where $q \in \mathbb{R}$ is simply a trainable scalar. And the learning dynamics for this model becomes (let $r = f - \Lambda$ be the residual) $$\dot{\mathbf{v}} = - r \mathbf{H}\mathbf{w}q, \quad \dot{\mathbf{w}} = - r\mathbf{H}\mathbf{v}q,\quad  \dot{q} = - r \mathbf{v}^T \mathbf{H} \mathbf{w}. $$
>
>      If we define $\tilde{\mathbf{w}} = \mathbf{w}q$ (like merging $\mathbf{W}_K$ and $\mathbf{W}_Q$), then the dynamics of this system becomes $$\dot{\mathbf{v}} = - r \mathbf{H}\tilde{\mathbf{w}}, \quad \dot{\tilde{\mathbf{w}}} = - r(q^2 \mathbf{I} + \mathbf{w}\mathbf{w}^T)\mathbf{H}\mathbf{v}.$$
>
>     Thus $\mathbf{v}$ and $\tilde{\mathbf{w}}$ now have learning dynamics of very different forms, and we think it might be hard to directly applying our strategy to exactly solve it. This might be one important reason why many works on the theoretical analysis of linear self-attention adopt the parameterization $\mathbf{W}_{KQ} $ rather than using $\mathbf{W}_K^T\mathbf{W}_Q$. And we believe it could be a highly interesting future direction to solve such dynamics. It might also be interesting to mention that the solution in Theorem 3.1 is an exact solution when $\boldsymbol{H}$ admits an idempotent-like structure, i.e., $\boldsymbol{H}^2 = \mu \boldsymbol{H}$ for some constant $\mu$, which is discussed in Appendix H.3 (page 34).
>
>     On the other hand, we conduct additional numerical experiments to examine whether our conclusions can still hold for the parameterization $\mathbf{W}_K^T\mathbf{W}_Q$, which will be discussed in the following two points.
>
> 2. *"Empirically: How well does the theoretical ... when using softmax attention?"*
>
>    **Response:** To address your concern, we supplement numerical experiments to examine neural scaling laws for the softmax attention, where, for completeness, we use the parameterization $W_{K}^TW_{Q}$ rather than a single merged $W_{KQ}$ in all the experiments. These additional experiments are presented in **Appendix I.1** (page 36 to 37). Specifically, we show that softmax self-attention still displays the same neural scaling laws with respect to time, model size, data size, and the optimal compute as linear self-attention when it is trained by gradient descent (Fig.8, page 36).
>
> 3. *"Empirically: Many attention-based models ... with AdamW+small learning rate?"*
>
>     **Response:** To examine the effects of optimization algorithms on neural scaling laws in the MSFR problem, we train softmax self-attention with AdamW. We also use the parameterization $W_{K}^TW_{Q}$. We report the results in **Appendix I.2** (from page 37 to 38).
>
>     In particular, we show that, while AdamW still leads to similar neural scaling laws with respect to model size and data size (Fig.10a and Fig.10b, page 38), it inevitably contributes to different neural scaling laws with respect to time $t$ and the optimal compute (Fig.10c and Fig.10d, page 38) compared to gradient descent. This is because AdamW has a very different learning dynamics (e.g., it could converge faster). The observation that optimization algorithms affect the neural scaling laws with respect to time (and thus the optimal compute) aligns with the empirical evidence in Hoffmann et al. (2022), where they revealed that AdamW shows a different test loss behavior against the training time when compared to Adam.
>
> ---
>
> #### **W2 Part**
>
> We thank the reviewer for the suggestions in this part. We make the corresponding changes in the revision and summarize these changes as follows.
>
> 1. *"It would be great if the authors could provide some practical ... in Section 4."*
>
>     **Response:** In Section 4.2, we find that it is better to balance the context sequence length for different tasks to obtain a satisfied test loss given a limitation of optimization steps, otherwise one would get a worse scaling law cure with respect to the training time $t$ (the cyan solid curve with varied context sequence length is worse than the black dotted curve with constant sequence length). In addition, we also find that diversity of task (varied task strength) is beneficial for model performance since it will lead to better scaling laws with respect to time, model size, data size, and the optimal compute (Fig.4c - Fig.4f).

---

> ### Author Response · Authors · 2024-11-22
> **Response, Part II/II**
>
> #### **W2 Part (continuing)**
>
> 2. *"It would be great if the authors could highlight more connections and ... with other empirical or theoretical results."*
>
>     **Response:** We thank the reviewer for suggesting this nice work. We cite the recommended reference in the main paper (line 69-70, page 2). In addition, due to the lack of space, we present a more detailed discussion of related works on learning dynamics of neural networks in **Appendix A.1** (page 15). We also elaborate the comparison between neural scaling laws in Table 1, 2 and other empirical results and theoretical results (line 447-450 in page 9).
>
> 3. *"In the paragraph above Theorem 3.1, could you highlight ..."*
>
>     **Response:** We highlight that Eq.(12) (line 314-315, page 6 of the revision) has the same form as Riccati equation before Theorem 3.1. We also put the definition of $\boldsymbol{H}_s^0$ and $\boldsymbol{H}_s^1$ into environments to make them more readable.
>
> 4. *"Typos"*
>
>     We thank the reviewer so much for pointing these typos out. For each of them,
>
>    - We use $a \sim b$ to mean that $a$ is approximately close to $b$ in the sense that we neglect both irrelevant coefficients and constants, e.g., $a \sim b$ can mean $a$ is approximately close to $Zb$ for some irrelevant constant $Z$. This notation is only used when we derive the neural scaling laws by investigating the asymptotic behaviors (thus we do not care about the irrelevant coefficients). We move the notation $a \sim b$ from the main body to Appendix;
>
>    - We fix this in line 124-125, page 3;
>
>    - Indeed, $\boldsymbol{H}_s$ is composed of the feature covariance matrix and target (please see line 236-237 in page 5 and Eq.(22) and (23) in page 17);
>
>    - We fix this as "approximately exact" in line 268, page 5;
>
>    - We change this to $\eta^T H\eta$ and $\rho^T H \rho$;
>
>    - We fix this in line 297, page 6;
>
>    - We change this to "a good approximation" in line 332;
>
>    - We add the missing word in these lines;
>
>    - We add legend in these figures;
>
>    - We add the missing word "of".
>
> ---
>
> ### **Questions Part**
>
> *"Why does the compute not ... should scale linearly in $N$."*
>
> **Response:** In this case, we assume that the number of data $N$ (it appears in the computation of gradients only in the form of #$_s/N$) is unlimited so that #$_s/N$ is close to the true distribution $s^{-\alpha}$. Thus the computation of gradients is not affected by $N$ in this case.
>
> ---
>
> **Reference**
>
> 1. Hoffmann et al. (2022). Training compute-optimal large language models.

---

> > ### Comment · Reviewer_7YzX · 2024-11-25
> >
> > Thanks for your response!
> >
> > Could you further elaborate on this response? I'm not sure I understand how you compute the gradients in a data-independent way. My current understanding would be that you have to employ some form of automatic differentiation, which would scale with $N$.
> >
> > > Response: In this case, we assume that the number of data
> >  (it appears in the computation of gradients only in the form of #
> > ) is unlimited so that #
> >  is close to the true distribution
> > . Thus the computation of gradients is not affected by
> >  in this case.

---

> ### Author Response · Authors · 2024-11-28
>
> We thank the reviewer for the reply! Below we answer your question.
>
> The setting for considering the optimal compute assumes an unlimited amount of data, i.e., the data size $N \to \infty$ should not be the bottleneck of the performance of the model and it is the training time $t$ or the model size $D$ that is the bottleneck of training. This is the case where a batch of data with a fixed sufficiently large batch size is taken from the unlimited training data set in each run during training, i.e., the compute resource allocated for the computation of gradient in each run is fixed (when the model size $D$ is fixed). The regime that we are interested in is, given a  compute resource  $\mathcal{C}=tD$, we should decide to choose a larger model (larger $D$) or train the model for a longer time (larger $t$), which can be done by comparing the training time $t$ and model size $D$ required for achieving the optimal loss given a fixed $\mathcal{C} = tD$.

---

### Official Review · Reviewer_5pWj · 2024-11-02

**Soundness:** 2
**Presentation:** 3
**Contribution:** 2
**Rating:** 5
**Confidence:** 3

**Summary:**

This paper presents a theoretical analysis of power laws in in-context learning using Transformers. Specifically, it introduces a task setup called multitask sparse feature regression and approximates the learning dynamics of self-attention trained by gradient descent on this setup through perturbation analysis. The authors verify that the test loss follows scaling laws with respect to training time, model size, the amount of training data, and the optimal compute budget.

**Strengths:**

- This paper demonstrates a commendable effort in addressing the challenging theoretical analysis of neural scaling laws in in-context learning, which is both complex and impactful.
- It provides a visually clear figure explaining the multitask sparse feature regression task, along with an accessible proof outline that enhances reader comprehension.
- The paper establishes an interesting connection between in-context learning dynamics and the Riccati equation. It is particularly intriguing that $g_s^0h_s^0$ emerges as a conserved quantity.

**Weaknesses:**

- **The problem setup appears somewhat artificial.** While the gradient descent analysis for multitask sparse feature regression is impressive, the problem setting itself seems contrived and may lack applicability. For instance:
  + The probability of task selection $\mathcal{P}(S=s)$, task strength $\Lambda_s$, and context sequence length $\psi_s$ are all exponentially proportional to $s$. The assumption that these three factors are coordinated (e.g., larger $\mathcal{P}(S=s)$ implies larger $\Lambda_s$ and $\psi_s$) seems limiting for practical applications.
  + In this multitask sparse feature regression task, only $\phi(s,\hat{\boldsymbol{x}})$ is necessary to predict $\hat{y}$, making $\phi(s,\boldsymbol{x}^{(1)}),\dots,\phi(s,\boldsymbol{x}^{(\psi_s)})$ effectively irrelevant. This raises doubt as to whether this setup truly qualifies as in-context learning. A token-wise feed-forward network would likely suffice, giving the impression that self-attention has been unnecessarily added to complicate the problem.
- **Several typos and ambiguities in presentation.** A few observations are listed below:
  + In eq. (2), $\mathbb{R}^d \times \mathbb{R}$ should be $\mathbb{R} \times \mathbb{R}^d$.
  + In Section 2.2, $f:\mathbb{R}^{(\mathcal{N}_S+1) \times (\psi_s+1)} \mapsto \mathbb{R}$,(page 4, line 182), the $s$ is incorrectly capitalized, and $\mapsto$ should be $\to$.
  + In Section 2.3, $\mathbb{R}^{\mathcal{N}_s \times (\psi_s+1)}$ should be $\mathbb{R}^{(\mathcal{N}_s+1) \times (\psi_s+1)}$.
  + Below eq. (7), the phrase “does not depend on $n$, where ...” is misleading, as eq. (7) depends on the task type $s^{(n)}$; rephrasing this may clarify the intent.
  + In the second line of eq. (26), it seems that $\mathcal{O}(\epsilon_s^2)$ is unnecessary, whereas it may be needed in eq. (28).

**Questions:**

- $f_s^0$ depends on the training data, so is it ok not to take the expectation over the training data in the definition of Test Loss (eq. (14))?
- In Section 2.2 on the generation of in-context data, is it correct to assume that the data sequence $\boldsymbol{x}^{(1)},\dots,\boldsymbol{x}^{(\psi_s)}$ is sampled independently and identically from $\mathcal{P}_X$?
- Apologies if mistaken, but in eq. (31), is $\frac{\psi_s \\#_s }{N}$ missing? Wouldn’t this result in $a_s = \psi_s^2 \\#_s (1+\Lambda_s^2) / N$?
- When transforming the summation to an integral in eq. (62), is $f_s^0$ defined for non-integer $s \in \mathbb{R} \setminus \mathbb{N}$? Similarly, does $\mathcal{P}(S=s)$ in eq. (64) have the same consideration?

---

> ### Author Response · Authors · 2024-11-22
> **Response, Part I/II**
>
> We thank the reviewer for the valuable comments. Below we address your concerns and questions point by point.
>
> ---
>
> ### **Weaknesses Part**
>
> 1. *"The problem setup appears somewhat ... may lack applicability."*
>
>    **Response:** We first discuss the applicability of our work. Our setup is motivated by the random feature model with structured covariance matrix $\Sigma = \text{diag}(\omega_1, \dots, \omega_d)$ such that the eigenvalues $\omega_k \propto k^{-\tau}$.
>
>      In particular,  when $\tau$ is large, the eigenvalue spectrum of $\Sigma$ decays very fast thus the first eigenvalue $\omega_1$ dominates the covariance matrix $\Sigma$, e.g., the feature obeys the Gaussian distribution with zero mean and the aforementioned covariance matrix $\Sigma$. In such case, the learning dynamics of the in-context learning regression feature model is very similar to that of the MSFR problem. Please see **Appendix H.1** of the revision (page 32) for a detailed discussion, where we also conduct numerical experiments to show that these two problems have very similar neural scaling laws (Fig.5, page 34).
>
>     Therefore, we believe that our MSFR problem is not a contrived one but a reasonable proxy model for understanding various intriguing properties of the self-attention (and possibly transformer). Considering that the theoretical analysis of transformers and self-attention is generally intricate and highly challenging, we would like to highlight that our results could be of broad interest as our solution captures the learning dynamics along the whole training trajectory for linear self-attention.
>
>     Finally, we would like to mention that we also discuss the generality of the MSFR problem in **Appendix H** (page 32 to 35 with numerical experiments), such as MSFR with approximately sparse feature (Appendix H.2) and tasks with idempotent-like $H$ (Appendix H.3).
>
> 2. *"The probability of task selection ... seems limiting for practical applications."*
>
>     **Response:** We would like to first clarify that our method for deriving the solution of the dynamics can be applied to any distributions of task type $s$, context sequence length $\psi_s$, and task strength $\Lambda_s$. We now discuss these assumptions separately.
>
>     - The assumption for the power law distribution of the task type is based on the empirical observation for the neural scaling laws in Michaud et al. (2023), where they showed that language models can exhibit different skills that obey a power law distribution.
>
>     - The assumption that the task strength $\Lambda_s \propto s^{-\gamma}$ is also from the random feature model under the source-capacity condition, while we note that this is not necessary for results in this paper and can be generalized to other types of distributions. We make this assumption because it aligns with the empirical observations (e.g., Cui et al. (2021)).
>
>     - The assumption that the sequence length $\psi_s\propto s^{-\beta}$ is inspired by the underlying power-law correlations in language sequence (e.g., Ebeling \& Poschel. (1994)). And, again, we note that this is not required for deriving the solution of the learning dynamics in this paper and can be generalized to other types of distributions. We make this assumption because it aligns with the empirical observations.
>
> 3. *"In this multitask sparse feature regression task ... unnecessarily added to complicate the problem."*
>
>    **Response:** We would like to kindly highlight that, the study of the regression problem (which certainly can be solved by feed-forward networks) for linear self-attention with the in-context learning is an important starting step towards ultimately understanding the self-attention and even transformers from a theoretical perspective, especially considering that the study of the realistic setting is extremely challenging and still an open problem. Our work also falls into this category and our purpose is devoting to understanding intriguing properties of self-attention, rather than complicating the problem unnecessarily. For example, we novelly characterize the neural scaling laws for linear self-attention with the role of context sequence length.
>
> 4. *"Several typos and ambiguities in presentation."*
>
>     **Response:** We thank the reviewer very much for pointing the typos. For each point, we
>
>    - fix this to $\mathbb{R}\times \mathbb{R}^d$ (Eq.(2), line 124-125 of page 3);
>
>    - fix this to $f: \mathbb{R}^{(\mathcal{N}_s + 1)\times (\psi_s + 1)} \to \mathbb{R}$ (line 182 of page 4);
>
>    - it should be $\mathbb{R}^{\mathcal{N}_s\times (\psi_s + 1)}$;
>
>    - rephrase the sentence (line 235, page 5);
>
>    - add $\mathcal{O}(\epsilon_s^2)$ to Eq.(28).

---

> > ### Author Response · Authors · 2024-11-22
> > **Response, Part II/II**
> >
> > ### **Questions Part**
> >
> > We thank the reviewer for the questions. We answer your questions below.
> >
> > 1. The expectation is taken over the data distribution.
> >
> > 2. The data sequence is indeed sampled independently and identically from $P_X$.
> >
> > 3. It is indeed missing in the second equality. We change this in the revision (Eq.(32), page 19). $a_s$ is still $\psi_s$#$_s(\Lambda_s^2 + 1) / N$ because one $\psi_s$ will go to $g_s^0$.
> >
> > 4. We indeed consider $f_s^0(t)$ for non-integer $s$, which is only for the purpose of theoretical analysis of the asymptotic behaviors. When the limit of $s$ is large, this is acceptable. We do not use non-integer $s$ when conducting experiments.
> >
> > ---
> >
> > **Reference**
> >
> > 1. Michaud et al. (2023). The quantization model of neural scaling.
> >
> > 2. Cui et al.(2021). Generalization error rates in kernel regression: The crossover from the noiseless to noisy regime.
> >
> > 3. Ebeling & Poschel. (1994). Entropy and long-range correlations in literary english.

---

> > > ### Comment · Reviewer_5pWj · 2024-11-25
> > >
> > > Thank you for your reply.
> > > Below, I have listed my comments on your rebuttal.
> > > 1. The detailed explanation added to Appendix H, as well as the connection to Lu et al. (2024), has helped me better understand the background of your problem setup and why you chose this formulation. Thank you for providing these clarifications.
> > > 2. Regarding the distributions of task type, task strength, and sequence length following a power law, I do not have significant objections. My concern lies instead in the assumption of a positive correlation among these variables—for example, that tasks with greater task strength tend to have longer sequence lengths, and vice versa. Is this assumption natural in real-world settings? I would appreciate further justification or discussion on this point.
> > > Additionally, as Reviewer FJbQ also noted, if your results remain valid even when these assumptions are relaxed, I believe explicitly stating such a generalized result would strengthen the impact and applicability of your work.
> > > It would also help address potential concerns from readers who may find the assumptions of the problem setup restrictive.
> > > 3. I understand the necessity of simplifying the problem setup to study the learning dynamics of self-attention in in-context learning, which is indeed a highly challenging. However, I am still unsure if the current problem setup can truly be referred to as in-context learning. It would be helpful if you could provide a more detailed explanation of why you consider this problem to fall under the category of in-context learning. Specifically:
> > >     - What does the model learn "in-context" from the input sequence other than the query token?
> > >
> > >       (Perhaps the model is learning task strength in-context, but I would like clarification on whether this is simply a result of the Assumption 3.1 $\psi_s \gg 1$, or if the current problem setup fundamentally requires the model to learn task strength in-context even without such an assumption.)
> > >     - How does solving this problem contribute to our understanding of in-context learning?
> > >
> > > And thank you for answering my questions.
> > > Overall, your work on analyzing the approximate learning dynamics of self-attention is impressive, and the revisions have made the positioning of this research much clearer.
> > > Additionally, several presentation issues have been effectively addressed in the revised version.
> > > However, I am still uncertain about whether the problem setup truly aligns with in-context learning, which makes it challenging for me to confidently assign a positive score to the paper.

---

> ### Author Response · Authors · 2024-11-25
> **Response**
>
> Thank you so much for your prompt reply and interesting questions. We are clearer about your concerns and below we answer your questions.
>
> 1. *Point 2 of the comment: "Regarding the distributions of ...  the problem setup restrictive."*
>
>     **Response:** We would like to first highlight that the current form of Theorem 3.1 is already applicable for any distributions of task type, task strength, and sequence length: the derivation of solution in Theorem 3.1 does not make any assumptions on their forms. We thank the reviewer for the nice suggestion and we state this point clearly below Theorem 3.1 in the updated revision (line 345-347) as suggested by the reviewer.
>
>     We specify the setting where a positive correlation exists in these variables in Section 4.2 (other sections are not related to this correlation) as an example to investigate how the neural scaling laws of linear self-attention are jointly affected by these variables. In this case, besides the task type power law distribution, one can also assume a power law type distribution only for the sequence length to examine how the neural scaling laws are affected (Fig.4a and Fig.4b where the task strength does not follow a power law).
>
> 2.  For the two questiones raised in Point 3:
>
>      - *"What does the model learn "in-context" from the input sequence other than the query token?"*
>
>         **Response:** The model is indeed learning task strength in-context, even without Assumption 3.1. In particular, the model prediction for the task $s$ can be written as $\boldsymbol{v}_s^T \boldsymbol{H}_s \boldsymbol{w}_s$, where $\boldsymbol{H}_s = \Phi(s, \mathbf{X})\Phi(s, \mathbf{X})^T$ is explicitly dependent on all tokens of the input sequence regardless of Assumption 3.1. As a result, the input sequence in fact determines the output along with the query token, i.e., the task strength is predicted "in-context". A simple example is that, at test time, an input sequence $\phi(s', x^{(1)}), \dots, \phi(s', x^{(\psi_s)}), \phi(s, \hat{x})$ would lead the trained model to predict a target that is different from $\hat{y}$, target of the query token $\phi(s, \hat{x})$, if $s \neq s'$.
>
>      - *”How does solving this problem contribute to our understanding of in-context learning?“*
>
>        **Response:** An immediate novel insight of our results is the characterization of the role of context sequence length $\psi_s$ in the in-context learning process, as the solution depends on $\psi_s$ explicitly through $a_s$. Furthermore, our main results in Section 4 novelly characterize the neural scaling laws of linear self-attention with in-context learning, which additionally allows us to compare these laws with that of other models from a theoretical perspective.

---

> > ### Comment · Reviewer_5pWj · 2024-11-26
> >
> > Thank you for your detailed explanation. It has greatly improved my understanding of the structure of your paper.
> > 1. If the assumptions about the distributions of task strength and sequence length are used only in Section 4.2, I believe it would be better to limit these assumptions to Section 4.2 rather than introducing them in Section 2.2. This approach would enhance the generality of the problem setup.
> > Additionally, in Section 4.2, you consider test loss under the condition that task strength is the same across all tasks while varying sequence length. However, this analysis remains bound by the constraint that task strength and sequence length are non-negatively correlated, and it does not cover scenarios where these variables exhibit non-trivial relationships with respect to $s$. The same can be said for the relationships between the probability of task selection and task strength, or the probability of task selection and sequence length.
> > From a technical perspective, I understand that the conditions under which the test loss follows a power law are quite complex, and that the examples of $\psi_s \propto s^{-\beta}$ and $\Lambda_s \propto s^{-\gamma}$ are provided to illustrate one such case. However, since these examples are included, it would be worthwhile to discuss their validity as well.
> >
> > 2.
> >   - I feel that simply stating that both $H_s$ and the model's prediction depend on all tokens of the input sequence is not sufficient to convincingly argue that the model is performing in-context learning.
> > While it is challenging to provide a precise definition of in-context learning, your statement at the beginning of Section 2.2 that "A remarkable ability of LLMs is that they can perform in-context learning to adapt to a specific task given a context in the form of instructions" provides an important perspective. To convincingly argue that the model performs in-context learning, I believe it is necessary to demonstrate that the model exhibits this kind of adaptive ability in a clear and compelling manner.
> > The reason for my concern is that the problem setup being studied is clearly solvable even by models that are not in-context learners.
> > In this paper, Assumption 3.1 $\psi_s \gg 1$ ensures that the tokens prior to the query token $\phi(s,x^{(1)}),\dots,\phi(s,x^{(\psi_s)})$ dominate as keys in the linear attention mechanism, effectively rendering the query token insignificant as $\epsilon H^1_s$ is treated as an insignificant perturbation.
> > If Assumption 3.1 were removed, for instance, a dataset consisting only of query tokens could likely still be effectively learned by the linear self-attention block. While this is an extreme example, it highlights the subtlety of this discussion: Is the model truly performing in-context learning, or is it merely predicting the label from the query token?
> > To provide stronger evidence that the model is actually performing in-context learning, it might be helpful to demonstrate, for example, that the trained model’s parameters do not depend on task strength and that the model learns it from the tokens preceding the query token during testing. Alternatively, showing that the model can predict labels for unseen task strengths would also support this argument.
> > It is admittedly a challenging question how to address this issue, but I would emphasize again that the root cause lies in the fact that the problem setup under consideration can also be solved by non-in-context learners.
> >   - Thank you for summarizing the implications of this research for in-context learning.

---

> ### Author Response · Authors · 2024-11-27
> **Response**
>
> We thank the reviewer for the quick reply. Below we address your concerns.
>
> 1. We would like to clarify that the independence on the assumptions for distributions or relations of task type, task strength, and sequence length of Theorem 3.1 in fact provides us a powerful tool to characterize neural scaling laws for different kinds of distributions of task type, task strength, and sequence length, no matter if there exist nontrivial relationships among them or not. More specifically, for general task strength $\Lambda_s = \mathcal{G}(s)$ and sequence length $\psi_s = \mathcal{F}(s)$, one can rewrite $a_s$ and other related quantities in the solution of Theorem 3.1 as $a_s = $#$_s \mathcal{F}(s)((\mathcal{G}(s))^2 + 1) / N$. Then one can study the scaling laws by analyzing the asymptotic behaviors of the solution either with analytical methods introduced in Section 4 when the induced problem can be solved analytically (e.g., Section 4.1) or with the direct numerical simulation otherwise.
>
> 2. We acknowledge that the problem setup considered in our paper can be solved by non-in-context learners, but if we organize the task in an in-context learning format, we can evidently show that the linear self-attention can indeed perform in-context learning to adapt to new tasks. In addition, the growing literature on in-context learning of linear regression, e.g., Zhang et al.(2023) and Lu et al.(2024), has already revealed the significance of such setting by showing that linear self-attention performs in-context learning by utilizing the input tokens prior to the query token, even though the linear regression problem can be solved by a non-in-context learner.
>
>    For our setting, the dependence of $\boldsymbol{H}_s$ on all input tokens, including those prior to the query token, indicates that the model will adapt itself for different input sequences. Below we provide a simple and straightforward example to explain this.
>
>    Consider the linear self-attention trained on the training sequence $\Phi(s, X)$ for the task $s$ where the task strength is $\Lambda_s$, the prediction for the query token $\hat{\phi}(s, \hat{x})$ and training sequence $\Phi(s, X)$ of the trained self-attention (i.e., the training time $t\to \infty$) can be written as $\boldsymbol{v}_s^T(\infty) \boldsymbol{H}_s \boldsymbol{w}_s(\infty)$ where $\boldsymbol{v}_s(\infty)$ means $\boldsymbol{v}_s(t)$ at $t = \infty$ and $\boldsymbol{H}_s = \Phi\Phi^T$. We assume a similar constraint for the model as in Zhang et al. (2023): the first $\mathcal{N}_s$ components of $\boldsymbol{v}_s$ and the last component of $\boldsymbol{w}_s$, i.e., $w _{ s; \mathcal{N}_s + 1}$, are all zero for any $t \geq 0$. In this case, the output of the model $f_s$ for the training sequence $\Phi(s, X)$ can be written as $$f_s = v _{s; \mathcal{N}_s + 1}(\infty)w _{s; s}( \infty ) \psi _s \Lambda _s.$$Because $f _s = \Lambda _s$ at $t = \infty$ (the model is perfectly trained on the training sequence), we obtain that $v _{s; \mathcal{N} _s + 1 }( \infty )w _{s; s} (\infty) = 1 / \psi _s$.
>
>    The question now becomes what the prediction of the self-attention for the query token $\hat{\phi}(s, \hat{x})$ will be if a different input sequence $\tilde{\Phi}(s, X)$ is given. As commented by the reviewer, we let the task strength for the input tokens in $\tilde{\Phi}(s, X)$ prior to the query token be $\tilde{\Lambda}_s \neq \Lambda_s$, i.e., the task strength is unseen in the training data. Then the prediction of the self-attention for the query token given the input sequence $\tilde{\Phi}(s, X)$ will become $$f_s = \boldsymbol{v}_s^T (\infty) \tilde{\Phi}\tilde{\Phi}^T \boldsymbol{w}_s (\infty)= v _{s; \mathcal{N} _s + 1}(\infty)w _{s; s}(\infty)\psi _s \tilde{\Lambda} _s = \tilde{\Lambda} _s.$$ Therefore, linear self-attention returns the unseen task strength $\tilde{\Lambda}_s$ by performing in-context learning to explore in-context data to adapt itself to a new task, rather than only relying on the query token.
>
>    - In addition, we would like to clarify that Assumption 3.1 does not completely render the query token as an insignificant perturbation. Specifically, by writing the output of linear self-attention explicitly as $VHW\hat{\phi}$, we can clearly see that the dependence of the output on the query token $\hat{\phi}$ cannot be ignored even when it is insignificant inside $H$.
>
> **Reference**
>
> Lu et al. (2024). Asymptotic theory of in-context learning by linear attention.
>
> Zhang et al. (2023). Trained Transformers Learn Linear Models In-Context.

---

> > ### Comment · Reviewer_5pWj · 2024-11-28
> >
> > Thank you for your detailed response.
> > 1. I agree with your clarification, which aligns with the points I raised in my previous comment.
> > Therefore, wouldn’t it be better to introduce task strength and sequence length in a more general form in Section 2.2, and then provide their specific forms along with a discussion on the validity of these assumptions in Section 4.2? (While there is no time left for further revisions, I believe this could further enhance clarity and rigor.)
> > 2. Apologies if I am mistaken, but from what I have seen, in the work by Zhang et al.(2023) and Lu et al.(2024), the input vectors $x$ in the input sequence (or training prompt) are sampled from a Gaussian distribution, resulting in a problem setup where the task or context cannot be inferred solely from the query token.
> > As for the example, thank you for presenting this concrete example; this is exactly the kind of clarification I was looking for.
> > However, I have questions regarding the constraint on $v$ and $w$ used in this example:
> >     - Were these constraints also assumed when deriving the learning dynamics in the paper?
> > Or are these constraints introduced for the first time in this rebuttal comment?
> >     - If these constraints were introduced here, do they affect the derived learning dynamics in any way?
> >
> >     Additionally, the constraint that the first $\mathcal{N}_s$ components of $v_s$ are all zero corresponds to ignoring the sparse feature $\phi$  within the in-context data $\Phi$ when calculating the value vector in self-attention. I believe it would be necessary to discuss the validity of this assumption as well.
> > 3. Of course the query token $\phi(s,\hat{x})$ is used as a query in self-attention, but I would like to ask about its role as a key.
> > From my understanding, the interaction between the query token $\phi(s,\hat{x})$ as a query and as a key within self-attention is isolated in $H_s^1$ as an insignificant perturbation. In other words, Assumption 3.1 $(\psi_s \gg 1)$ seems to force the query token to interact only with the other tokens.

---

> ### Author Response · Authors · 2024-11-29
> **Response, Part I/II**
>
> We thank the reviewer for the reply and questions.
>
> 1. We thank the reviewer for the suggestion. We will revise our manuscript accordingly in the next revision as we are currently not allowed to submit a new revision.
>
> 2. We would like to first clarify that whether the input vector $x$ is sampled from a Gaussian distribution (or some other distribution) is not essential for the problem setup where the task cannot be inferred solely from the query token. Instead, the crucial part is the necessary exploration of the context information, i.e., the relation between the input vector $x$ and its target $y$ encoded in tokens prior to the query token. More specifically, the in-context prediction for the same query token $\hat{x}$ will change when the relation between $x$ and $y$ in the input sequence prior to the query token changes, regardless of the type of distribution of $x$.
>
>    We now answer your questions regarding the constraints.
>
>     - *"Were these constraints also assumed when deriving the learning dynamics in the paper?"*
>
>        **Response**: They are not assumed in the paper as our paper aims to characterize the solution in the general case, i.e., without constraints either on the initialization or on the model parameters.
>
>     -  *"If these constraints were introduced here, do they affect the derived learning dynamics in any way?"*
>
>         **Response**: The interesting part is that they do not affect the learning dynamics, i.e., the learning dynamics here has almost the same form as that in the paper so that our method for solving the dynamics in the paper can be exactly applied here even without Assumption 3.1, i.e., the solution obtained here will be an exact one.
>
>        To see this point more clearly, we can rewrite the output of the model with the constraints as
>        $$
>         f_s : = \mathbf{v} _s \mathbf{H} _s \mathbf{w} _s = v _{s; \mathcal{N} _s + 1}\psi _s \Lambda _s w _{s; s}
>         $$
>         where $\mathbf{v} _s = v _{s; \mathcal{N} _s + 1}$ and $\mathbf{w} _s = w _{s; s}$ are 1-dimensional vectors while $\mathbf{H}_s = \psi_s\Lambda_s$ is a $1\times 1$ matrix that satisfies the idempotent-like condition. Thus the dynamics will be exactly the same as that in the paper (Eq.(9)), which can be solved using the method introduced in our paper and Theorem 3.1 can still be applied (a minor difference is that $a_s$ has a different value here).
>
>     - Regarding *"constraints on $v_s$"*
>
>        **Response:** We would like to first point out that linear self-attention is hard to perfectly adapt itself to arbitrary (linear) relationships between $x$ and $y$ without certain conditions. For example, Lu et al. (2024) showed that the in-context generalization error for linear self-attention can even be unbounded in certain limits.
>
>       Coming back to our setting, for the task $s$ and the input sequence $\Phi(s, X)$ with task strength $\Lambda_s$, the output of the model is
>       $$
>        \boldsymbol{v} _ s ^T  \Phi  \Phi ^T \boldsymbol{w} _ s  = \psi _s (v _{s; s}  w _{s; s} + \Lambda _s (v _{s; \mathcal{N} _s + 1} w _{s; s} + w _{s;  \mathcal{N} _s + 1}  v _{s; s}) + \Lambda _s^2 v _{s; \mathcal{N} _s + 1} w _{s; \mathcal{N} _s + 1}).
>       $$
>
>       We note that the first term does not depend on $\Lambda_s$ explicitly, meaning that it does not explore the relationship between $\phi(s, x)$ and $y$, thus it does not contribute to the in-context learning task. In addition, the last term depends on $\Lambda_s$ in a non-linear way, leading to an inappropriate dependence on the relationship between $x$ and $y$. Due to such terms, without any conditions, the linear self-attention is hard to perfectly adapt itself for new input sequences $\tilde{\Phi}(s, X)$ with different task strengths. This problem is caused by its structure. A possible way to achieve a perfect in-context adaption is imposing the constraints $v_{s;s} = w_{s; \mathcal{N}_s + 1} = 0$ (the constraints used in our example and Zhang et al. (2023)) to eliminate the aforementioned terms.
>
>       Therefore, we expect that an optimal in-context learner should set the first $\mathcal{N}_s$ components as $0$, since they contribute to many terms that are not necessary for the model to achieve perfect in-context adaption. In addition, although the feature vectors $\phi$ are ignored in the  calculation of the value vector, they are necessary for the calculation of the key vector. Thus we believe the constraint is acceptable.

---

> ### Author Response · Authors · 2024-11-29
> **Response, Part II/II**
>
> 2. - Regarding *"constraints on $v$"*
>
>         **Response:**(continuing) In fact, in our case, we can show that the output of the trained model (without any constraints, either on the initialization or on the model parameters) for any new input sequence $\tilde{\Phi}(s, X)$ with unseen task strength $\tilde{\Lambda}_s \neq \Lambda_s > 0$ is $\tilde{f}_s = \Lambda_s(1 + \tilde{\Lambda}_s\Lambda_s)^2 / (1 + \Lambda_s^2)^2$. Though $\tilde{f}_s$ is not equal to $\tilde{\Lambda}_s$ exactly, we can see that $\tilde{f}_s > \Lambda_s$ when $\tilde{\Lambda}_s > \Lambda_s$ and $\tilde{f}_s < \Lambda_s$ when $0< \tilde{\Lambda}_s < \Lambda_s$, i.e., the model attempts to adapt its prediction by exploring the in-context information though imperfectly.
>
>     Overall, by organizing the task in an in-context learning format, the linear self-attention can indeed perform in-context learning to adapt to new tasks (imperfectly without constraints).
>
> 3. In our opinion, the fact that the query token inside $H$ can be ignored for large $\psi$ is acceptable. From a more general point of view, the query token $\hat{\phi}$ appears in $H$ only in the form $\sum_{l = 1}^{\psi} \phi_{l}\phi_{l}^T + \hat{\phi}\hat{\phi}$, which is used to estimate the covariance matrix of $\phi$ (for both sparse feature regression and linear regression). When $\psi$ is large,
>     $\sum_{l = 1}^{\psi} \phi_{l}\phi_{l}^T / \psi$ is already a good estimation thus it is acceptable to ignore $\hat{\phi}$.
>
> Finally, we would like to highlight the main contributions of our work: a closed-form (approximate) solution of linear self-attention for the MSFR problem along the whole training trajectory and a thorough characterization of the neural scaling laws of linear self-attention. As the understanding of more complex models with self-attention, e.g., softmax self-attention coupled with non-linear MLP, is still a highly challenging and open problem from the theoretical perspective, our results could be significantly helpful towards this ultimate goal. In addition, our setup as well as the method for solving the learning dynamics can also be generalized to more scenarios. Therefore, we believe that our results should be of broad interest for both the study of self-attention and the study of learning dynamics of neural networks.

---

> > ### Comment · Reviewer_5pWj · 2024-12-02
> >
> > Thank you very much for engaging in this extensive discussion. It has greatly enhanced my understanding of your paper as well as the works of Zhang et al.(2023) and Lu et al.(2024).
> >
> > - Apologies for the confusion.
> > The point I wanted to make is that in the problem setups considered by Zhang et al.(2023) and Lu et al.(2024), the query token $x$ in the input sequence (or training prompt) does not contain any information about the task or context, while the problem setup in this paper allows the task to be inferred solely from the query token.
> > This raises some concerns because such a problem setup not only limits its applicability to practical in-context learning scenarios but also leaves it unclear whether the trained model is genuinely performing in-context learning. To address this, it becomes crucial to demonstrate explicitly that the model indeed engages in in-context learning under this setup.
> > Zhang et al.(2023) and Lu et al.(2024), for instance, thoroughly discuss this aspect in their respective works.
> >
> > - Regarding the example you presented, if the constraints on $v$ and $w$ were not introduced in the paper and appear here for the first time, it should be recognized that whether the model performs in-context learning in the absence of such constraints remains unclear.
> >
> > - As for Assumption 3.1, which allows the query token to be treated as an insignificant perturbation, I understand its use as a simplifying assumption for analysis. However, I believe it is important to note that Assumption 3.1 inherently excludes certain scenarios where the model does not perform in-context learning, such as when the input sequence consists only of query tokens.
> >
> > Overall, I acknowledge the significant achievement of deriving approximate learning dynamics for the MSFR problem, and also simplifying the problem setup for the sake of theoretical analysis is understandable.
> > However, I believe that when simplifying a problem setup, it is important to provide a compelling justification for the simplifications and ensure that the results offer insights for future studies that relax these simplifications.
> > At present, the presentation of the paper does not sufficiently address whether the proposed problem setup genuinely qualifies as in-context learning. I also believe that the line of discussion during this rebuttal period, such as whether the model is truly performing in-context learning when the constraints on $v$ and $w$ you introduced are incorporated into the problem setup, could be further deepened.
> > For these reasons, I have decided to maintain my current score.

---

> ### Author Response · Authors · 2024-12-03
> **Response and Summary of our main points**
>
> We thank the reviewer for all the replies during the discussion period. Below we would like to summarize our main points.
>
> The first point we would like to make is the main scope of our paper: deriving a closed-form (approximate) solution for linear self-attention along the whole training trajectory and theoretically characterizing the neural scaling laws of self-attention, both of which are first investigated in our paper to the best of our knowledge. Our results are remarkable since finding closed-form solutions for non-linear dynamics is generally a challenging and nontrivial problem. In addition, we have effectively indicated the generality of our problem setup by connecting it with several more general ones, suggesting how our setup can be reused in the future as well as its role as a helpful tool towards understanding self-attention and learning dynamics. Therefore, our results should not be underestimated.
>
> Although showing how and why in-context learning is effective is not our main point, we have justified that, by considering the similar constraints as in previous works (e.g., Zhang et al. (2023), Lu et al. (2024), and Wu et al.(2023)), linear self-attention can still be exactly solvable and the trained model can perfectly adapt itself to new tasks (any unseen task strength) at test time by exploring the in-context information, which cannot be achieved by solely relying on the query token. While the case without these constraints is unclear currently, our results confirm that the reduced linear self-attention is indeed performing in-context learning.
>
> More importantly, our results can inspire future works to relax certain assumptions to study various intriguing properties of training dynamics of self-attention, e.g., the type of solution that a self-attention with perfect adaption ability is likely to converge to without any constraints on the parameters (i.e., implicit bias of gradient descent for self-attention).
>
> Therefore, we believe the results presented in our paper should be of broad interest. Finally, we provide the following table to further indicate the scope and main contributions of our work.
>
> | | Solution during training|Neural scaling laws| Task complexity (diversity) scaling | Effectiveness of in-context learning|
> |:-------| :----: |:----: |:----: |:----: |
> |Zhang et al.(2023)| $\times$|$\times$|$\times$|$\checkmark$ with constraints|
> |Wu et al.(2023)|$\times$|$\times$|$\checkmark$|$\checkmark$ with constraints|
> |Lu et al.(2024)|$\times$|$\times$|$\checkmark$|$\checkmark$ with constraints|
> | This work |$\checkmark$|$\checkmark$|$\times$| $\times$ without constraints, $\checkmark$ otherwise|
>
> **Reference**
>
> Zhang et al.(2023). Trained Transformers Learn Linear Models In-Context.
>
> Lu et al.(2024). Asymptotic theory of in-context learning by linear attention.
>
> Wu et al.(2023). How Many Pretraining Tasks Are Needed for In-Context Learning of Linear Regression?

---

### Official Review · Reviewer_FJbQ · 2024-11-03

**Soundness:** 3
**Presentation:** 3
**Contribution:** 3
**Rating:** 6
**Confidence:** 4

**Summary:**

The paper performs a theoretical study on the problem of in-context learning for attention models, where a number of $(x,y)$ pairs compose the input string to the model. Compared to a standard transformer, the paper simplifies the architecture by analyzing a single self-attention layer. Thus, the learnable parameters are a matrix corresponding to the merged queries a keys' parameters, and the value's parameter matrix. The data distribution is composed of a power-law distribution over a number of tasks, a sparse feature extractor, and a "task strength" parameter that weights the sparse feature extractor to generate the labels. The sequence length and the task strength also follow a power law distribution.

The paper shows that the training dynamics of this model can be solved in closed form, assuming the sequence length is sufficiently large for all tasks, and keeping the zeroth order term of the expansion at large sequence length. The authors then use these closed-form solutions to get the test loss and devise neural scaling laws in various scenarios by varying the variables (model size, time, number of data points). They also cover the compute-optimal setting.

**Strengths:**

1. The exposition of the theory is overall clear. I particularly appreciated the "Procedure sketch", which guides the reader through the proof technique, from the non-tractability of the original problem to perturbation analysis and the several changes of variables involved. Also, I appreciate the structure and attention to detail of the appendix. It can be seen that the authors put considerable effort into making the proofs clear and well-organized.

2. The results in Section 4.1. recover well-known scaling laws previously observed both empirically and theoretically on simplified models. The results in Section 4.2 are more specific to the in-context setting and the toy model proposed and seem entirely new to me.

3. The theory is supported by a sufficient suite of experiments.

**Weaknesses:**

In many cases, it is unclear which set of assumptions are due to reasonable empirical observations in in-context learning and neural scaling laws, and which ones are mainly there to obtain tractable calculations (and if so, why so). For instance:

1. Sparsity. Why the feature extractor can only be {-1, 0, 1}. No justification is provided as to why this should be realistic. In fact, it seems precisely constructed to have the expansion of $H$ as the sum of two matrices, with the second one having the sequence length as a multiplicative prefactor.

2. Why should the task strength follow a power law distribution?

3. Why the distribution of the tasks should also be a power law?

I would appreciate it if the authors stated the purpose of the assumptions more explicitly. More broadly, this task seems specifically designed to get closed-form solutions and it is thus unclear how they can be re-used in future works.

Another issue is that the analysis is the *sequential* scaling of large context length limit, and then the other quantities (time, number of samples, model size). Assumption 3.1 on large context length allows the authors to use perturbation analysis to get the zero-th order correction of the expansion at large context length (which results in Theorem 3.1), and essentially drop all the higher order terms. However, in Section 4.2 they study the *joint* behavior of sequence length with the other scaling quantities. How do the authors know that other terms of the expansion would not be relevant if the joint limit were taken?

Other issues:
1. $f^0_s(t)$, $f^1_s(t)$ is defined in Theorem 3.1 but referred to earlier (line 303).
2. Why notation changes from d to D in Section 4.
3. Why approx in Eq. 15.
4. I feel this paper on the asymptotic theory of in-context learning should be cited [1].

[1] Asymptotic theory of in-context learning by linear attention (https://arxiv.org/abs/2405.11751)

**Questions:**

See weaknesses.

---

> ### Author Response · Authors · 2024-11-22
> **Response, Part I/II**
>
> We thank the reviewer so much for the valuable comments and appreciation of our work and efforts. Below we address your concerns and questions point by point.
>
> ---
>
> ### **Main Part**
>
> 1. *"In many cases, it is unclear which set of assumptions are due to ... if so why so)."*
>
>     **Response:** As suggested in the comment, we divide the assumptions into two groups, and discuss why these assumptions are acceptable to answer the questions raised in the comment.
>
>       - **Assumptions for tractable calculations.** The sparse feature assumption and large context sequence length assumption (Assumption 3.1) are sufficient for us to obtain the tractable solution of the learning dynamics. In particular, the sparse feature assumption is for the purpose of obtaining the idempotent-like structure of $H_s^0$, while the large context sequence length assumption allows us to perform the perturbation analysis.
>
>         Our assumption of the sparse feature is inspired by the random feature model under the source-capacity condition (e.g., Cui et al (2020)), i.e., eigenvalues of the feature covariance matrix $\Sigma = \text{diag}(\omega_1, \omega_2, \dots, \omega_d)$ obey a power law $\omega_k \propto k^{- \tau}$. When $\tau$ is large, the eigenvalue spectrum of $\Sigma$ decays very fast and only the first eigenvalue $\omega_1$ is significant. This can be the case when the feature obeys the Gaussian distribution with zero mean and the aforementioned covariance matrix $\Sigma$. In such case, our feature $\boldsymbol{\phi}(s, \boldsymbol{x})$ can exhibit a very similar behavior and, more importantly, lead to a similar feature covariance matrix. We thus believe that the sparse assumption is acceptable as a limiting case of the aforementioned setting.
>
>          In addition, since the covariance matrix is crucial for the in-context learning dynamics (because the covariance matrix determines the matrix $\boldsymbol{H}$ in Eq.(9) and $\boldsymbol{H}$ determines the learning dynamics), we expect that MSFR problem will show similar neural scaling laws as the regression problem under source-capacity condition. Please see **Appendix H.1** (page 32 of the revision) where we discuss this more closely and conduct numerical experiments to verify it (Fig.5, page 34).
>
>    - **Assumptions from empirical observations.** The assumptions that the task type and task strength follow power law distributions are from empirical observations. These assumptions are used to derive the neural scaling laws and are not needed for obtaining the solution of the dynamics.
>
>       In particular, the assumption for the power law distribution of the task type is based on the empirical observation for the neural scaling laws in Michaud et al. (2023), where they showed that language models can exhibit different skills that obey a power law distribution.
>
>       The assumption that the task strength $\Lambda_s \propto s^{-\gamma}$ is also from the random feature model under the source-capacity condition, while we note that this is not necessary for solution of linear self-attention and can be generalized to other types.
>
> 2. *"More broadly, this task seems specifically ... re-used in future works."*
>
>    **Response:** We thank the reviewer so much for this insightful question. To address this concern, we carefully discuss how the MSFR problem can be connected to or generalized to other types of tasks in the revision. For a detailed discussion, please see **Appendix H** (page 32 to 35 of the revision), while we summarize the generality of the MSFR problem to three types of tasks in the following.
>
>     - As discussed in our previous response to the assumption of the sparse feature, MSFR can be seen as a limiting case of the multitask in-context regression under the source-capacity condition when the eigenvalue spectrum decays very fast (**Appendix H.1**, page 32, with numerical experiments).
>
>    - If we are only interested in the solution to the zero-th order of $\epsilon_s$, i.e.,  $f_s^0(t)$, then the feature $\boldsymbol{\phi}(s, x)$ does not need to be exactly sparse as in Eq.(2) (line 124-125, page 3). Instead, we show that $f_s^0(t)$ can still be very exact when the feature $\boldsymbol{\phi}(s, x)$ is only approximately sparse, e.g., by adding a small random noise. We verify this point in Fig.6 (page 35). The corresponding discussion is presented in **Appendix H.2** (page 34).
>
>    - More broadly, for any task where the learning dynamics has the form of Eq.(9) (line 249-251, page 5) and that $\boldsymbol{H}_s$ is idempotent-like, i.e., $\boldsymbol{H}^2 = \mu \boldsymbol{H}$ for some constant $\mu$, then Theorem 3.1 can be applied and the solution $f_s^0(t)$ is now an exact solution rather than an approximate one. We also conduct numerical experiments to verify this (Fig.7, page 35). We believe that it might be an interesting future direction to explore learning tasks that have such form of dynamics. These discussions are presented in **Appendix H.3** (page 35).

---

> ### Author Response · Authors · 2024-11-22
> **Response, Part II/II**
>
> ### **Main Part (continuing)**
>
> 3. *"Another issue is that the analysis is the... if the joint limit were taken?"*
>
>    **Response:** We would like to first kindly mention that we have derived the solution to the first order of $\epsilon_s$ in Appendix G. In the following, we discuss separately why the perturbation analysis is still valid when taking the three different joint limits in Section 4.2.
>
>     - Only $N \to \infty.$ Whenever $N$ appears in the dynamics, it is in the form #$ _s/N$. Therefore, the limit of $N \to \infty$ will lead to #$_s/N \to s^{-\alpha}$, which does not affect other terms of the expansion.
>
>     - Only $t \to \infty$. The perturbation analysis does not make requirement on the time $t$, thus it will also be valid when $t \to \infty$, i.e., it is always $f_s^0(t)$ that dominates the learning dynamics.
>
>     - Both $t\to\infty$ and $N\to \infty.$ Based on the above arguments, we can safely take these two limits jointly.
>
> ---
>
> ### **Other issues**
>
> 1. Thanks for pointing this out. We add the definition of $f_s^0(t)$ and $f_s^1(t)$ (line 298) before Theorem 3.1.
>
> 2. We intended to use $D$ to denote that this is a symbol for "model size".
>
> 3. This is because Eq.(15) (Eq.(16) of the revision) is based on $f_s^0(t)$ and $f_s^0(t)$
> is an approximate solution.
>
> 4. Thanks for the suggestion. We add a detailed discussion of this work in Appendix A.3 (page 16) in the revision.
>
> ---
>
> **Reference**
>
> 1. Cui et al.(2021). Generalization error rates in kernel regression: The crossover from the noiseless to noisy regime.
>
> 2. Michaud et al. (2023). The quantization model of neural scaling.

---

> ### Comment · Reviewer_FJbQ · 2024-11-24
> **Response to Rebuttal**
>
> I thank the authors for the detailed response! On the rebuttal:
>
> 1. To my understanding (also, reading the author's response to Reviewer 5pWj), it seems that many of the assumptions are unnecessary for the results. Then I wonder why the author didn't aim for a more general type of theorem, and then specialize it to the empirical observations (which results in the theorem with the assumptions of the current version of the paper).
>
> 2. I went through the new Appendix H. It is now clearer how the proposed setting relates to Cui et al.,2022 and Lu et al. 2024. I thank the authors for that. In particular, it is a special case of the former work with large $\tau$ (which is my understanding that "sparsifies" the feature extractor in Cui et al, 2022). If possible, I would appreciate it if the authors included a comment in an updated version of the paper. Anyways, this sparsification is a crucial simplification compared to the general case: not only we need the assumption that the eigenvalues $\eta_1, \dots, \eta_p$ (in their notation) of the covariance matrix decay as a power law, but they do so with effectively infinite $\tau$ that exactly sparsifies the problem.
>
> 3. First of all, I would like to point out that it should be explicitly said that $\epsilon_s:= 1/\psi_s$ (due to Assumptions 3.1) around line 274. The perturbation analysis is an important step in the analysis, providing the crucial approximation that leads to the simplifications of the subsequent derivations. Coming back to the issue of the order of the limits, it is unclear why considering large $\psi$ expansion should not lead to an error that scales with the number of data points, or with the training time. Is it an artifact of the MSFR setup, which effectively makes all but one term of the sequence irrelevant? Also, I understand that the authors compute also the first-order term in the expansion of the dynamics, but it is unclear how this would affect the results of Section 4.2 (as far as I can see, the scaling laws results of 4.2 are obtained only with the zeroth order term $f_{0}$).
>
> Finally, I would suggest adding a more comprehensive set of notations to the Section "Notation and Definitions", including all the notation used. This would make it easier to navigate the paper.
>
> Overall, I am in favor of acceptance, but I have concerns about raising my score to an 8.

---

> ### Author Response · Authors · 2024-11-26
> **Response to "Response to Rebuttal"**
>
> We thank the reviewer very much for the prompt reply and nice suggestions! We answer your questions below.
>
> 1. Currently Theorem 3.1 is indeed applicable when removing the assumptions for the specific distributions for task type, sequence length, and task strength. Theorem 3.1 does not require these assumptions. As suggested by the reviewer, we state this point clearly below Theorem 3.1 in the updated version (line 345-347).
>
> 2. As suggested by the reviewer, we include a comment in the updated version to indicate that the MSFR problem with in-context learning can be seen as a limiting case of the in-context regression under source/capacity condition (line 182-184, page 4), i.e., the generalization of setup in Lu et al.(2024) with that of Cui et al.(2022).
>
> 3. We write explicitly $\epsilon_s = 1 / \psi_s$ around line 274 in the updated revision. Below we explain the validity of the perturbation analysis given that $H$ can be written as $H^0 + \epsilon H^1$ at small $\epsilon$ (large $\psi$ in the MSFR setup), where our discussion is from a general point of view that can cover the MSFR setup.
>
>     Specifically, we use $l_{(n)}:=l(f(H^{(n)}; \theta), \hat{y}^{(n)})$ to represent the loss function for the $n$-th data point, where $\theta$ is the model parameter. The dynamics is then $\dot{\theta} = - \frac{1}{N}\sum_{n = 1}^N \nabla_{\theta} l_{(n)}$. For each $n$, we can effectively expand $\nabla_{\theta} l_{(n)} = G_{(n)}^0 + \epsilon G_{(n)}^1 + \mathcal{O}(\epsilon^2)$ at small $\epsilon$, where $G_{(n)}^0$ is more significant than $\epsilon G^1_{(n)}$. Since the summation over $n$ does not affect this relation, we can write $\dot{\theta} = - G^0 - \epsilon G^1 := - \frac{1}{N}\sum_{n = 1}^{N} G^0_{(n)} - \frac{1}{N}\sum_{n = 1}^{N}\epsilon G^1_{(n)}$, where $G^0$ is more significant than $\epsilon G^1$. Then we can solve the dynamics with the perturbation analysis by writing $\theta = \theta^0 + \epsilon \theta^1$ and matching terms to different orders of $\epsilon$ on both sides of the dynamics equation. We note that the perturbation analysis is valid for a general $N$, which includes the large $N$ limit as a special case.
>
>       Furthermore, when solving the differential equation of $\theta$ ($g$ and $ h$ in the MSFR setup) with the perturbation method, we are aiming at the general analytical forms, which holds on any $t$. Hence, the limit $t \to \infty$, as a subset of the general case, holds by nature. For a more formal discussion of the perturbation analysis, we recommend the reference Murdock. (1991) and  Hinch. (1995) , which can support our use of perturbation analysis.
>
>      Finally, results in Section 4.2 are indeed obtained only with $f_s^0(t)$. We find that $f_s^0(t)$ can match the numerical experiments well at large $\psi_s$, while it might also be interesting for future works to characterize the neural scaling laws with both $f_s^0(t)$ and $f_s^1(t)$.
>
> 4. As suggested by the reviewer, we add a definition and notation table in Appendix J with the corresponding index to navigate the paper.
>
> ---
>
> **Reference**
>
> 1. James Murdock. Perturbations: theory and methods. 1991.
> 2. E.J. Hinch. Perturbation Methods. 1995.

---

### Author Response · Authors · 2024-11-22
**General Response to All Reviewers and Summary of Changes of the Revision**

We thank all reviewers a lot for the valuable comments and insightful questions and suggestions.

To answer questions and address concerns of the reviewers, we make the corresponding changes in the rebuttal revision, which will be summarized as follows, including generality of MSFR setup, additional numerical experiments on softmax self-attention, addition discussion on related works, and other points. For clarity, all changes made in the revision are highlighted by blue color fonts.

---

### **Generality of  MSFR Setup**

As suggested by Reviewer **FJbQ** and **9RQV**, we discuss the generality of the MSFR problem more comprehensively in Appendix H (start from page 32 of the revision). Specifically, we show that:

1. The MSFR problem considered in this paper can be seen as a limiting case of the (multitask) in-context regression under the source/capacity condition (i.e., with structured covariance) when the eigenvalue spectrum of the feature covariance matrix decays fast. And we expect that the neural scaling laws derived in the MSFR problem can be generalized to the (multitask) in-context regression under the source/capacity condition. This is verified by numerical experiments (Fig.5 in page 34) on softmax self-attention where it exhibits similar scaling laws with respect to time and the optimal compute as in the MSFR problem. These discussions are presented in **Appendix H.1** (page 32 to 34).

2. If we are only interested in the solution to the zero-th order of $\epsilon_s$, i.e.,  $f_s^0(t)$, then the feature $\boldsymbol{\phi}(s, \boldsymbol{x})$ does not need to be exactly sparse as in Eq.(2) (line 124-125, page 3). Instead, we show that $f_s^0(t)$ can still be very exact when the feature $\boldsymbol{\phi}(s, \boldsymbol{x})$ is only approximately sparse, e.g., by adding a small random noise to the original $\boldsymbol{\phi}(s, \boldsymbol{x})$. We verify this point in Fig.6 (page 35). The corresponding discussion is presented in **Appendix H.2** (page 34).

3. More broadly, we show that, for any task where the learning dynamics has the form of Eq.(9) (line 249-251, page 5) and that $\boldsymbol{H}$ is idempotent-like, i.e., $\boldsymbol{H}^2 = \mu \boldsymbol{H}$ for some constant $\mu$, then Theorem 3.1 can be applied and the solution $f_s^0(t)$ is now an exact solution rather than an approximate one. We also conduct numerical experiments to verify this (Fig.7, page 35). We believe that it might be an interesting future direction to explore learning tasks that have such form of dynamics. These discussions are presented in **Appendix H.3** (page 35).

---

### **Additional Numerical Experiments on Softmax Self-Attention**

As suggested by Reviewer **7YzX** and **9RQV**, we supplement numerical experiments to examine neural scaling laws for the softmax attention, where, for completeness, we use the parameterization $W_K^T W_Q$ rather than a single merged $W_{KQ}$ in all the experiments. These additional experiments are presented in **Appendix I** (page 36 to 38). Specifically:

1. We show that softmax self-attention still displays the same neural scaling laws as linear self-attention when it is trained by gradient descent (Fig.8, page 36). We report the results in **Appendix I.1** (page 36 to 37).

2. To examine the effects of optimization algorithms on neural scaling laws in the MSFR problem, we also train softmax self-attention with AdamW. We report the results in **Appendix I.2** (page 37 to 38).

    In particular, we show that, while AdamW still leads to similar neural scaling laws with respect to model size and data size (Fig.10a and Fig.10b, page 38), it inevitably contributes to different neural scaling laws with respect to time $t$ and the optimal compute (Fig.10c and Fig.10d, page 38) compared to gradient descent. This is because AdamW has a very different learning dynamics (e.g., it could converge faster). The observation that optimization algorithms affect the neural scaling laws with respect to time (and thus the optimal compute) aligns with the empirical evidence in Hoffmann et al. (2022), where they revealed that AdamW shows a different test loss behavior against the training time when compared to Adam.

---

### **Additional Discussion on Related Works**

As suggested by Reviewer **FJbQ, 7YzX**, and **9RQV**, we make a more extensive discussion on related works about learning dynamics of neural networks (Appendix A.1, page 15), neural scaling laws (Appendix A.2, page 15), and analysis of in-context linear regression (Appendix A.3, page 16). We present the corresponding discussion in **Appendix A** due to the lack of space.

---

### **Additional Points**

Throughout the paper, we now state accurately "linear self-attention" when we are discussing linear self-attention, rather than simply stating "self-attention".

---

### Meta-Review · Area_Chair_LUL5 · 2024-12-17

**Metareview:**

The authors study a toy setup with of in-context learning with a linear self-attention model, where the training dynamics given by an ODE system can be solved via a Riccati equation. While some simplifying assumptions and modifications are chosen, most of the reviewers tend to agree that it is nice to achieve a solvable model.

The two main criticisms that seems unresolved are

1. Building on the point of reviewer FJbQ and 9RQV on the large $\psi$ or small $\epsilon$ expansion, the accumulation of the correction terms could be exponential in training time unless otherwise justified. This limitation, while technically challenging to fully resolve, could lead to different limits depending on the exact scaling or order of limits.

2. Reviewer 5pWj had a long discussion on whether the problem setup qualifies as in-context learning. That being said, the reviewer still agrees that this modification is intended to simplify the setup to achieve a solvable model.

Altogether, I believe both of these issues are not critical for a first attempt at a solvable model, as the authors have always intended on studying a toy model to begin with. Based on the extended discussions and overall positive reviews, I would recommend accepting this manuscript.

**Additional Comments On Reviewer Discussion:**

The longest discussion, while inconclusive, was with reviewer 5pWj on whether or not the problem setup is considered in-context learning. The criticism has merit, although I believe this is a rather distracting point to the main goals of this paper, which is to simplify the transformer until it is solvable.

I believe the criticism of reviewer FJbQ and 9RQV on the precision of small $\epsilon$ expansions is actually more serious, as it can affect the end result based on how limits are taken. While I would prefer if the authors can make the limit more precise as a result, this is not a critical concern that would derail the overall positive reviews.

---

### Decision · Program_Chairs · 2025-01-22

Accept (Poster)